# Patterned invagination prevents mechanical instability during gastrulation

Bruno C. Vellutini[1 ✉], Marina B. Cuenca[1], Abhijeet Krishna[1,2,3], Alicja Szałapak[1,2,3], Carl D. Modes[1,2,3] & Pavel Tomancak[1,2,3 ✉]

Mechanical forces are crucial for driving and shaping tissue morphogenesis during embryonic development[1–3]. However, their relevance for the evolution of development remains poorly understood[4]. Here we show that an evolutionary novelty of fly embryos—the patterned embryonic invagination known as the cephalic furrow[5–7]—has a mechanical role during *Drosophila* gastrulation. By integrating in vivo experiments and in silico simulations, we demonstrate that the head–trunk boundary of the embryo is under increased compressive stress due to the concurrent formation of mitotic domains and germ band extension and that the cephalic furrow counteracts these stresses, preventing mechanical instabilities during gastrulation. Then, by comparing the genetic patterning of species with and without the cephalic furrow, we find evidence that changes in the expression of the transcription factor *buttonhead* are associated with the evolution of the cephalic furrow. These results suggest that the cephalic furrow may have evolved through the genetic stabilization of morphogenesis in response to the mechanical challenges of dipteran gastrulation. Together, our findings uncover empirical evidence for how mechanical forces can influence the evolution of morphogenetic innovations in early development.

Morphogenesis is a process that shapes embryonic tissues through cell-generated mechanical forces[1,2]. These forces drive tissue movements that push and pull on their neighbouring regions. Such physical interactions provide essential information to embryonic cells and ultimately shape the morphology of tissues and organs[3]. Despite the importance of mechanical forces to embryogenesis, their role in the evolution of morphogenesis remains elusive[4]. To investigate the evolutionary interplay between genetics and mechanics, we studied an enigmatic epithelial fold that forms at the head–trunk boundary of flies during gastrulation: the cephalic furrow[5,7].

The cephalic furrow forms under strict genetic control. In *Drosophila*, it begins as paired lateral indentations that invaginate to form a deep epithelial fold at the boundary between the head and trunk[5–7]. Its position along the anteroposterior axis is determined by the zygotic expression of two transcription factors, *buttonhead* (*btd*) and *even skipped* (*eve*), which overlap at the head–trunk boundary by a few rows of blastoderm cells[8]. These initiator cells drive the invagination by shortening along the apicobasal axis via lateral myosin contractility[9], whereas the mechanical coupling between cells ensures the propagation of a morphogenetic wave of tissue folding[9,10]. The resulting fold spans the entire lateral surface, making the cephalic furrow a landmark of *Drosophila* gastrulation[5,6].

Unlike other embryonic invaginations, the cephalic furrow has no obvious function during development. Despite its prominence and patterned formation, it does not give rise to specific structures and unfolds, leaving no trace[5]. It has been proposed that the invagination may serve as temporary storage[11] or an anchor for tissues during gastrulation[12,13], but these hypotheses have not been investigated in vivo or in a phylogenetic context. The cephalic furrow is an evolutionary novelty of dipteran flies[14], making it an ideal model for investigating the evolution of patterned morphogenetic processes in embryonic development.

Our work integrates genetics and mechanics to uncover the developmental role and patterning evolution of the cephalic furrow. We first show that the absence of the cephalic furrow increases the mechanical instability of the blastoderm epithelium and that the primary sources of mechanical stress are the formation of mitotic domains and extension of the germ band. Then, we demonstrate that the formation of the cephalic furrow absorbs these compressive stresses and mitigates the epithelial instability at the head–trunk boundary. This suggests that the cephalic furrow has a mechanical role during gastrulation. Next, we compared the expression of head–trunk genes between species with and without the cephalic furrow. We found that the typical *btd*–*eve* overlap, which is required for the specification of initiator cells in *Drosophila*, is absent in species without the cephalic furrow. This indicates that changes in the expression of *btd* at the head–trunk boundary are associated with the evolution of the cephalic furrow. Together, these results suggest that the establishment of a novel gene expression territory may have enabled the genetic stabilization of cephalic furrow morphogenesis and that the underlying selective pressure may have been the mechanical instability during gastrulation.

[1]Max Planck Institute of Molecular Cell Biology and Genetics, Dresden, Germany. [2]Center for Systems Biology Dresden, Dresden, Germany. [3]Cluster of Excellence Physics of Life, Technische Universität Dresden, Dresden, Germany. ✉e-mail: vellutini@mpi-cbg.de; tomancak@mpi-cbg.de

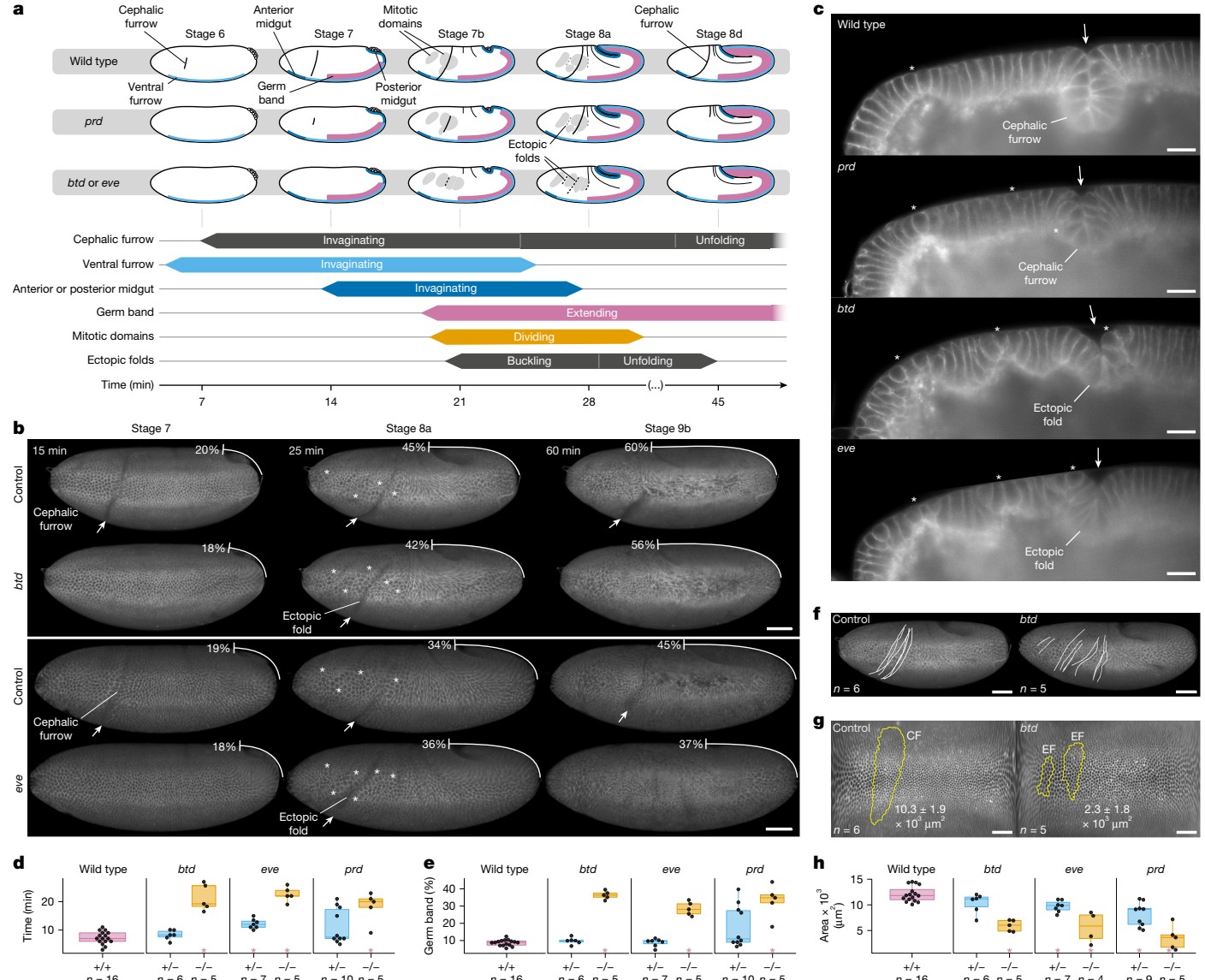

**Fig. 1 | Formation of ectopic folds in cephalic furrow mutants. a**, Timing of developmental events in wild type and *prd*, *btd* and *eve* mutants. **b**, Lateral view of *btd* and *eve* mutants during gastrulation. Controls are heterozygote siblings. The percentages indicate the extent of germ band extension. Arrows indicate tissue folds; asterisks indicate mitotic domains. Scale bars, 50 μm. **c**, Profile view of the head–trunk epithelium around stage 8. Scale bars, 20 μm. **d**, Timing of wild-type cephalic furrow formation compared with *btd* (*P* = 0.283), *eve* (*P* < 0.001) and *prd* heterozygotes, and to ectopic folds in *btd* (*P* = 0.001), *eve* (*P* = 0.001) and *prd* (*P* = 0.003) homozygotes. **e**, Percentage of germ band extension at cephalic furrow formation in wild type compared with ectopic

folding in *btd*, *eve* and *prd* homozygotes (*P* < 0.001). **f**, Position variability of cephalic furrow formation and ectopic folding in *btd* heterozygotes and homozygotes, respectively. Scale bars, 50 μm. **g**, Folded area (yellow outline) of the cephalic furrow (CF) and ectopic folds (EFs) in *btd* mutants. The numbers show average area and standard deviation. Cartographic projections of lateral views. Scale bars, ≈50 μm. **h**, Total folded area of wild-type cephalic furrow compared with *btd* (*P* = 0.133) *eve* (*P* < 0.001) and *prd* (*P* < 0.001) heterozygotes, and to ectopic folds in *btd*, *eve* and *prd* homozygotes (*P* < 0.001 for each). The membrane marker in panels **b**,**c**,**f**,**g** is Gap43–mCherry. Embryonic staging is based on standard developmental tables[32,33].

## Analyses of cephalic furrow mutants

To understand the physical consequences of perturbing the formation of the cephalic furrow in *Drosophila*, we investigated the tissue dynamics at the head–trunk boundary in known cephalic furrow mutants: *btd*, *eve* and *prd*[8,15]. To this end, we generated fluorescent lines carrying a loss-of-function allele of these genes and imaged the embryos in toto using light-sheet microscopy with high temporal resolution to uncover differences in developmental events and characterize mutant phenotypes during gastrulation (Fig. 1a).

## Formation of ectopic folds

We first analysed the behaviour of initiator cells in the three mutant backgrounds. The typical shortening of initiators is perturbed in *prd*

mutants and severely disrupted in *btd* and *eve* mutants (Extended Data Fig. 1, Supplementary Video 1 and Supplementary Note 1). Disruption of initiator cell shortening is associated with a strong phenotype in *btd* and *eve* mutants: the formation of late epithelial folds near the canonical cephalic furrow site (Fig. 1b,c and Extended Data Fig. 1a). These ectopic folds—as they will be referred to from here on—have been previously observed in *btd* and *eve* mutants[8,9]. However, the mechanisms causing their formation and their relation to the cephalic furrow have not been investigated.

Ectopic folds appear around the head–trunk boundary of *btd* and *eve* embryos, about 20 min after gastrulation, when the germ band is extended 35% of the egg length (Table 1, Fig. 1a–e and Supplementary Videos 2 and 3). Although they can superficially resemble a cephalic furrow, ectopic folds lack the typical symmetry and formation

Table 1 | Relative timing differences between developmental events in cephalic furrow mutants

| Stock | n | Zygosity | n | Event | TAG (min) | GBE (%) |
|---|---|---|---|---|---|---|
| Wild type | 16 | +/+ | 16 | CF | 7.2±2.2 | 8.7±1.7 |
| | | | | MD | 14.6±2.4 | 21.4±1.9 |
| btd | 11 | +/− | 6 | CF | 8.2±1.7 | 9.8±1.9 |
| | | | | MD | 14.6±3.7 | 22.1±4.2 |
| | | −/− | 5 | EF | 21.3±4.7 | 36.3±2.4 |
| | | | | MD | 17.0±3.7 | 28.1±2.3 |
| eve | 12 | +/− | 7 | CF | 12.2±1.7 | 8.8±2.3 |
| | | | | MD | 18.2±1.6 | 19.9±4.1 |
| | | −/− | 5 | EF | 22.6±2.6 | 28.5±3.9 |
| | | | | MD | 16.9±1.1 | 19.0±2.7 |
| prd | 15 | +/− | 10 | CF | 11.3±6.2 | 18.0±12.2 |
| | | | | MD | 12.8±1.1 | 20.4±3.8 |
| | | −/− | 5 | CF | 18.2±5.4 | 33.0±9.5 |
| | | | | MD | 13.0±2.0 | 23.4±3.0 |
| btd+eve | 23 | +/− | 13 | CF | 10.4±2.6 | 9.2±2.1 |
| | | | | MD | 16.6±3.2 | 20.9±4.1 |
| | | −/− | 10 | EF | 21.9±3.6 | 32.4±5.1 |
| | | | | MD | 17.0±2.6 | 23.6±5.3 |

We measured the time after gastrulation (TAG) and the percentage of germ band extension (GBE) at the moment of formation of the cephalic furrow (CF), mitotic domains (MDs) and ectopic folds (EFs). For a more general comparison, we also pooled the data for mutants in which the cephalic furrow is absent (btd+eve).

Table 2 | Surface area of ectopic folds in cephalic furrow mutants

| Stock | n | Zygosity | n | Type | n | Area ×10³ (µm²) |
|---|---|---|---|---|---|---|
| Wild type | 16 | +/+ | 16 | CF | 16 | 11.4±1.2 |
| | | | | EF | 14 | 0.6±0.4 |
| btd | 11 | +/− | 6 | CF | 6 | 10.3±1.9 |
| | | −/− | 5 | EF | 12 | 2.3±1.8 |
| eve | 11 | +/− | 7 | CF | 7 | 9.7±1.1 |
| | | −/− | 4 | EF | 7 | 3.2±2.1 |
| prd | 14 | +/− | 9 | CF | 9 | 7.6±2.6 |
| | | | | EF | 7 | 0.8±0.5 |
| | | −/− | 5 | CF | 1 | 7.2 |
| | | | | EF | 9 | 1.3±0.9 |

dynamics of the wild-type invagination[7]. Their cleft is loose and asymmetric without wedge-shaped or arched-shaped cells (Fig. 1c and Supplementary Videos 4 and 5); they occupy one-quarter of the area and one-fifth of the depth of the cephalic furrow (Tables 2 and 3, Fig. 1g,h and Extended Data Fig. 2b,f); and they also fold and unfold faster (Fig. 1a,b, Extended Data Figs. 1a and 2c–e and Supplementary Video 6). In addition, ectopic folding is more variable than cephalic furrow formation, as the position of ectopic folds along the head–trunk region differs between individual mutant embryos (Fig. 1f, Extended Data Fig. 2a, Supplementary Videos 7 and 8 and Supplementary Fig. 1). Together, these differences in morphology, timing and position suggest that ectopic folds and the cephalic furrow form via distinct mechanisms.

Ectopic folds can also appear in heterozygote and wild-type embryos, but with lower frequencies and smaller sizes. Although more than 92% of btd and eve homozygote embryos show one or more ectopic folds (2.2 ± 0.4 and 1.8 ± 0.6, respectively; Extended Data Fig. 2h and Supplementary Video 8), between 18% and 27% of heterozygotes and about 78% of wild-type embryos form an ectopic fold anterior or posterior to the cephalic furrow during gastrulation (Table 4 and Extended Data Fig. 2i,j). The area of these ectopic folds, however, is significantly smaller, about one-quarter of the area of ectopic folds in btd and eve embryos (Table 2 and Extended Data Fig. 2g,k–n). These observations provide evidence that the head–trunk interface of Drosophila is a region prone to the formation of ectopic folds during gastrulation and that the absence of the cephalic furrow increases the magnitude of these ectopic folding events.

### Evidence of tissue compression

Although we cannot exclude the possibility that defects in patterning may contribute to the formation of ectopic folds, the variability in ectopic folding suggests that, unlike the cephalic furrow, the ectopic folds are not under genetic control and form as the result of physical interactions in the tissue. Our analysis shows that the formation of ectopic folds coincides spatially and temporally with two other processes of gastrulation: the expansion of mitotic domains and the extension of the germ band (Fig. 1a,c).

Mitotic domains are groups of blastoderm cells that divide in synchrony during nuclear cycle 14, first appearing on the head of the embryo 20 min after gastrulation[6]. In btd and eve mutants, ectopic folds form between or adjacent to mitotic domains (Fig. 2a). When mitotic cells begin to divide, they lose their basal attachment, round up at the apical side and more than double their apical area during anaphase (Supplementary Fig. 2a). This apical expansion compresses the adjacent, non-dividing cells, which are the first to fold inwards (Fig. 2b). Mitotic expansions always precede ectopic folding (Table 1, Fig. 2c,d and Extended Data Fig. 1a). This suggests that the formation of mitotic domains may generate buckling instability in the monolayer epithelium, contributing to the appearance of ectopic folds.

To estimate the mechanical forces acting on the tissues, we measured the rate of tissue deformation (strain rate) at the head–trunk and trunk–germ regions using particle image velocimetry (Fig. 2f). At the head–trunk interface, control embryos exhibit a peak of strain rate that correlates with the late phase of cephalic furrow formation, when the initiator cells move into the yolk (Fig. 2g, Supplementary Fig. 2b and Supplementary Video 9). This is absent in btd mutants. Instead, mutants show a higher peak of strain rate that coincides with the maximum expansion of mitotic domains during telophase and the folding of the tissue (Fig. 2g, Supplementary Fig. 2b and Supplementary Video 9). This suggests that mitotic expansions can generate a substantial amount of local tissue deformation.

At the trunk–germ interface, the cells between a posterior mitotic domain and the extending germ band become increasingly anisotropic (Fig. 2f). The strain rate in this region steadily increases over time (Supplementary Fig. 2c), suggesting that the tissue is under increased compression. To test this hypothesis, we performed laser cuts at the trunk–germ interface of wild-type embryos. We ablated the apical membrane of multiple cells (3–4) with cuts oriented orthogonal to the direction of the germ band extension and then tracked the distance between non-ablated cells on each side of the cut (Fig. 2h). This distance remains constant in control embryos, but it decreases in ablated embryos immediately after the cut (Fig. 2h). This result suggests that the tissue may be 'collapsing on itself', which supports the hypothesis that the trunk–germ interface is under compression from the extending germ band.

Together, these analyses suggest that the formation of mitotic domains and germ band extension are potential sources of compressive stress that could contribute to the formation of ectopic folds at the head–trunk boundary during gastrulation.

### Mitotic domains and germ band

To test the role of mitotic domains and germ band extension in the mechanical stability of the blastoderm, we performed a series of perturbation experiments in vivo.

**Table 3 | Maximum depth of ectopic folds in cephalic furrow mutants**

| Stock | n | Zygosity | n | Type | n | Depth (µm) |
|---|---|---|---|---|---|---|
| *btd* | 39 | +/− | 32 | CF | 52 | 71.6±8.0 |
| | | | | EF | 6 | 32.5±3.6 |
| | | −/− | 7 | EF | 28 | 52.5±12.1 |
| *eve* | 24 | +/− | 20 | CF | 34 | 59.0±6.8 |
| | | | | EF | 4 | 36.1±4.4 |
| | | −/− | 4 | EF | 15 | 42.1±11.7 |

We first asked whether mitotic expansions are required for the formation of ectopic folds. To that end, we generated a double-mutant line lacking both cephalic furrow and mitotic domains, using the loss-of-function alleles of *btd* and *string* (*stg*), the *cdc25* phosphatase orthologue that regulates the formation of mitotic domains in *Drosophila*[16]. The absence of mitotic domains in *stg* mutants does not affect the formation of the cephalic furrow or other gastrulation movements[16] (Extended Data Fig. 3a,b and Supplementary Videos 10 and 11). However, the absence of mitotic domains in *btd–stg* double mutants suppresses the formation of ectopic folds (Fig. 2e, Extended Data Fig. 3c,d and Supplementary Videos 12 and 13). This indicates that mitotic expansions are necessary for the appearance of ectopic folds in cephalic furrow mutants.

Next, we asked whether the extension of the germ band is required for the formation of ectopic folds. To prevent the germ band from extending, we cauterized a patch of posterodorsal tissue at the onset of gastrulation to mechanically attach it to the vitelline envelope (Fig. 2i and Extended Data Fig. 3e). Blocking the germ band extension in wild-type embryos does not prevent the formation of the cephalic furrow (Extended Data Fig. 3e), as initiator cells generate tension in the head–trunk epithelium autonomously (Supplementary Fig. 3); the invagination in cauterized wild-type embryos is about 15% shallower, but the difference was not statistically significant compared with controls (Extended Data Fig. 3f). By contrast, when we block the germ band extension in *btd* and *eve* mutants, no ectopic folds appear at the head–trunk interface (Fig. 2i and Supplementary Videos 14–16). Mitotic expansions still compress the non-dividing neighbouring cells, but no buckling occurs (Fig. 2j). The epithelium of cauterized mutant embryos undergoes less deformation and ectopic folding events than non-cauterized samples (Fig. 2k,l and Extended Data Fig. 3g,h). These experiments reveal that germ band extension is necessary for the appearance of ectopic folds in cephalic furrow mutants.

Overall, we conclude that neither the mitotic domains nor the germ band alone can induce the formation of ectopic folds, but when both events occur at the same developmental stage, the epithelial monolayer of the embryo becomes unstable and more prone to buckling (Fig. 2m).

## Physical model of folding dynamics

To determine the relative contribution of mitotic domains and germ band extension as sources of mechanical stress at the head–trunk boundary, we created a physical model of the blastoderm and ran in silico simulations of the tissue mechanics in mutant and wild-type conditions.

### Design and general properties

Our model represents an epithelial monolayer confined inside a rigid shell (Fig. 3a, Extended Data Fig. 4a and Supplementary Note 2). The monolayer is composed of equidistant particles connected by springs. Mitotic domains are regions with a higher density of particles, which tend to expand; the cephalic furrow is a region with an intrinsic negative curvature, which tends to invaginate; and the germ band is a static limit in the posterior end of the monolayer (Fig. 3b). In this tissue, the total energy per unit length ($W_T$) is the sum of stretching ($W_s$) and bending ($W_b$) energy components, each associated with a stretching ($K_s$) and a bending ($K_b$) rigidity. We combined these two parameters into a single, dimensionless bending rigidity ($K_b^*$) and used it as the main parameter in the model and simulations (Fig. 3c).

**Table 4 | Folding statistics in cephalic furrow mutants**

| Stock | n | Zygosity | n | Type | n | Frequency (%) | A (%) | M (%) | P (%) | Folds |
|---|---|---|---|---|---|---|---|---|---|---|
| Wild type | 36 | +/+ | 36 | CF | 36 | 100 | 0 | 100 | 0 | 1.0±0.0 |
| | | | | EF | 28 | 77.8 | 42.9 | 0 | 71.4 | 1.1±0.3 |
| *btd* | 46 | +/− | 33 | CF | 33 | 100 | 0 | 100 | 0 | 1.0±0.0 |
| | | | | EF | 6 | 18.2 | 0 | 0 | 100 | 1.0±0.0 |
| | | −/− | 13 | CF | 0 | 0 | 0 | 0 | 0 | – |
| | | | | EF | 12 | 92.3 | 50 | 100 | 75 | 2.2±0.4 |
| *eve* | 36 | +/− | 26 | CF | 26 | 100 | 0 | 100 | 0 | 1.0±0.0 |
| | | | | EF | 7 | 26.9 | 14.3 | 0 | 85.7 | 1.0±0.0 |
| | | −/− | 10 | CF | 0 | 0 | 0 | 0 | 0 | – |
| | | | | EF | 10 | 100 | 40 | 70 | 90 | 1.8±0.6 |
| *prd* | 40 | +/− | 26 | CF | 26 | 100 | 0 | 100 | 0 | 1.0±0.0 |
| | | | | EF | 7 | 26.9 | 71.4 | 0 | 57.1 | 1.3±0.5 |
| | | −/− | 14 | CF | 7 | 50.0 | 0 | 100 | 0 | 1.0±0.0 |
| | | | | EF | 10 | 71.4 | 50 | 80 | 70 | 1.9±0.8 |
| *stg* | 46 | +/− | 33 | CF | 33 | 100 | 0 | 100 | 0 | 1.0±0.0 |
| | | | | EF | 12 | 36.4 | 8.3 | 0 | 91.7 | 1.0±0.0 |
| | | −/− | 13 | CF | 13 | 100 | 0 | 100 | 0 | 1.0±0.0 |
| | | | | EF | 3 | 23.1 | 0 | 0 | 100 | 1.0±0.0 |

We calculated the percentage of embryos having a cephalic furrow or ectopic folds for each stock and genotype (frequency), including the position of folding along the head–trunk boundary (anterior (A), middle (M) and posterior (P)). In addition, we calculated the average number of folds per embryo side (folds). For example, 28 out of 36 wild-type embryos show ectopic folds (77.8%); 42.9% of these embryos have folds at the anterior region and 71.4% form them posterior to the head–trunk boundary; each embryo forms 1.1±0.3 folds on each side. The n includes datasets imaged from the lateral and dorsal sides.

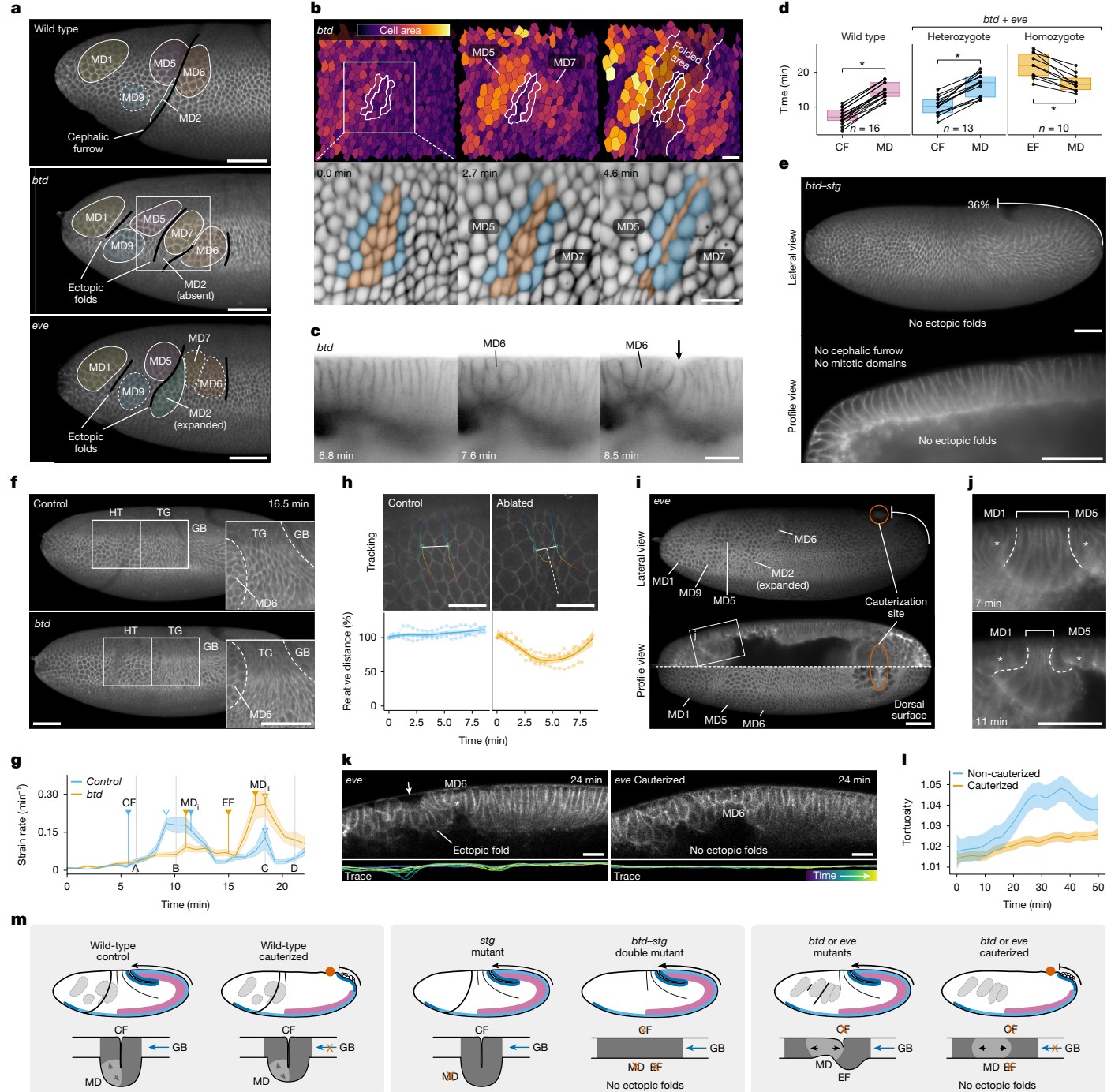

**Fig. 2 | Role of mitotic domains and germ band in ectopic folding. a**, Folding positions (black lines) relative to mitotic domains (MD) in wild-type, *btd* and *eve* embryos. Scale bars, 50 µm. **b**, Apical cell area during ectopic folding. The white outlines highlight a subset of non-dividing (orange) and dividing (blue) cells. Scale bars, ≈20 µm. **c**, Mitotic expansion preceding ectopic folding (arrow) in the *btd* mutant. Scale bar, 20 µm. **d**, Time of cephalic furrow (CF) and ectopic fold (EF) formation relative to mitotic expansion in wild-type (*P* < 0.001), heterozygote (*P* = 0.002) and homozygote (*P* = 0.002) embryos. **e**, Lateral and profile views of *btd–stg* double mutants. Scale bars, 50 µm. **f**, Tissue compression around the head–trunk (HT) and trunk–germ (TG) regions (white outlines) in the *btd* mutant. Scale bars, 50 µm. GB, germ band. **g**, Strain rate at the head–trunk region in *btd* heterozygotes (*n* = 3) and homozygotes (*n* = 3; combined isotropic and anisotropic). Filled triangles denote the formation of CF, MD and EFs, and empty triangles indicate strain rate peaks. A–D refer to frames from Supplementary Fig. 2b. MD$_i$, metaphase; MD$_{ii}$, telophase. **h**, Dynamics after trunk–germ laser cuts in wild-type embryos. The tracks (rainbow) show the distance between cell vertices (solid white line) in control (*n* = 3) and ablated (*n* = 3) embryos. The dashed white line indicates the cut location. Scale bars, 20 µm. The line and shaded area represent mean and 95% confidence interval, respectively (**g**,**h**). **i**, Germ band cauterization (orange circle) in the *eve* mutant under light-sheet microscopy. Scale bar, 50 µm. **j**, Compressed non-dividing cells between mitotic domains (from panel **i**). Scale bar, 50 µm. **k**, Germ band cauterizations in *eve* mutants with traces of epithelial deformations over time. Scale bars, 20 µm. **l**, Tortuosity of epithelial traces in non-cauterized (*n* = 3) and cauterized (*n* = 4) *eve* mutants. **m**, Summary of the live experiments in cephalic furrow mutants.

To characterize the energy dynamics of the system, we ran simulations using a single bending rigidity value and percentage of germ band extension. When a fold begins to form, the bending energy increases, releasing a large amount of stretching energy, which decreases the total energy of the system (Fig. 3d). The increase in bending energy coincides with a rapid deepening of the fold. Once

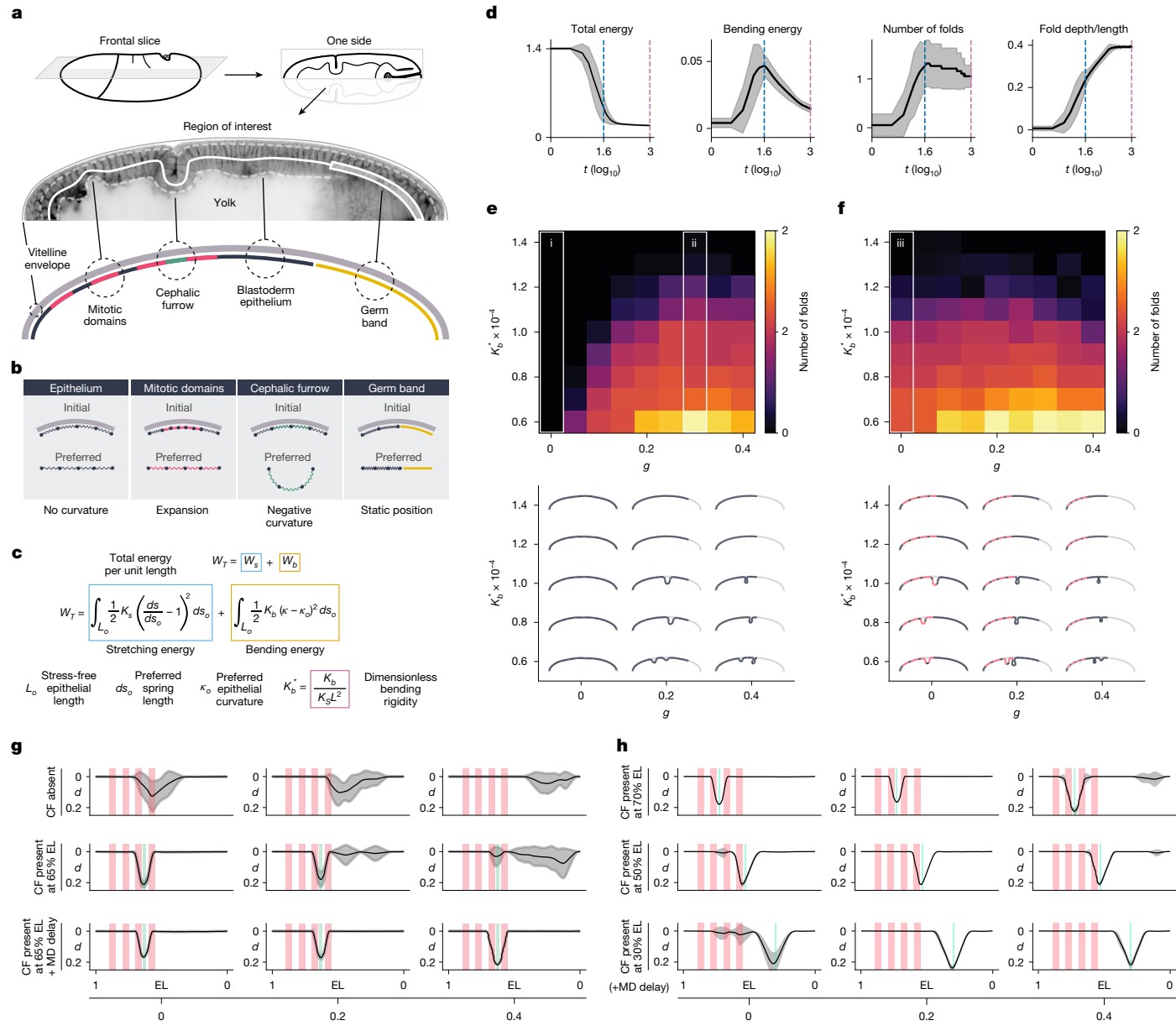

**Fig. 3 | Model and simulations of folding mechanics. a**, Region of interest of the model. **b**, States of the different components based on particles connected by springs. **c**, Energy equation describing the stretching and bending components and the dimensionless bending rigidity. **d**, Energy and folding dynamics in simulations. The black line and shaded area represent the mean and standard deviation across simulations ($t$) ($n = 20$). The dashed blue line indicates the peak of bending energy, and the dashed pink line denotes the last iteration. Energy values are normalized by the initial total energy. **e**, Parameter sweep for cephalic furrow mutants without mitotic domains. The heatmap shows the average number of folds for different bending rigidities ($K_b^*$) and percentages of germ band extension ($g$). Outlined in white are conditions without folding (i) and with most folding events (ii). Representative simulations are rendered below. **f**, Parameter sweep for cephalic furrow mutants with mitotic domains. Outlined in white is a condition with folding events without germ band extension (iii). **g**, Simulations testing the effect of the cephalic furrow on ectopic folding in three conditions: only mitotic domains (top row), mitotic domains and cephalic furrow (middle row), and delayed mitotic domains and cephalic furrow (bottom row). The added delay mimics the relative timing in wild type. $t_{MD} = 1$ corresponds to $10^5$ computational timesteps. The cephalic furrow is $\kappa_o^{CF} = 2.0$. The black line and shaded area represent the mean and standard deviation across simulations ($n = 20$). Red bars indicate the position of mitotic domains. The green bar indicates the position of the cephalic furrow. EL, egg length. **h**, Simulations testing the effect of cephalic furrow position on ectopic folding. Representative samples with the cephalic furrow more anterior (top row), central (middle row) and posterior (bottom row) along the anteroposterior axis. The black line and shaded area represent the mean and standard deviation across simulations ($n = 20$). Red bars represent the position of mitotic domains. The green bar indicates the position of the cephalic furrow.

the bending energy reaches a peak, the fold continues to deepen more gradually but the number of folds rarely changes afterwards (Fig. 3d and Extended Data Fig. 4b). As folding events are stochastic and occur at distinct iterations in each simulation, we used the peak of bending energy as a reference point to standardize the comparison between simulations.

## Ectopic folding in mutant conditions

To obtain realistic values of the dimensionless bending rigidity $K_b^*$, we performed a sweep across the parameter space in mutant conditions. We found that the number of ectopic folds is higher in softer conditions (lower bending rigidity; Fig. 3e,f and Extended Data Fig. 4c,f).

In simulations without mitotic domains, we observed no folding events without the germ band (Fig. 3e, column i). With the germ band extended, the probability of buckling increases (Extended Data Fig. 4d) and the time to folding decreases (Extended Data Fig. 4e). The parameter sweep shows a clear transition in the phase space, with the buckling probability reaching a plateau around $K_b^* \approx 1.2 \times 10^{-4}$ (Fig. 3e). In these stiffer conditions, the germ band, even at its maximum extension, cannot drive the formation of ectopic folds.

Adding mitotic domains to the simulations changed the phase diagram, increasing the probability of folding. We found that, at lower bending rigidity values ($K_b^* \le 1.0 \times 10^{-4}$), mitotic domains alone can induce ectopic folding (Fig. 3f, column iii). For lower values of germ band extension, the number of folds is higher (Extended Data Fig. 4g) and the time to folding is lower (Extended Data Fig. 4h) than the simulations without mitotic domains. These simulations reveal that the germ band or mitotic domains alone can only drive ectopic folding in softer conditions, but that their combined action increases the mechanical instability of the tissues.

To determine where the real embryo lies in this parameter space, we identified the bending rigidity value in the simulations that recapitulates the main insight from our experimental data, that neither mitotic domains nor germ band alone can promote ectopic folding in vivo (Fig. 2). In the simulations, this corresponds to the bending rigidity where the average number of folds falls below 1 in mitotic domain-only and germ band-only conditions (Extended Data Fig. 4c,f). The criterion is fulfilled when $K_b^* \approx 1.0 \times 10^{-4}$. To compare this value to direct measurements, we calculated $K_b^*$ for the bending rigidity estimates in 3D-cultured epithelial monolayers[17] (Supplementary Note 3). After adjusting for differences in epithelial thickness, we calculated a $K_b^* = 2.05 \times 10^{-4}$ for the cultured monolayers. This is around a factor of 2 of the bending rigidity for the embryonic blastoderm that we estimated from our experimental data, suggesting that our reference value is consistent with existing measurements in other tissues. Thus, having established and validated this important bridge between experiments and simulations, we used this biologically relevant reference bending rigidity for subsequent simulations.

### Role of the cephalic furrow

The finding that ectopic folds are less frequent and smaller when the cephalic furrow is present (Tables 2–4, Fig. 1 and Extended Data Fig. 2) suggests that the invagination may counteract the mechanical instability generated by mitotic domains and germ band extension, potentially absorbing the compressive stresses at the head–trunk boundary. To explore this hypothesis of the cephalic furrow as a mechanical buffer, we analysed how the presence of the invagination impacts the dynamics of ectopic folding in simulations.

We programmed the cephalic furrow in our model by setting an intrinsic negative curvature ($\kappa_o^{CF}$) to a narrow region of the particle–spring blastoderm that matches the span of the initiator cells in vivo (Fig. 3a,b). Using our reference bending rigidity value of $K_b^* = 1.0 \times 10^{-4}$, we ran a parameter sweep for different $\kappa_o^{CF}$, and established a baseline where the invagination forms in a robust manner, with minimal variability, and phenocopies the cephalic furrow in vivo ($\kappa_o^{CF} > 0.3$; Extended Data Fig. 4i).

We first evaluated how the $\kappa_o^{CF}$ strength impacts ectopic folding. In conditions without the germ band ($g = 0$), the formation of the cephalic furrow reduces the spread and frequency of ectopic folding at the head–trunk boundary (Fig. 3g and Extended Data Fig. 4l). This reduction correlates with higher $\kappa_o^{CF}$ strength (Extended Data Fig. 4j). In conditions in which the germ band is extended, ectopic folding increases at the posterior region and can inhibit cephalic furrow formation when $\kappa_o^{CF} \le 1$ and $g \ge 0.2$ (Extended Data Fig. 4j). Therefore, although higher pull strengths are more effective at preventing ectopic folding, conflicting mechanical forces can diminish the buffering effect of the cephalic furrow.

In the simulations above, cephalic furrow and mitotic domains initiate at the same iteration. However, in wild-type embryos, the cephalic furrow forms 15 min before mitotic domains (Fig. 1a). To match the conditions in vivo and to test whether their relative timing of formation impacts ectopic folding, we delayed the formation of mitotic domains relative to the cephalic furrow in the simulations. In this condition, the cephalic furrow is very effective in preventing ectopic folding even for lower $\kappa_o^{CF}$ values (Fig. 3g and Extended Data Fig. 4j,k,l). Ectopic folding occurs only at the posterior region with the extended germ band (Extended Data Fig. 4k), similar to our observations in wild-type embryos in vivo (Table 4 and Extended Data Fig. 2i–m). These results reveal that relative timing, rather than pull strength, is more important at preventing ectopic folding.

Finally, we tested how the position of the cephalic furrow impacts its ability to prevent ectopic folding. In simulations without the germ band ($g = 0.0$), positioning the cephalic furrow more anteriorly (more than 70% of embryo length) still prevents ectopic folding in the head effectively compared with the wild-type position (65% of embryo length), but when placing it more posteriorly (less than 50% embryo length), the ectopic folding around mitotic domains becomes more frequent (Fig. 3h). Conversely, with the germ band extended ($g = 0.4$), positioning the cephalic furrow more posteriorly (less than 30% embryo length) prevents ectopic folding in the posterior region, whereas placing it more anteriorly (more than 50% embryo length) fails to do so (Fig. 3h). These simulations show that the cephalic furrow is the most effective at preventing ectopic folding when positioned between 40% and 60% of the length of the embryo, depending on the percentage of germ band extension.

Our physical model provides a theoretical basis that an epithelial fold such as the cephalic furrow—when forming before other morphogenetic movements around the middle of the anteroposterior axis—can absorb compressive stresses and prevent, to a substantial degree, mechanical instabilities in embryonic tissues during gastrulation.

## Evolution of gene expression

As described above, our analyses suggest that the effectiveness of the cephalic furrow in preventing epithelial instabilities depends on the position and time of the invagination. In *Drosophila*, this spatiotemporal control is determined genetically by the combinatorial expression of *btd*, *eve* and *prd* at the head–trunk boundary[8,15]. However, the specific genetic traits associated with the evolution of the cephalic furrow patterning cascade remain unclear. To address this question, we identified additional cephalic furrow genes and analysed their expression in dipteran species with and without the cephalic furrow.

### Patterning of the cephalic furrow

To identify cephalic furrow genes, we performed a live-imaging screen (Supplementary Note 4) and found that the *sloppy paired* (*slp*) transcription factors have a role in the positioning of initiator cells in *Drosophila* (Extended Data Fig. 5, Supplementary Video 17 and Supplementary Note 5). In wild-type embryos, the expression domains of *slp1* and *eve* demarcate the head–trunk boundary from the onset of zygotic activation until gastrulation (Fig. 4a and Extended Data Fig. 6a). Although early *slp1* transcripts are limited to the anterior end, early *eve* transcripts are initially ubiquitous[18] but begin to clear from the anterior end at nuclear cycle 11 (Extended Data Fig. 6a). At nuclear cycle 13, the two genes form broad, complementary territories that correspond to the head and trunk regions of the embryo, with the domains juxtaposed around 70% of the embryo length (Extended Data Fig. 6a). We first detected *btd* and *prd* transcripts at this interface (Extended Data Fig. 6a,b). During subsequent stages, the *slp1*–*eve* boundary progressively resolves into narrow abutting stripes that give rise to the row of initiator cells (Fig. 4a and Extended Data Fig. 6a). Together, the data suggest that *slp1* activity contributes to

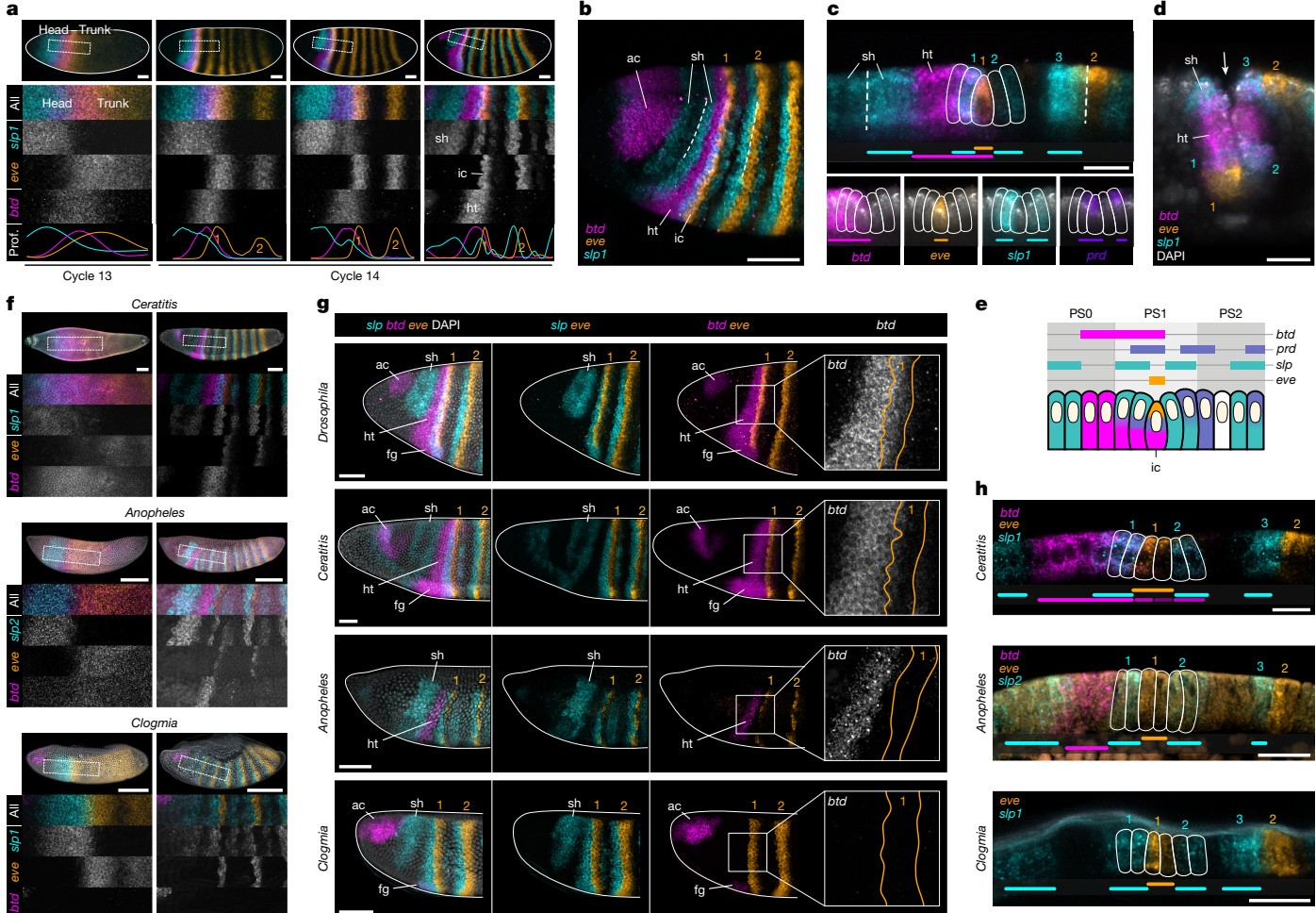

**Fig. 4 | Genetic patterning of the head–trunk boundary in *Drosophila*, *Ceratitis*, *Anopheles* and *Clogmia*. a**, Expression of *btd*, *eve* and *slp1* before gastrulation in *Drosophila*. Early *slp1* and *eve* transcripts demarcate the head–trunk boundary and resolve to sharp stripes with *btd* transcripts at the interface. The numbers 1 and 2 in orange indicate *eve* stripe 1 and 2, respectively. Scale bars, 50 μm. ht, *btd* head–trunk domain; ic, *eve* stripe 1 with initiator cells; sh, *slp* head domain. **b**, Expression patterns at the onset of gastrulation in *Drosophila* (lateral view). *slp1* stripes demarcate the outer edges of the cephalic furrow (dashed lines). Scale bar, 50 μm. ac, *btd* acron domain. **c**, Expression patterns at the onset of gastrulation in *Drosophila* (profile view). *eve*-expressing initiator cells also express *btd* and are abutted by *slp1* stripes. *prd* is offset from *slp1* by the one-cell row. The dashed lines demarcate the outer edges of the cephalic furrow. Scale bar, 20 μm. **d**, Expression patterns of the invaginated cephalic furrow in *Drosophila*. Scale bar, 20 μm. **e**, Schematic of the combinatorial expression at the head–trunk boundary of *Drosophila*. **f**, Expression of *btd*, *eve* and *slp* during nuclear cycles 13 (left) and 14 (right) in *Ceratitis*, *Anopheles* and *Clogmia* embryos. Scale bars, 100 μm. **g**, *btd*–*eve* overlap at the head–trunk boundary of different dipterans; it is present in species with a cephalic furrow (*Drosophila* and *Ceratitis*) and absent in species without (*Anopheles* and *Clogmia*). Scale bars, 50 μm. fg, *btd* foregut domain. **h**, Expression patterns at the onset of gastrulation in *Ceratitis*, *Anopheles* and *Clogmia* (profile view). Scale bars, 20 μm.

restricting the anterior boundary of *eve* expression during early stages of zygotic activation, determining the position of initiator cells along the anteroposterior axis and, consequently, the site of invagination of the cephalic furrow.

At the onset of gastrulation, the expression of *btd*, *eve*, *slp* and *prd* at the head–trunk boundary of *Drosophila* forms a unique combinatorial code that coincides with the different portions of the cephalic furrow (Fig. 4b and Extended Data Fig. 6c). The central row of *eve*-expressing initiator cells are surrounded by *slp1*-expressing adjacent cells, with *prd* expression offset by a single row of cells, relative to the inner *slp1* stripes (Fig. 4b–d and Extended Data Fig. 6b,c). Moreover, *slp1*-expressing cells also demarcate the outer edges of the invagination (Fig. 4d). This molecular arrangement is disrupted in mutants that exhibit cephalic furrow defects (*btd*, *eve* and *prd*; Extended Data Fig. 7 and Supplementary Note 6). This combinatorial expression suggests that each row has a unique transcriptional identity, and its disruption in mutants indicates that this specific molecular profile may be important for the patterning of the invagination in *Drosophila* (Fig. 4e).

## Innovation at the head–trunk boundary

To uncover the differences in the head–trunk genetic patterning associated with the evolution of the cephalic furrow, we compared the expression patterns in *Drosophila* with three other dipteran species: one from a family known to have a cephalic furrow, the Mediterranean fruit fly *Ceratitis capitata* (Tephritidae)[19,20], and the other two belonging to families in which the cephalic furrow has not been observed[14], the malaria mosquito *Anopheles stephensi* (Culicidae)[21,22] and the drain fly *Clogmia albipunctata* (Psychodidae)[14,23].

The three species show early, juxtaposing domains of *slp* and *eve* demarcating the head and trunk regions in a pattern highly similar to that of *Drosophila* (Fig. 4a,f and Extended Data Fig. 8a–c). Moreover, the late pattern of abutting *slp* and *eve* stripes is nearly identical between the four species (Fig. 4a,f,g), and *prd* expression differs only in *Clogmia*, where *prd*-expressing cells are not offset from *slp* and *eve* stripes (Extended Data Fig. 8d,e). The main difference that we observed between species with and without the cephalic furrow is the expression

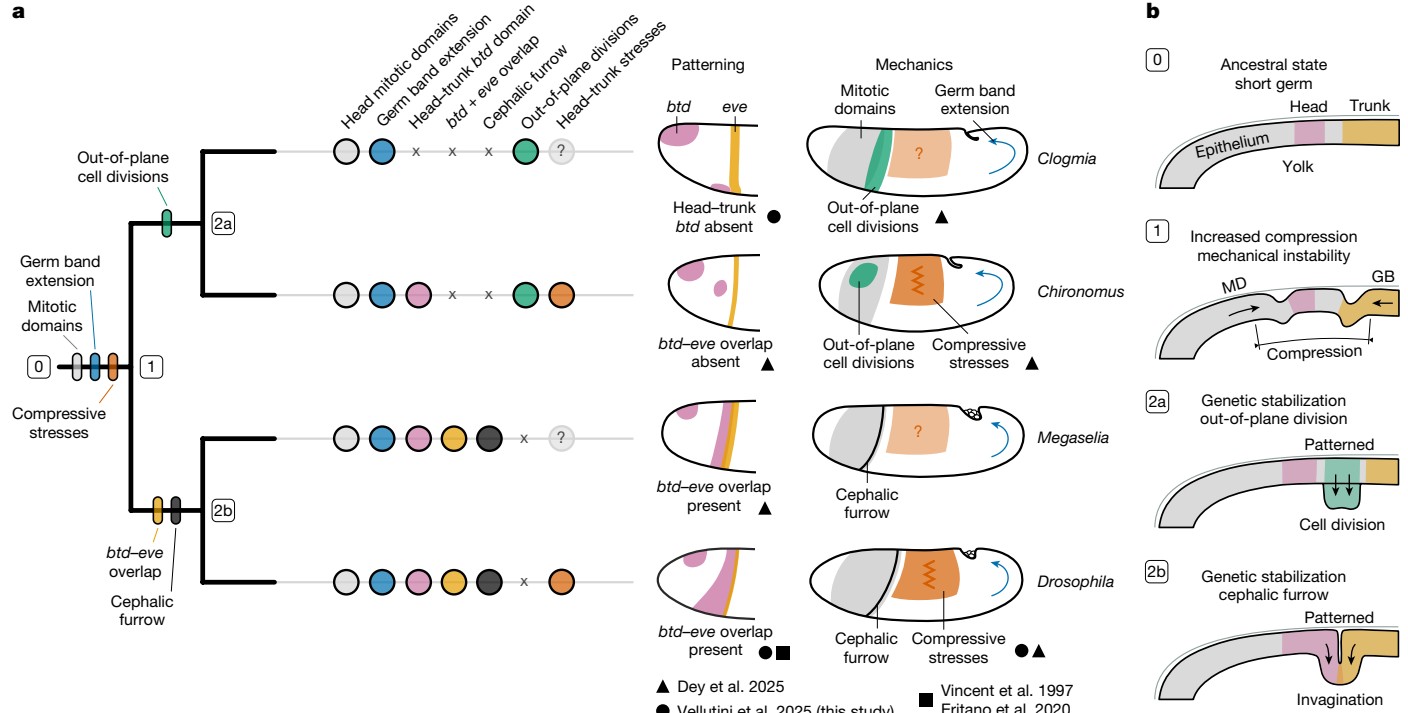

**Fig. 5 | Interplay between genetics and mechanics during cephalic furrow evolution. a**, Cephalic furrow traits mapped onto a simplified dipteran phylogeny (based on ref. 34). Germ band extension and mitotic domains are ancestral, suggesting that compressive stresses at the head–trunk boundary were present since the dawn of Diptera. The cephalic furrow is present in the common ancestor of *Megaselia* and *Drosophila* (cyclorrhaphan flies) and correlates with the presence of a *btd*–*eve* overlap at the head–trunk boundary. Out-of-plane divisions are present at the head–trunk boundary of the non-cyclorrhaphan flies *Clogmia* and *Chironomus*. Data sources are annotated with geometrical symbols: this study (black circle), Dey, Kaul, Kale et al.[14] (black triangle), and Eritano et al.[9] and Vincent et al.[8] (black square). **b**, Evolutionary scenario for the origin of morphogenetic innovations in Diptera. In the ancestral state (short germ), there was no mechanical instability at the head–trunk boundary during gastrulation (0). The appearance and concurrent formation of mitotic domains and germ band extension increased the compressive stresses and ectopic folding around the head–trunk boundary (1). As epithelial instability could be detrimental to developmental robustness and individual fitness, morphogenetic processes mitigating these effects may have been favoured by natural selection. In this light, the out-of-plane divisions (2a) and the cephalic furrow (2b) might have evolved in response to mechanical instability, representing independent solutions to a common challenge. For the cephalic furrow, this might have occurred through the genetic stabilization of ectopic folds into a patterned embryonic invagination.

of *btd*. Although the head–trunk domain of *btd* overlaps with *eve* stripe 1 in *Drosophila* and *Ceratitis*, it does not overlap with *eve* in *Anopheles* and is entirely absent in *Clogmia* (Fig. 4g,h). The *btd*–*eve* overlap is also absent in *Chironomus*, another dipteran that lacks a cephalic furrow[14]. These results suggest that cephalic furrow evolution may have been associated with spatial changes in *btd* expression and the emergence of a *btd*–*eve* overlap at the head–trunk boundary.

## Discussion

Our work investigates the developmental role and evolution of the cephalic furrow. We found that tissues at the head–trunk boundary of *Drosophila* are under increased compressive stress due to the concomitant formation of mitotic domains and germ band extension. Without the cephalic furrow, these stresses build up, and the outwards forces exerted by cell divisions in a compressed epithelial monolayer cause mechanical instability and tissue buckling[24,25] (Supplementary Note 7). Our results provide evidence that the formation of the cephalic furrow counteracts these compressive stresses and prevents epithelial instabilities at the head–trunk boundary. Therefore, we propose that the cephalic furrow has a mechanical role during *Drosophila* gastrulation.

This physical role is intriguing and raises the idea that the cephalic furrow may have evolved in response to the mechanical challenges of dipteran gastrulation with mechanical instability acting as a selective pressure. For this to be the case, we expect mechanical instability to be detrimental to embryogenesis and the fitness of individuals.

Although mechanical compression can trigger ATP release[26], calcium signalling[27] and DNA damage[28], and ectopic folding can potentially disrupt short-range signalling and cell-to-cell interactions, investigating these effects in vivo is challenging. There is evidence, however, that inhibiting the cephalic furrow via optogenetics increases the frequency of distorted ventral midlines[14], suggesting that mechanical instability may affect the robustness of developmental processes.

The distribution of cephalic furrow traits onto the dipteran phylogeny is consistent with the hypothesis of mechanical instability as a selective pressure (Fig. 5 and Extended Data Fig. 9). Mitotic domains and germ band extension—the sources of stress—are ancestral and evolved before the cephalic furrow, a derived trait and evolutionary novelty of cyclorrhaphan flies[14] (Fig. 5a). Species without a cephalic furrow show out-of-plane cell divisions at the head–trunk boundary[14], an alternative strategy for mitigating compressive stresses (Fig. 5b). As the establishment of a *btd*–*eve* overlap is associated with the presence of the cephalic furrow[14] (Figs. 4 and 5a), differences in the genetic patterning of the dipteran head–trunk boundary might have contributed to the origin of distinct morphogenetic solutions in response to a similar mechanical selective pressure (Fig. 5b).

Classical theoretical works have raised the hypothesis that physical processes were essential drivers of morphological innovation before the emergence of genetic programs[29–31]. The findings described here and in ref. 14 provide supporting empirical evidence that mechanical forces have a role in the origin of morphogenetic innovations and

that the genetic stabilization of mechanical conflicts may be a more broadly occurring mechanism generating morphogenetic diversity in embryonic development.

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

## Methods

### *Drosophila* stocks and genetic crossings

To generate fluorescent cephalic furrow mutants, we performed genetic crosses using the loss-of-function alleles *btd*[XA] (FBal0030657), *eve*[3] (FBal0003885), *prd*[4] (FBal0013967), *slp*[Δ34B] (FBal0035631) and *stg*[2] (FBal0247234); the membrane fluorescent marker *Gap43-mCherry* (FBal0258719; a gift from K. Skouloudaki); and the green fluorescent balancers *FM7c, Kr-GFP* (FBst0005193), *CyO, twi-GFP* (gift from A. Jain) and *TM3, Kr-GFP* (FBst0005195). We established stable lines balancing the loss-of-function alleles with fluorescent balancers and used the lack of GFP signal to identify homozygous embryos in our live-imaging recordings. For genes on chromosomes 1 and 2 (*btd*, *eve* and *prd*), we added the membrane marker on chromosome 3 ((*btd*[XA]/*FM7c, Kr-GFP;; Gap43-mCherry/MKRS*), (*eve*[3]/*CyO, twi-GFP; Gap43-mCherry/MKRS*) and (*slp*[Δ34B]/*CyO, twi-GFP; Gap43-mCherry/TM6B*)). For *stg*, which is located on chromosome 3, we recombined the allele with Gap (*Gap43-mCherry, stg*[2]/*TM3, Kr-GFP*). As the *btd–stg* double-mutant stable line is weak, we imaged the progeny of *btd*[XA]/*FM7c, Kr-GFP;; Gap43-mCherry, stg*[2]/*Gap43-mCherry* flies, identifying *btd* homozygosity by the GFP and *stg* homozygosity by the lack of cell divisions after gastrulation. For laser ablations, we used a *moe-GFP* line (a gift from E. Knust). The wild-type stocks contain the Gap43–mCherry marker in the Oregon-R genetic background. We obtained the founder fly stocks from the Bloomington Drosophila Stock Center and the Kyoto Stock Center and deposited the lines in the MPI-CBG stock collection. The complete list of FlyBase[35] accession numbers and genotypes is available in the data repository for the project[36].

### Animal husbandry and embryo collection

We maintained the *Drosophila* stocks in 50-ml hard plastic vials containing standard fly food and enclosed with a foam lid to allow air exchange. They were kept in an incubator with a constant 25 °C temperature and 65% humidity and a 12–12 h light cycle. For imaging, we first amplified the stocks in larger 200-ml vials for a few weeks. We then narcotized the flies with $CO_2$ and transferred them to a cage with a plate attached to one end containing a layer of apple juice agar and a slab of yeast paste on top. The flies were left to acclimatize in the cage for 2 days before the experiments. To guarantee that the embryos are at a similar developmental stage, we exchanged the agar plate once per hour at least twice (pre-lays) and let the flies lay the eggs on the agar for 1 h before collecting the plate. After filling the plate with water, we used a brush to release the eggs from the agar and transferred them to a cell strainer with 100-μm nylon mesh (VWR). To remove the chorion, we immersed the embryos in 20% bleach (sodium hypochlorite solution; 1.05614.2500, Merck) for 90 s, washed abundantly with water and proceeded to mounting for live imaging.

We maintained *Clogmia* flies in 9-cm-wide plastic Petri dishes with a layer of wet cotton at room temperature and fed them weekly with powdered parsley. To obtain embryos for fixation, we collected the adult flies in a 200-ml hard plastic vial with wet cotton and let them mate for 2–3 days. Then, we anaesthetized the flies with $CO_2$, dissected the ovaries from ripe females and released the eggs using tweezers in deionized water, which activates embryonic development[37,38]. We let embryos develop in deionized water at room temperature until the desired stage. To remove the chorion, we transferred the embryos to a glass vial with 0.5× PBS using a fine brush, exchanged the medium for 5% bleach in 0.5× PBS for 2 min and washed abundantly with 0.5× PBS. Using the diluted PBS solution instead of water prevents the embryos from bursting after dechorionation.

We obtained pupae of the EgyptII wild-type strain of *Ceratitis* from the Insect Pest Control Laboratory of the International Atomic Energy Agency. Adult flies were kept at 25 °C, 65% humidity and 12–12 h light cycle, inside 49 × 30 × 30 cm plexiglass cages with the front and back ends covered by a nylon mesh. We provided water through a soaked towel and food as a 3:1 sugar:yeast mixture. As ripe females laid eggs through the nylon mesh, we placed a plastic container with water at the back end of the cage for several hours to collect eggs. We dechorionated *Ceratitis* embryos using the *Drosophila* protocol.

We performed the collection of *Anopheles* embryos at the Center for Integrative Infectious Diseases Research at Heidelberg University. To collect embryos, we placed a glass container with water and filter paper inside a cage with 300 mated females, which were fed a blood meal 72 h before, and put the cage in the dark for 2 h at 29 °C. We then removed the container from the cage and let the embryos develop until the desired stage. To dechorionate, we collected the embryos on a cell strainer, incubated them in 5% bleach for 75 s and washed them thoroughly with deionized water.

### Embryo fixation and in situ hybridization

For *Drosophila* and *Ceratitis*, we transferred dechorionated embryos to a glass vial containing equal volumes of 4% formaldehyde in PBS and *n*-heptane and let the vial shake at 215 rpm for 45 min. For *Clogmia*, we diluted the fixative in 0.5× PBS. After removing the fixative (lower phase) using a glass pipette, we added an equal volume of 100% methanol and shook the vial vigorously by hand for 1 min. We then removed the *n*-heptane (upper phase) and collected the embryos on the bottom of an Eppendorf tube and washed several times with 100% methanol. For *Anopheles*, we followed a similar protocol that includes a longer 30-min wash in water after fixation, a 30-s boiling water step followed by 15 min in ice-cold water, until the final methanol washes[21]. All the samples were stored in 100% methanol at −20 °C.

We performed the in situ hybridization of *btd*, *eve*, *prd* and *slp* genes using the Hybridization Chain Reaction (v3.0; HCR)[39] reagents, except for the probe sets, which we designed using a custom script. The oligos were obtained from Sigma-Aldrich. We selected the HCR amplifiers to allow for triple (multiplexed) in situ combinations of *btd* + *eve* + *slp* or *prd* + *eve* + *slp*. Before starting, we manually devitellinized *Anopheles* embryos using fine tweezers. Then, we rehydrated the embryos in 100% methanol with a series of washes to 100% PBT. We permeabilized the embryos with 1:5,000 dilution of proteinase K (20 mg ml⁻¹) for 5 min, except for *Drosophila*. All samples were re-fixed in 4% formaldehyde for 40 min and washed thoroughly with PBT. We then followed the 'In situ HCR v3.0 protocol for whole-mount fruit fly embryos revision 4 (2019-02-21)' from Molecular Instruments. After the protocol, we stained the embryos with 1:1,000 DAPI in 5× SSCT solution for 2 h and mounted the embryos in 80% glycerol in 5× SSCT for imaging.

### Sample mounting for microscopy

For most of our live imaging, we used a Zeiss Lightsheet Z.1 microscope running ZEN 2014 SP1 (v9.2.10.54). To increase the throughput of samples imaged in one session, we optimized a mounting strategy previously developed in our laboratory[40]. First, we cut a 22 × 22 mm glass coverslip (0.17 mm thickness) into 6 × 15 mm strips using a diamond knife and attached a single strip to a custom sample holder using silicon glue, letting it harden for 15 min. We then coated the coverslip strip with a thin layer of heptane glue and let it dry while preparing the embryos. Using a fine brush, we transferred the embryos collected in the cell strainer onto an agar pad and oriented them manually with a blunt cactus spine under a stereomicroscope. We aligned about 20 embryos in a single line (head to tail) along the main axis of the strip with the left or ventral sides up, depending on the experiment. To attach the embryos to the coverslip, we carefully lowered the sample holder over the agar pad until the glass coated with heptane glue touched the embryos. We placed the sample holder into the microscope chamber filled with water and rotated it so that the samples were facing the detection objective directly and the coverslip was orthogonal to the detection objective; this is important to prevent the light sheet from hitting the glass edges. With the embryos oriented vertically along the coverslip, the light sheet generated from the illumination objectives coming from

the sides only needed to pass through the width of the embryo (about 200 μm). This approach gives the best results for recording lateral and dorsal views and is ideal for live-imaging homozygote embryos as they are only about one-fourth of the total number of imaged embryos. For imaging fixed in situ samples, we used an inverted Zeiss LSM 700 confocal microscope running ZEN 2012 SP5 FP3 (v14.0.25.201). We mounted the samples immersed in 80% glycerol between a slide and a glass coverslip supported by tape.

## Microscopy acquisition parameters

For the light-sheet lateral datasets, we used a Zeiss ×20/1 NA Plan-Apochromat water immersion objective to acquire stacks with 0.28-μm $xy$ resolution and 3-μm $z$ resolution covering half of the volume of the embryo in a single view. This $z$ resolution was restored to 1 μm during image processing (see below). For the dorsal datasets, we used a Zeiss ×40/1 NA Plan-Apochromat water immersion objective to acquire stacks with 0.14-μm $xy$ resolution and 3-μm $z$ resolution covering a volume around in the middle section of the anterior end of the embryo. We adjusted the time resolution between 45 s and 60 s per frame to maximize the number of embryos acquired in one session. To visualize both the membrane signal (mCherry) and the green balancer signal (GFP), we acquired two channels simultaneously using the 488-nm and 561-nm lasers at 3% power with an image splitter cube containing a LP560 dichromatic mirror with SP550 and LP585 emission filters. All live-imaging recordings were performed at 25 °C. For the confocal datasets, we used a ×20/0.8 Plan-Apochromat Zeiss air objective to acquire four-channels using three tracks (405, 488 and 639, and 555 nm, respectively) with a BP575-640 emission filter and about 0.4-μm $xy$ resolution and 2-μm $z$ resolution covering about half the volume of the embryo.

## Image processing and visualization

We converted the raw light-sheet imaging datasets into individual TIFF stacks for downstream processing using a custom macro (Process-Z1Coverslip.ijm) in Fiji/ImageJ (v2.16.0/1.54p) with Java (v1.8.0_172)[41,42]. To visualize the presence and dynamics of ectopic folds, we generated 3D renderings of the surface of embryos in lateral recordings using a custom animation (3D_animation.txt) in the Fiji plugin 3Dscript (v0.2.1)[43]. For analysing the entire epithelial surface, we first improved the signal-to-noise ratio and $z$ resolution of lateral datasets from 3 μm to 1 μm by training a deep learning upsampling model using CARE CSBDeep (v0.3.0)[44]. Then, we created cartographic projections of the lateral recordings using the ImSAnE toolbox (v3a7be24)[45] by loading the restored data in MATLAB (R2015b)[46], segmenting the epithelial surface using ilastik (v1.3.3b2)[47], and generating 3D cartographic projections of lateral views following a workflow established for fly embryos[48]. As the pixel size varies across the projection, the provided scale bars represent approximate values at the central portion of the image. To visualize in situ hybridization data, we performed maximum intensity projections or extracted single slices from the raw volumes. For all microscopy images, we only performed minimal linear intensity adjustments to improve their contrast and brightness[49]. The imaging data for the light-sheet and in situ hybridization experiments analysed in this study are available on Zenodo[50].

## Ectopic fold analyses

To characterize the relative timing of ectopic folding, we annotated the position of the germ band and the number of frames after the onset of gastrulation at the initial buckling, when the first cells disappear from the surface in the lateral 3D renderings. We defined the onset of gastrulation ($T = 0$) as the moment immediately after the end of cellularization and immediately before the beginning of the ventral furrow invagination. To visualize the variability of ectopic folding, we manually traced the fold outlines in lateral recordings using Fiji. Because embryos have different sizes, we first used the plugin bUnwarpJ

(v2.6.13)[51] (https://imagej.net/plugins/bunwarpj) to register individual frames and then applied the same transformation to the fold traces for a standardized comparison. We analysed the dynamics of ectopic folds by measuring the relative angle and tortuosity of the segmented line traces over time and to visualize the kinetics, we generated colour-coded temporal projections using the script Temporal Color Code (v101122; https://imagej.net/plugins/temporal-color-code) with the perceptually uniform mpl-viridis colour map (https://bids.github.io/colormap) bundled in Fiji.

To estimate the folded area in the cephalic furrow and ectopic folds, we annotated the region of the blastoderm before gastrulation that infolded in the cartographic projections using Fiji and calculated the area, correcting the pixel dimensions according to the coordinates in the projection. For the fold depth, we measured the distance between the vitelline envelope to the tip of the fold at the moment of maximum depth in the dorsal recordings. For the analysis of the epithelial surface, we used the plugin MorphoLibJ (v1.6.0)[52] (https://imagej.net/plugins/morpholibj) to segment, measure and colour-code the cell apical areas, and the plugin Linear Stack Alignment with SIFT (v1.5.0)[53] (https://imagej.net/plugins/linear-stack-alignment-with-sift) to register cells between timepoints.

## Laser cauterization experiments

We performed laser cauterization experiments in two microscope setups, a light-sheet Luxendo MuVi SPIM with a photomanipulation module and a confocal Zeiss LSM 780 NLO with multiphoton excitation running ZEN Black (v14.024.201). For the MuVi SPIM, we embedded dechorionated embryos in 2% low-melting agarose and mounted the samples in glass capillaries to obtain in toto recordings. We used a pulsed infrared laser at 1,030–1,040 nm with a 200-fs pulse duration and 1.5 W power to cauterize the posterior region of the dorsal embryonic surface, attaching the blastoderm to the vitelline envelope. Using an Olympus ×20/1.0 NA water immersion objective, we acquired stacks with 0.29-μm $xy$ resolution and 1-μm $z$ resolution of four different angles every 1 min. For the Zeiss microscope, we attached the embryos with the dorsal side down onto coverslips using heptane glue and immersed them in halocarbon oil. We cauterized the embryos sequentially using a near-infrared 800-nm laser (Chameleon Vision II) through a single pixel line (210 nm per pixel and 100 μs per pixel) around the same dorsal region to block the germ band extension. We used a Zeiss ×25/0.8 NA LD LCI Plan-Apochromat glycerol immersion objective to acquire every 2:38 min two different planes of the blastoderm: (1) the surface to monitor the germ band extension, and (2) 40 μm deep in the equatorial region to monitor the occurrence of ectopic folding. The stacks had 0.21-μm $xy$ resolution and 1-min time resolution. To obtain a quantitative measure of ectopic folding, we analysed the degree to which the tissues deform between non-cauterized and cauterized mutants, using as a proxy the tortuosity of the epithelium outline. For that, we took the profile slices from dorsal recordings and transformed the curved vitelline envelope into a straight line using the Straighten tool of ImageJ (Supplementary Fig. 4a). We then cropped a 200 × 25 μm region along the head–trunk interface and applied Gaussian blur, thresholding and edge detection to obtain the epithelium outline for individual timepoints covering about 50 min after gastrulation (Supplementary Fig. 4a,b). We extracted measurements from the epithelium outlines using the ImageJ plugin Analyze Skeleton (v3.4.2)[54] (https://imagej.net/plugins/analyze-skeleton) and generated the colour-coded temporal projections as described above. The imaging data for the laser cauterization experiments are available on Zenodo[55].

## Laser ablation experiments

We performed laser ablations in a Yokogawa CSU-X1 spinning disk confocal with an EMCCD camera (Andor iXon DU-888) and the software AndorIQ for image acquisition. We attached dechorionated embryos

laterally to a MatTek glass-bottom Petri dish and covered the samples with water. Then, we performed the ablations using a Titanium Sapphire Chameleon Ultra II (Coherent) laser at 800 nm tuned down from 80 MHz to 20 kHz with a pulse-picker. The laser power measured before the microscope port was 6 mW, and the pixel dwell time for scanning was 2 μs. To ensure the cut, we repeated the scan ten consecutive times along a single cell, acquiring a single slice with a ×60/1.2 NA water immersion objective with 0.18-μm $xy$ resolution and 200-ms time steps. We ablated each embryo just once. The temperature was maintained at 28 °C. To analyse the ablation data, we created a line crossing the edges of the ablated cell perpendicular to the cut and generated a kymograph using the Fiji plugin Multi Kymograph (v3.0.1; Supplementary Fig. 5). We then binarized the kymographs, measured the distance between cell edges over the first 30 s after the cut and performed a linear fit of the data to obtain the recoil velocity (Supplementary Fig. 5). We performed additional laser ablations in an inverted Zeiss Axio Observer. Z1 spinning disk confocal microscope running ZEN Blue (v3.2) with a Rapp OptoElectronic setup for photo-manipulation running SysCon2. The imaging data for the laser ablation experiments are available on Zenodo[55].

### Strain rate analysis

To estimate the strain rates, we first performed particle image velocimetry on cartographic projections using the ImageJ plugin iterativePIV (v2.0)[56] (https://sites.google.com/site/qingzongtseng/piv). Then, we used the equation

$$E = \left| \frac{1}{2}(\vec{\nabla} \cdot \vec{v}) + \frac{1}{2}(\partial_x v_y + \partial_y v_x) \right|$$

to define and calculate the magnitude of the strain rate, where $v$ is the displacement obtained in the particle image velocimetry analysis divided by the time in minutes. The measurements combine isotropic and anisotropic strain rates. We used these values to create a colour-coded overlay for the strain rate (Supplementary Fig. 2b). To generate the line plots, we averaged the strain rate along the dorsoventral axis in two predefined regions, the head–trunk (canonical cephalic furrow position) and the trunk–germ (posterior to the mitotic domain 6; Supplementary Fig. 2b).

### Model and simulations

Our model follows an approach similar to a previously published model of epithelial buckling under confinement[17]. It represents the monolayer epithelium of the early *Drosophila* embryo in a cross-section as a single line through the apicobasal midline of the epithelial cells. The tissue is modelled as an elastic rod with a stretching energy per unit length $W_s$ and bending energy per unit length $W_b$ so that the total energy per unit length is $W_T = W_s + W_b$. In full,

$$W_T = \int_{L_o} \frac{1}{2} K_s \left( \frac{ds}{ds_o} - 1 \right)^2 ds_o + \int_{L_o} \frac{1}{2} K_b (\kappa - \kappa_o)^2 ds_o$$

where $K_s$ is the stretching rigidity and $K_b$ is the bending rigidity of the tissue; $ds_o$ and $ds$ are the preferred and current lengths of the curve, respectively; and $\kappa$ is the curvature of the rod. $L_o$ is the total length of the tissue in a stress-free condition. To perform numerics, we discretize the curve into $N$ particles indexed by $i$. The total energy per unit length for this discretized model is given by

$$W_T^* = \frac{1}{2} K_s \sum_{i=2}^{N-3} \left( \frac{\Delta r_i}{\Delta r_o} - 1 \right)^2 \Delta r_o + \frac{1}{2} K_b \sum_{i=2}^{N-3} (\kappa_i - \kappa_{o,i})^2 \Delta r_o$$

where $\Delta r_o$ is the preferred length of springs connecting consecutive points (equal for all springs); $\Delta r_i$ is the current length between $i$ and

$i + 1$; $\kappa_i$ is the discretized curvature at point $i$; and $\kappa_{o,i}$ is the preferred curvature at point $i$ (equal to 0, except when specified). The first and last two points of the curve are fixed in space. To obtain a physically meaningful dimensionless bending rigidity, we divided the bending rigidity by the factor $K_s L^2$ as

$$K_b^* = \frac{K_b}{K_s L^2}$$

where $L$ is the semi-major axis of the embryo. To minimize the total energy, we added a ground level of noise to the particles and let the particles move in the direction of the forces. The motion of the particles is governed by

$$\frac{\Delta \vec{r_i}}{\Delta t} = -\frac{L}{K_s \tau} \frac{\partial W^*}{\partial \vec{r_i}} + \vec{\zeta_i}$$

where $\vec{r_i}$ is the current position of the $i$th particle; $\tau$ represents an arbitrary timescale introduced here to balance dimensions (set to 1); $\Delta t$ are the timesteps (set to $10^{-5} \times \tau K_s/L$); and $\vec{\zeta_i}$ is the noise chosen from a Gaussian distribution with mean 0 and standard distribution $10^{-5} \times L$.

In our model, the position of the germ band corresponds to the position of the last particle in the curve on the semi-ellipse that represents the embryonic blastoderm. The extent of the germ band is given by $g$, which is the projection of the germ band arc length onto the mid-axis of the embryo normalized by the embryo length ($2L$). When $g = 0$, the tissue is free of stretching stress, but at any other $0 < g < 1$, the blastoderm will be compressed. The preferred lengths of the individual springs are obtained by dividing the elliptical arc length into $N$ equal segments. The length of each segment is given by $\Delta r_o = \frac{1}{N} \left( L \int_0^\pi \sqrt{1 - e^2 \cos^2(u)} \, du \right)$. To find the initial lengths of the springs, we used

$$\Delta r(t = 0) = \frac{1}{N} \left( L \int_{u'}^{\pi} \sqrt{1 - e^2 \cos^2(u)} \, du \right)$$

where $e = \sqrt{1 - (0.4)^2}$ and the angle $u'$ corresponds to the position of the blastoderm end. $u'$ is obtained for a given value of $g$ by $u' = \cos^{-1}(1 - 2g)$. Here we obtained the initial lengths by dividing the compressed blastoderm into $N$ equal segments. For any simulation, the value of $g$ is constant (the blastoderm end is static in position). To model mitotic domains, we introduced new particles and springs on the midpoints between two particles in specific regions of length $0.5L$. The new springs were given the same $\Delta r_o$ as the rest of the springs in the tissue. The blastoderm is confined by a rigid boundary in the shape of a semi-ellipse. Any particle that lands outside this boundary at any timestep was repositioned onto the rigid boundary. This new position was prescribed by taking the intersection point of the rigid boundary curve and the line segment that connects the position before this iteration (which was inside or on the vitelline envelope) and the position outside the vitelline envelope. Finally, we defined and counted a fold when we found that the distance of a particle from the rigid boundary is greater than a threshold value. To calculate this threshold, we measured the maximum distance that particles can achieve when the tissue is in a stress-free state. This threshold was calculated to be $0.035L$. The code for the model and the simulation data are available in the theory repository on Zenodo[57].

### Data visualization and figure assembly

We created illustrations and assembled the final figure plates using Inkscape (v1.2.2)[58]. For microscopy videos, we exported the original stacks as AVI without compression at 10–15 fps using Fiji and post-processed them to MPEG-4 format 1,080p resolution using the H.264 encoding at a constant bitrate quality factor of 18 for visualization using HandBrake

(v1.6.1)[59]. The high-resolution figures and videos are available in a Zenodo repository[60]. We performed the data wrangling, statistical analyses and plotting in R (v4.2.1)[61] using R Markdown notebooks in RStudio (v2022.7.2.576)[62], and in Python (v3.10.7) using Jupyter notebooks (v6.5.4)[63]. The source files and analysis pipelines are available in the main repository on Zenodo[36].

## Statistics and reproducibility

The phenotypes that we report in this study were reproducible across multiple independent experiments. For the live imaging, we performed 7 experiments in *btd* mutants (total of 50 embryos), 5 experiments in *eve* mutants (total of 36 embryos) and 3 experiments in *prd* mutants (total of 41 embryos; Fig. 1b,c,f,g, Extended Data Fig. 3c, Extended Data Fig. 1a,b, Extended Data Fig. 2a,b,h,k,l and Fig. 2a–c,f, respectively). The phenotypes were also consistent across 6 experiments in *slp* mutants (total of 39 embryos; Extended Data Fig. 5a,g), 3 experiments in *stg* mutants (total of 46 embryos; Extended Data Fig. 3a,b), 6 experiments in *btd–eve* double mutants (total of 35 embryos; Extended Data Fig. 3d) and 2 experiments in wild-type embryos (total of 36 embryos; Fig. 1c, Extended Data Fig. 2k,l,m and Fig. 2a). For the germ band cauterization, we performed 6 experiments in *btd* mutants (total of 10 embryos; Extended Data Fig. 3g), 5 experiments in *eve* mutants (total of 10 embryos; Fig. 2i–k) and 8 experiments in wild-type embryos (total of 12 embryos; Extended Data Fig. 3e,f). For the gene expression, the wild-type patterns of *btd*, *eve* and *slp* were highly consistent across 3 experiments in *Drosophila* (total of 26 embryos; Fig. 4a–d,g and Extended Data Fig. 6a), 3 experiments in *Ceratitis* (total of 38 embryos), 4 experiments in *Anopheles* (total of 43 embryos) and 4 experiments in *Clogmia* (total of 44 embryos; Fig. 4f–h and Extended Data Fig. 8a–c). We also obtained consistent patterns of *prd* expression among 4 experiments in *Drosophila* (total of 10 embryos; Fig. 4c and Extended Data Fig. 6b,c) and 1 experiment in *Clogmia* (total of 20 embryos; Extended Data Fig. 8d,e). The expression patterns in mutant embryos were repeatable across 4 independent experiments in *slp* mutants (total of 30 embryos; Extended Data Fig. 5c–f,h), 5 experiments in *btd* mutants (total of 20 embryos), 2 experiments in *eve* mutants (total of 12 embryos) and 2 experiments in *prd* mutants (total of 12 embryos; Extended Data Fig. 7a–c).

We performed no previous estimation for sample size and no randomization or blinding strategy for experiments. Following previous studies in the field, we determined the number of experiments and sample size based on the repeatability of the observed phenotypes. Sample numbers refer to biological replicates. In all boxplots, the centre represents the median, the lower and upper hinges correspond to the first and third quartiles (25th and 75th percentiles) and the whiskers extend from the hinges until 1.5 times the interquartile range. The asterisks in plots indicate $P < 0.05$ in a two-sided Mann–Whitney $U$-test contrasting the condition against wild type; exceptions are described in figure legends. We report $P$ values rounded to three decimal places and show values lower than 0.001 as $P < 0.001$; the exact values with full decimal places for each contrast are available in the main repository on Zenodo[36].

## Reporting summary

Further information on research design is available in the Nature Portfolio Reporting Summary linked to this article.

## Data availability

All the data supporting the findings of this study have been deposited on Zenodo: the main repository containing the raw data, analyses pipelines and source files for figures and text (https://doi.org/10.5281/zenodo.7781947)[36]; the imaging data for the light-sheet and in situ hybridization experiments (https://doi.org/10.5281/zenodo.15876638)[50]; the imaging data for the laser perturbation experiments (https://doi.org/10.5281/zenodo.15876646)[55]; the theory repository containing the code and scripts of the model, the output data of simulations, and notebooks of analyses and plotting (https://doi.org/10.5281/zenodo.7784906)[57]; and the media repository containing the high-resolution figures and videos (https://doi.org/10.5281/zenodo.7781916)[60]. Source data are provided with this paper.

## Code availability

All the code necessary to reproduce the data processing and downstream analyses in this study is available as documented scripts and computational notebooks in the main repository (https://doi.org/10.5281/zenodo.7781947)[36] and in the theory repository (https://doi.org/10.5281/zenodo.7784906)[57] on Zenodo.

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

**Acknowledgements** We thank all current and former LoPaTs for discussions and support during this project; A. Jain and V. Ulman for the initial assistance with cartographic projections; G. Serafini for the help with fly crosses; A. Bailles for constructive feedback on the physics; M. Burkon and P. Mejstrik for technical support; J. Brugués and K. Ishihara for the laser ablation setup; the MPI-CBG's Light Microscopy Facility and Computer Services Facility for assistance with data acquisition and processing; S. Ssykor and C. Maas for support with fly stocks;

R. Žídek for essential help in construct design; S. Lemke and Y.-C. Wang for the cephalic furrow discussions and sharing of unpublished data; the Lemke laboratory for generous help in setting up *Clogmia* cultures; F. Frischknecht and M. Reinig for enabling and supporting the collection of *Anopheles* embryos; K. Bourtzis for kindly providing *Ceratitis* pupae; C. Blanch-Mercader for feedback on simulations; J. Roscito for text revisions; M. Marass for crucial editorial input; M. Akam for drawing B.C.V.'s attention to the cephalic furrow; and A. Martinez-Arias and T. Brunet for constructive feedback. This work was supported by the MPI-CBG core funding from C.D.M. and P.T. laboratories and by a European Research Council Advanced Grant (ERC-AdG 885504 GHOSTINTHESHELL) awarded to P.T. A.S. was supported by funding from the European Union's Horizon 2020 Research and Innovation Programme under grant agreement no. 829010 (PRIME). B.C.V. was supported by an EMBO Long-Term Fellowship (ALTF 74–2018).

**Author contributions** B.C.V. and P.T. conceived the study. B.C.V. designed experiments, generated the fly stocks, acquired microscopy data, performed in situ hybridization, and processed and analysed the in vivo data. M.B.C. conceived and conducted the laser ablation and cauterization experiments, and analysed the laser ablation and strain rate data. C.D.M., A.K. and A.S. designed the model. A.K. and A.S. programmed the model, performed the simulations and analysed the in silico data. B.C.V. wrote the initial versions of the manuscript. All authors revised and contributed to the text. B.C.V. and P.T. polished the final version of the manuscript.

**Funding** Open access funding provided by Max Planck Society.

**Competing interests** The authors declare no competing interests.

**Additional information**
**Correspondence and requests for materials** should be addressed to Bruno C. Vellutini or Pavel Tomancak.

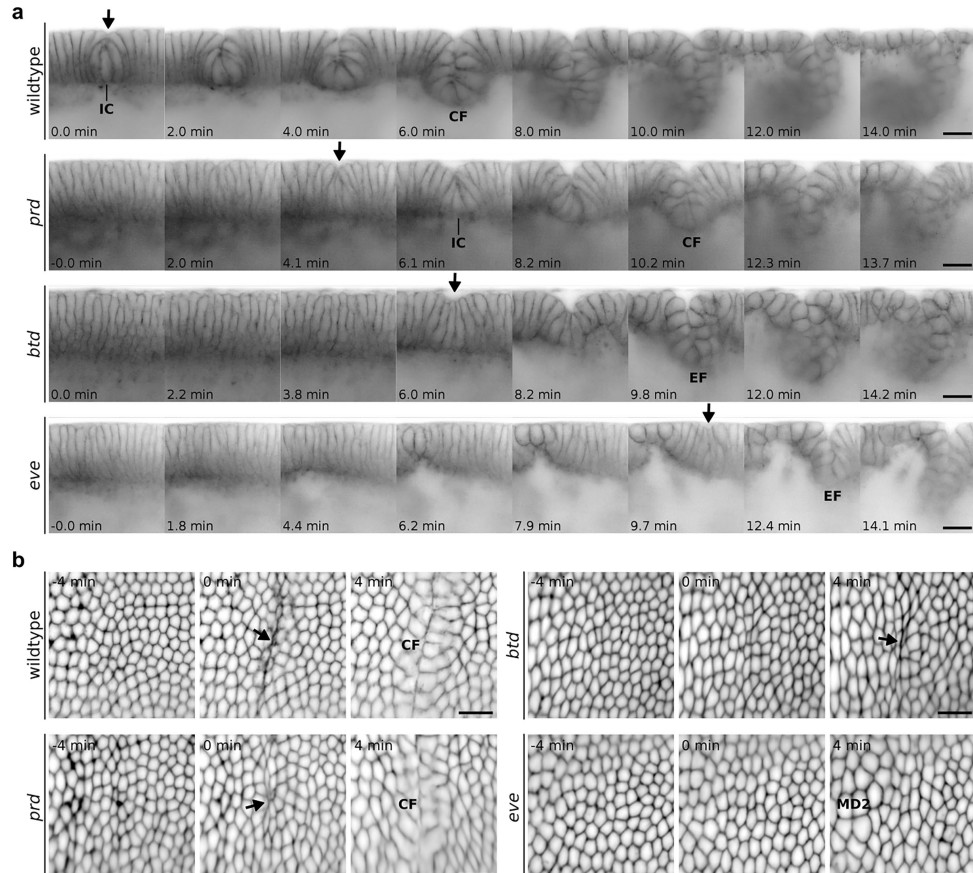

**Extended Data Fig. 1 | Disruption of initiator cell behavior in cephalic furrow mutants. a**, Profile view of the head–trunk boundary epithelium in wildtype embryos and *prd*, *btd*, and *eve* mutants. Samples synchronized by the end of cellularization (0.0 min). Arrow indicates the first infolding of the tissue. Initiator cells (IC) in wildtype embryos are tightly connected to adjacent cells, which become arched in the early invagination of the cephalic furrow (CF). This arrangement is perturbed in mutants. In *prd*, adjacent cells do not arch over the initiator row (6.1 min), and the invagination is delayed. In *btd*, there is no initiator shortening, only a partial apical constriction bulging the epithelium (6.0 min). In *eve*, there is no shortening or apical constriction. Both *btd* and *eve* form ectopic folds (EF) about ten minutes after the end of cellularization. Scale bars = 20 μm. **b**, Surface view of the head–trunk boundary epithelium on 2D cartographic projections showing abnormal apical constriction in *prd* and *btd* mutants and absence of this behavior in *eve* mutants. Scale bars ≈ 20 μm.

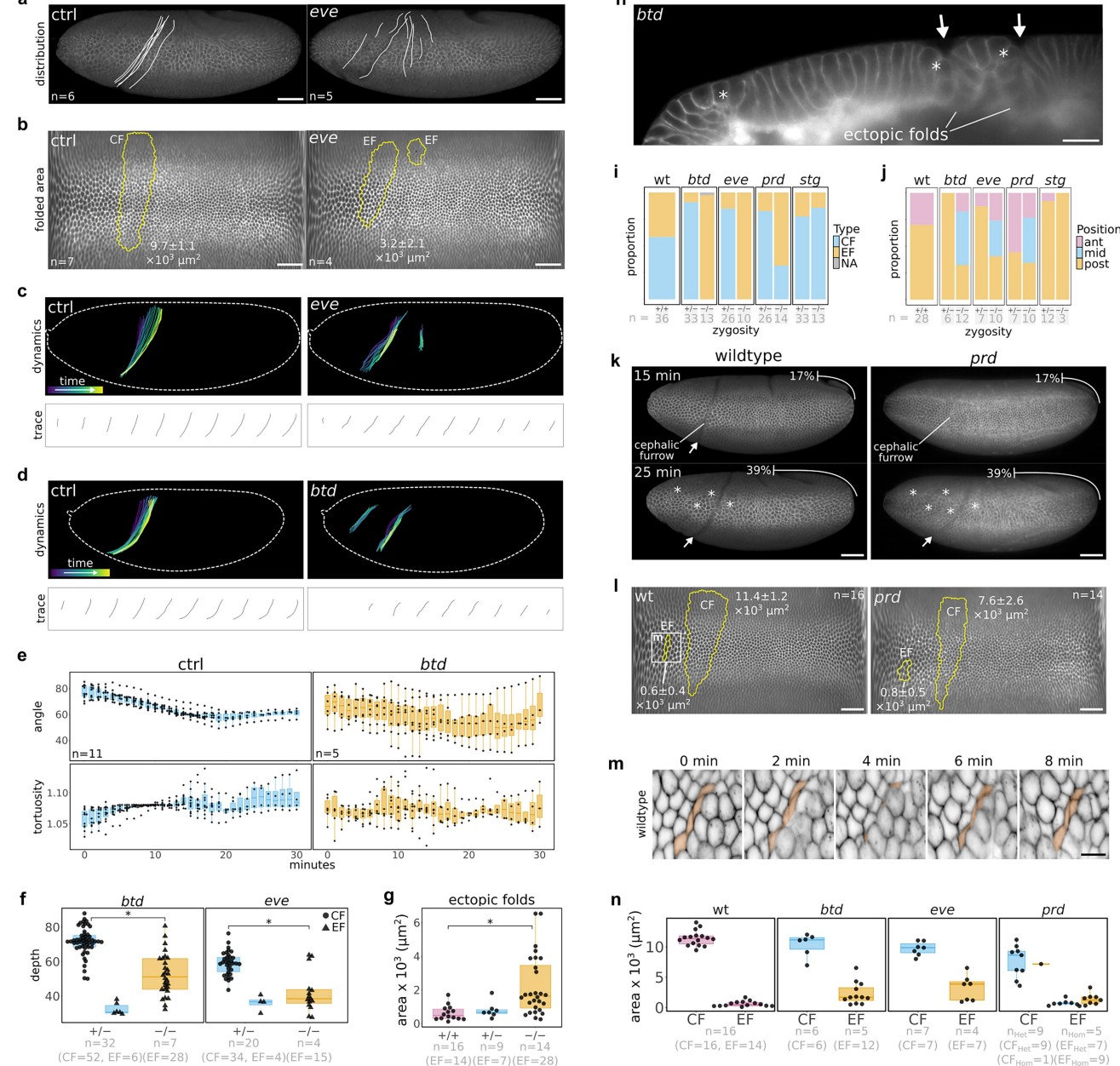

**Extended Data Fig. 2 | Differences between cephalic furrow (CF) and ectopic folds (EF). a**, Position variability in *eve* heterozygotes (n = 6) and homozygotes (n = 5). Scale bars = 50 μm. **b**, Folded area (yellow outline) in *eve* heterozygotes (n = 7) and homozygotes (n = 4). Numbers indicate average and standard deviation. Cartographic projections of lateral views. Scale bars ≈ 50 μm. **c**, Folding dynamics in *eve* mutant. **d**, Folding dynamics in *btd* mutant. **e**, Folding angle and tortuosity in *btd* mutants (n = 6). **f**, Maximum fold depth in *btd* heterozygotes (n = 32) and homozygotes (n = 7) and *eve* heterozygotes (n = 20) and homozygotes (n = 4). Ectopic folds are shallower than the cephalic furrow (p < 0.001). Data points correspond to individual folds. **g**, Area of ectopic folds in wildtype (n = 16), heterozygotes (n = 9), and homozygotes (n = 14) (pooled *btd*, *eve*, and *prd*). Ectopic folds in wildtype are smaller than in mutants (p < 0.001). **h**, Multiple ectopic folds (arrows) near dividing cells (asterisks) in *btd* mutant. Scale bar = 20 μm. **i**, Proportion of fold types in wildtype (n = 36), *btd* heterozygotes (n = 33) and homozygotes (n = 13), *eve* heterozygotes (n = 26) and homozygotes (n = 10), *prd* heterozygotes (n = 26) and homozygotes (n = 14), *stg* heterozygotes (n = 33) and homozygotes (n = 13). **j**, Proportion of ectopic folding positions (anterior, middle, posterior) in wildtype (n = 28), *btd* heterozygotes (n = 6) and homozygotes (n = 12), *eve* heterozygotes (n = 7) and homozygotes (n = 10), *prd* heterozygotes (n = 7) and homozygotes (n = 10), and *stg* heterozygotes (n = 12) and homozygotes (n = 3). **k**, Lateral view of wildtype and *prd* embryos. Scale bar = 50 μm. **l**, Folded area (yellow outline) in wildtype (n = 16) and *prd* (n = 14) embryos. Numbers indicate average and standard deviation. Cartographic projections of lateral views. Scale bars ≈ 50 μm. **m**, Ectopic folding in wildtype embryo from **l**. Scale bar = 10 μm. **n**, Folded area by fold type in wildtype (n = 16), *btd* heterozygotes (n = 6) and homozygotes (n = 5), *eve* heterozygotes (n = 7) and homozygotes (n = 4), *prd* heterozygotes (n = 9) and homozygotes (n = 5).

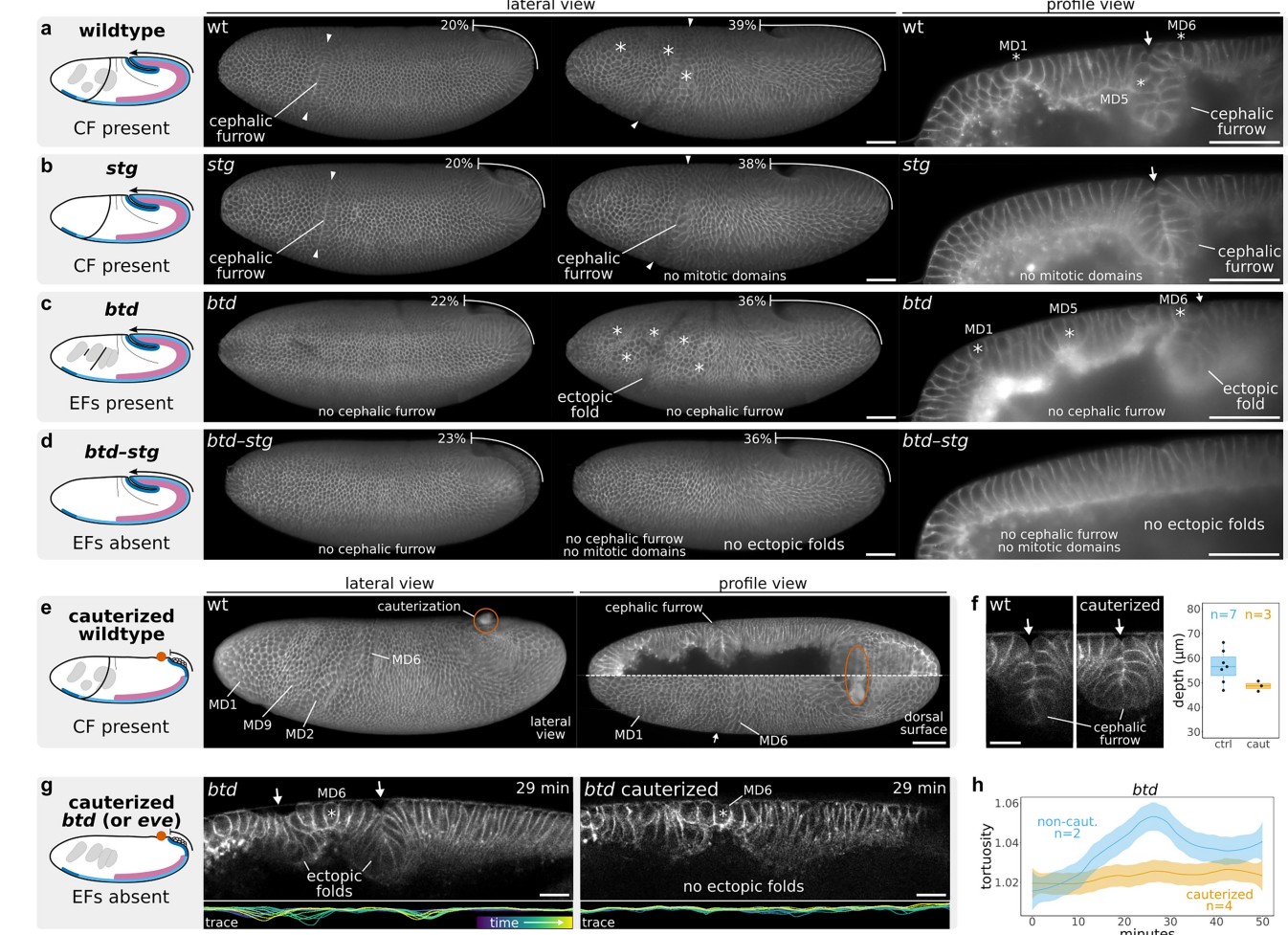

**Extended Data Fig. 3 | Additional in vivo experiments in cephalic furrow mutants. a**, Formation of the cephalic furrow (CF) and mitotic domains (MD) in wildtype embryos. Arrows indicate tissue folds. Asterisks indicate mitotic domains. Scale bars = 50 μm. **b**, Formation of the cephalic furrow in *stg* mutants. Scale bars = 50 μm. **c**, Formation of mitotic domains and ectopic folds (EFs) in *btd* mutants. Scale bars = 50 μm. **d**, Absence of ectopic folds in *btd–stg* double mutants. Scale bar = 50 μm. **e**, Germ band cauterization (orange) in wildtype embryo under multiview lightsheet microscopy. We quantified and corroborated the phenotype under confocal microscopy (see **f**). Scale bar = 50 μm. **f**, Cephalic furrow in non-cauterized and cauterized wildtype embryos. Scale bar = 20 μm. **g**, Germ band cauterizations in *btd* mutants under confocal microscopy showing the traces of epithelial deformations over time. Scale bars = 20 μm. **h**, Tortuosity of epithelial traces in *btd* non-cauterized (n = 2) and cauterized (n = 3). Plots show mean predicted values from regression with a 95% confidence interval shaded band.

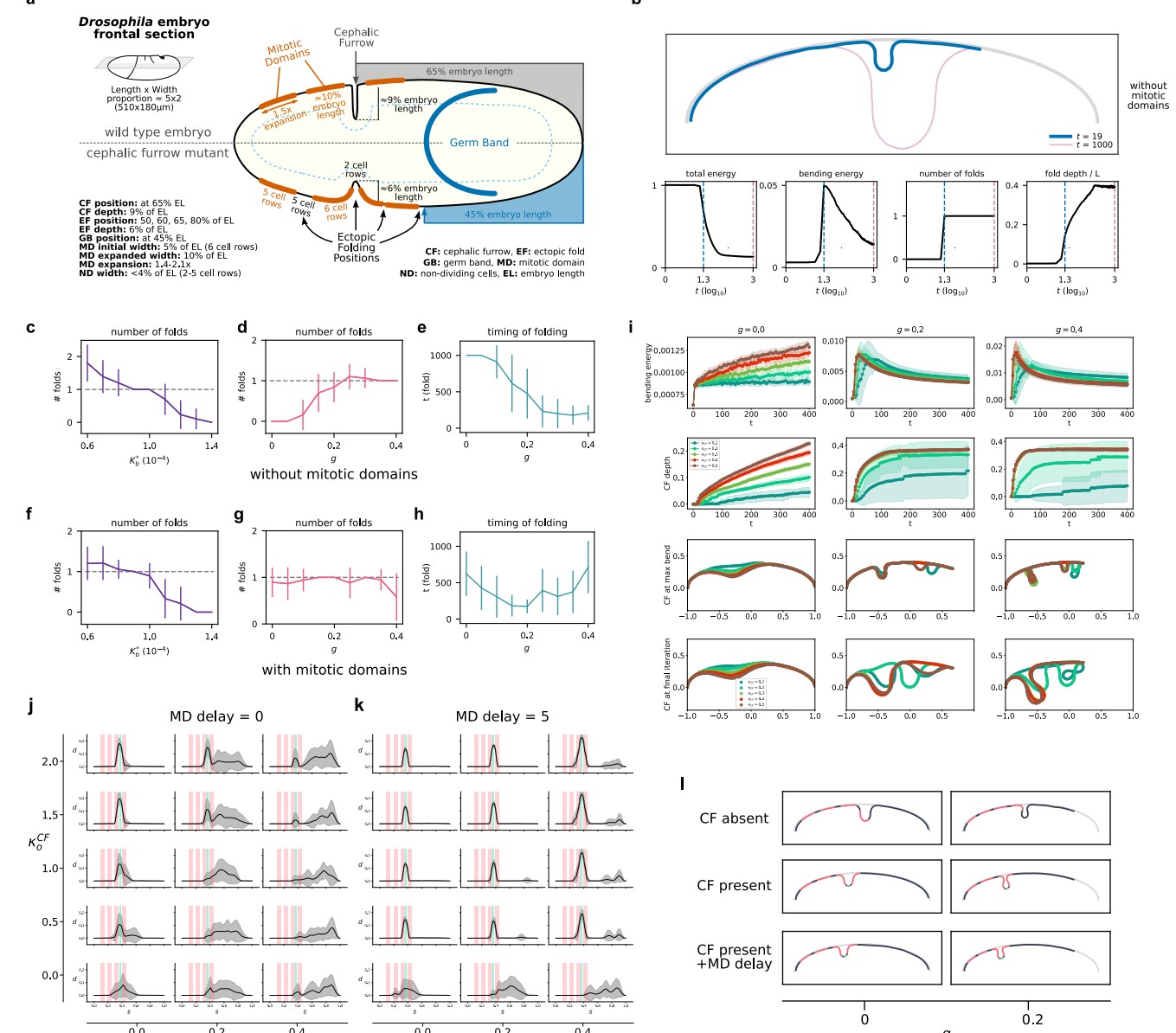

**Extended Data Fig. 4 | Model properties and parameter sweeps.**
**a**, Reference embryonic proportions in wildtype and cephalic furrow mutants used for the model. Sizes and positions of embryonic traits are relative to embryo length. **b**, Example simulation ($K_b^* = 7 \times 10^{-5}$ and $g = 0.3$) showing the tissue shape at $t = 19$ (blue) and $t = 1000$ (pink). These timepoints are marked as dashed lines in the descriptive plots below. $t = 1$ corresponds to $10^5$ computational steps. The X axis is in $log_{10}$ scale to improve the visualization. **c–e**, Parameter sweeps without mitotic domains for the number of folds by bending rigidity using $g = 0.3$ (**c**), the number of folds by germ band extension ($g$) using $K_b^* = 1.0 \times 10^{-4}$ (**d**), and the timing of folding by germ band extension using

$K_b^* = 1.0 \times 10^{-4}$ (**e**). The plot shows the mean value and standard deviation across simulations (n = 20). **f–h**, Parameter sweeps with mitotic domains for the same conditions as above. **i**, Parameter sweeps for cephalic furrow formation for different values of $\kappa_o^{CF}$ (colors) and germ band extension. **j**, Fine-grained parameter sweep of ectopic folding at different $\kappa_o^{CF}$ values with the simultaneous formation of the cephalic furrow and mitotic domains ($t_{MD} = 0$). **k**, Fine-grained parameter sweep of ectopic folding at different $\kappa_o^{CF}$ values with a relative delay between cephalic furrow and mitotic domain formation ($t_{MD} = 5$). Values of $\kappa_o^{CF}$ are shown in units of $1/L$. $t_{MD} = 1$ corresponds to $10^5$ computational timesteps. **l**, Representative simulations from Fig. 3g at 0 and 20% of germ band extension.

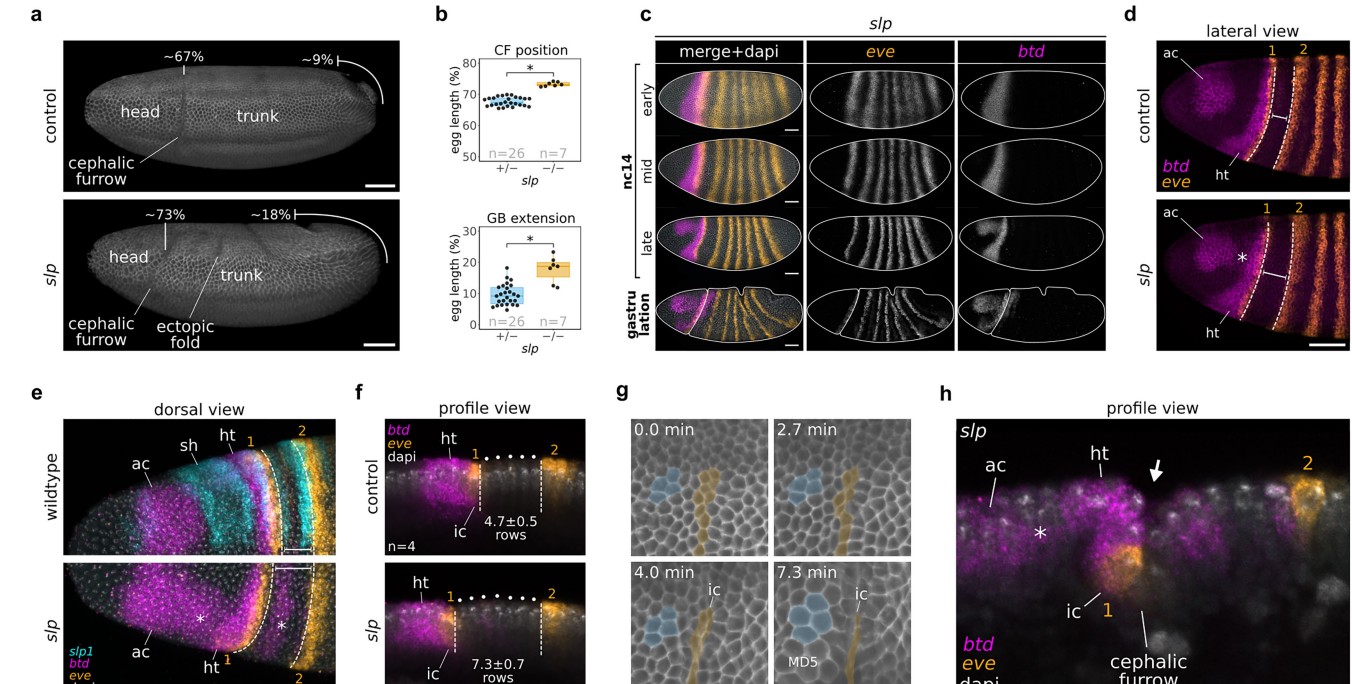

**Extended Data Fig. 5 | Analyses of *slp* mutants in *Drosophila*. a**, Lateral view of *slp* mutants at the onset of initiator cell behavior. Cephalic furrow formation is delayed and shifted forward. Scale bars = 50 μm. **b**, Position of the cephalic furrow (CF) and germ band (GB) in *slp* mutants. CF is displaced anteriorly, and GB is further extended (p < 0.001) between heterozygotes (n = 26) and homozygotes (n = 7). **c**, Expression of *btd* and *eve* in *slp* mutants before gastrulation. Scale bars = 50 μm. **d**, Expression of *btd* and *eve* in the head–trunk boundary of *slp* mutants. Distance between *eve* stripe 1 and 2 (dashed lines) is larger in *slp* embryos. Asterisk indicates ectopic expression of *btd* between acron (ac) and head–trunk (ht) domains. Scale bars = 50 μm. **e**, Dorsal view of expression domains in *slp* mutants. sh, *slp1* head domain. Asterisks indicate ectopic expression of *btd*. Scale bars = 20 μm. **f**, Profile view showing the increased number of cell rows between *eve* stripe 1 and 2 in *slp* mutants. ic, initiator cells. Scale bars = 20 μm. **g**, Behavior of initiator (orange) and dividing (blue) cells in *slp* mutants. Scale bar = 20 μm. **h**, Profile view showing the asymmetric cephalic furrow in *slp* mutants. Scale bar = 20 μm.

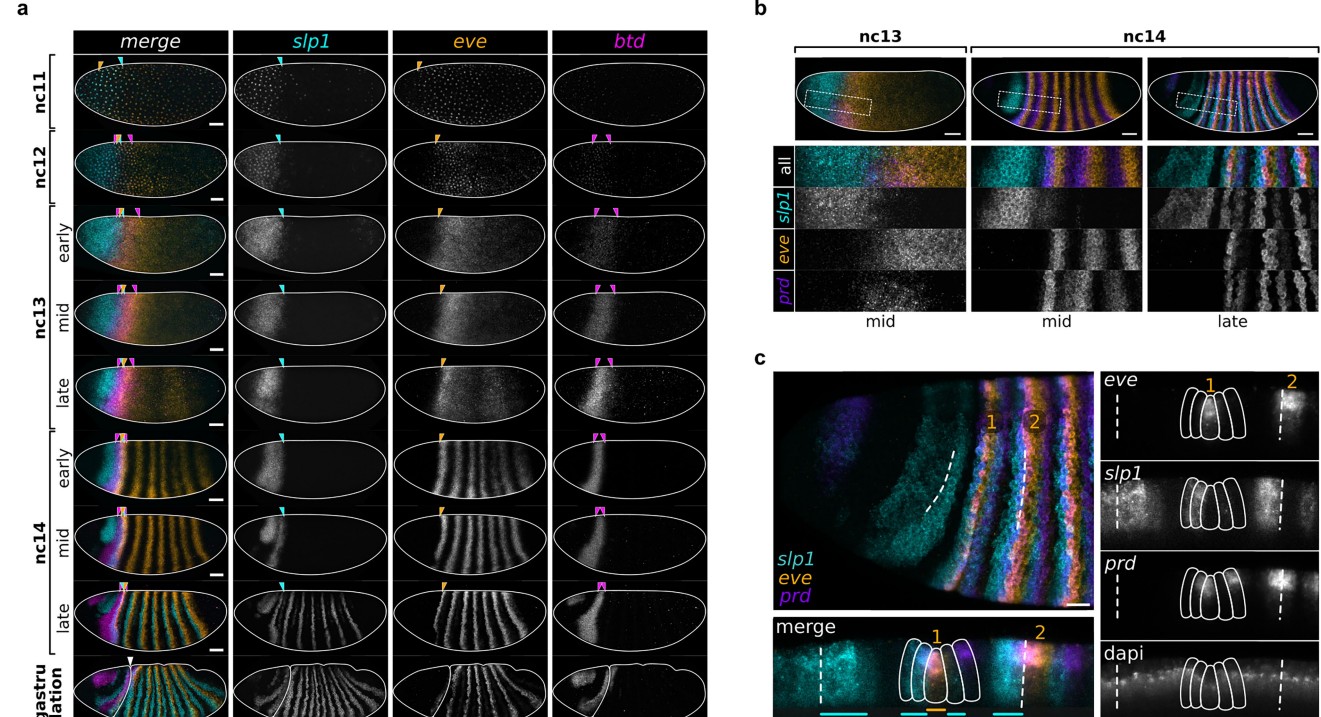

**Extended Data Fig. 6 | Genetic patterning of the head–trunk boundary in *Drosophila*. a**, Expression of *slp1*, *eve*, and *btd* in wildtype from nuclear cycle (nc) 11 to gastrulation. Scale bars = 50 μm. **b**, Expression of *prd*, *slp1*, and *eve* in wildtype embryos. Scale bars = 50 μm. **c**, Expression of *prd*, *slp1*, and *eve* in wildtype embryos in a lateral view of the head and a profile view of the head–trunk epithelium. Scale bars = 20 μm.

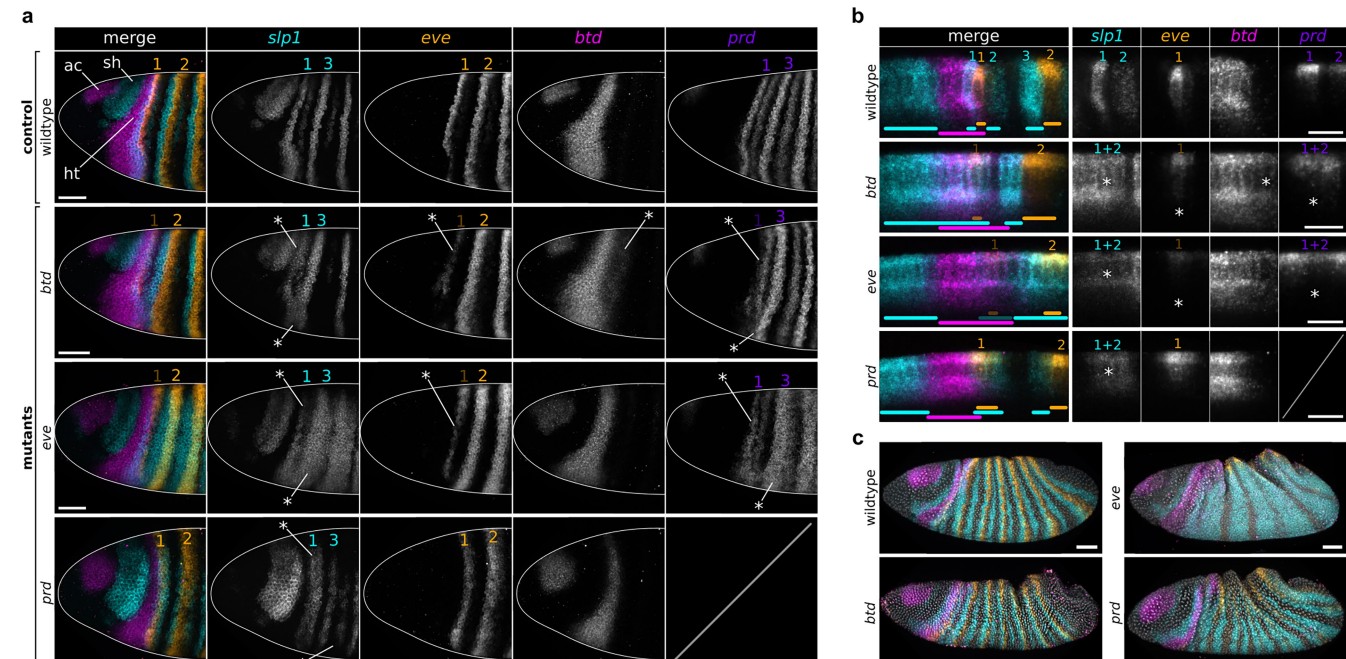

**Extended Data Fig. 7 | Disruption of gene expression patterns at the head–trunk boundary of cephalic furrow mutants in *Drosophila*. a**, Expression of *btd*, *eve*, *slp1*, and *prd* in *btd*, *eve*, and *prd* mutants. Asterisks indicate altered expression patterns. Stripe numbers are color-coded for each gene. Scale bars = 50 μm. **b**, Profile views of **g** showing the altered gene expression patterns (asterisks) of epithelial cells at the head–trunk boundary of cephalic furrow mutants. Scale bars = 20 μm. **c**, Lateral views of *btd*, *eve*, and *slp1* expression in *btd*, *eve*, and *prd* mutants after gastrulation. Scale bars = 50 μm. ac: *btd* acron domain, sh: *slp* head domain, ht: *btd* head–trunk domain.

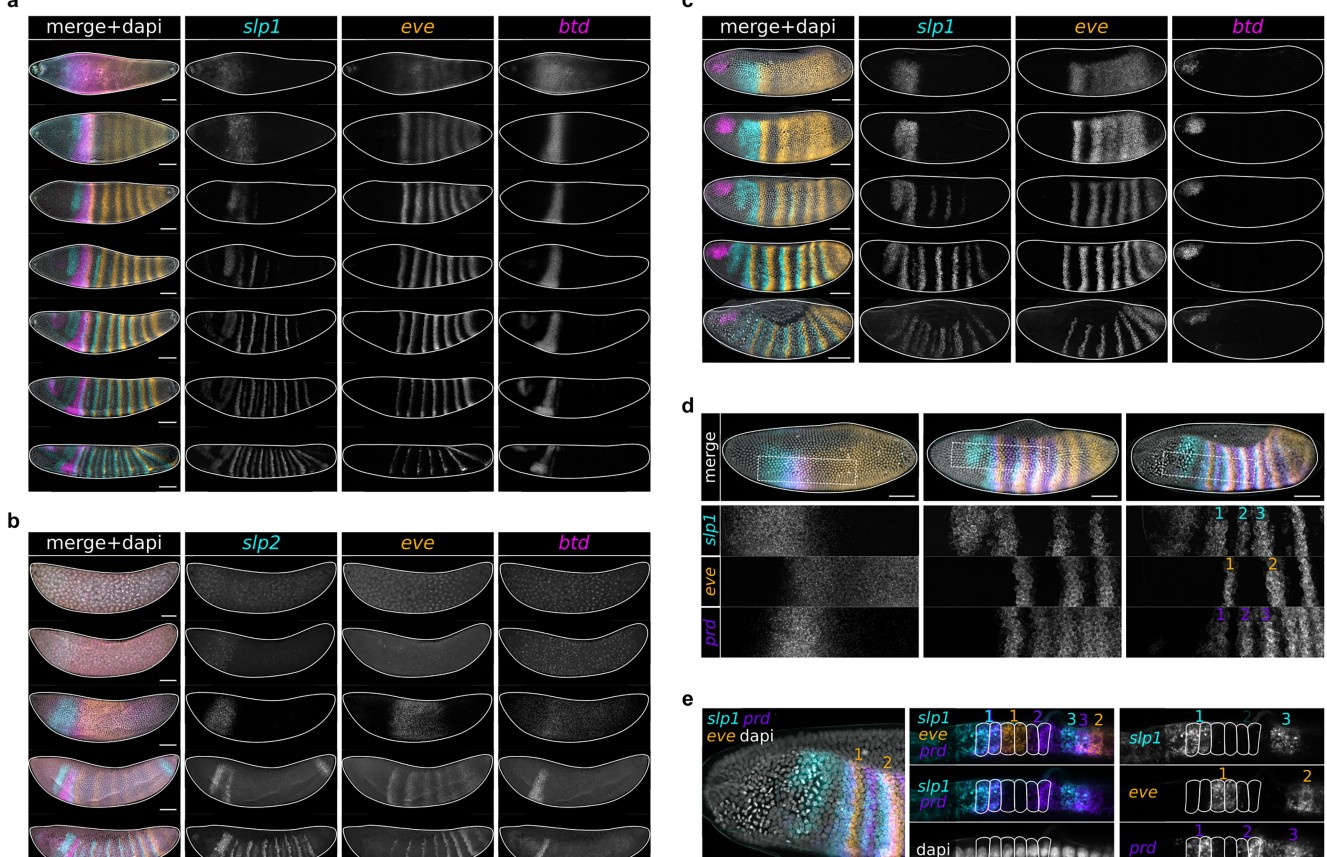

**Extended Data Fig. 8 | Genetic patterning of the head–trunk boundary in other dipteran species. a**, Expression of *slp1*, *eve*, and *btd* in *Ceratitis* developmental stages before gastrulation. Scale bars = 100 µm. **b**, Expression of *slp2*, *eve*, and *btd* in *Anopheles* developmental stages before gastrulation. Scale bars = 50 µm. **c**, Expression of *slp1*, *eve*, and *btd* in *Clogmia* developmental stages before gastrulation. Scale bars = 50 µm. **d**, Expression of *slp1*, *eve*, and *prd* in *Clogmia* before gastrulation. Scale bars = 50 µm. **e**, Expression of *slp1*, *eve*, and *prd* at the head–trunk boundary of *Clogmia* showing a lateral and profile views of the epithelium. Stripe numbers are color-coded for each gene. Scale bars = 20 µm.

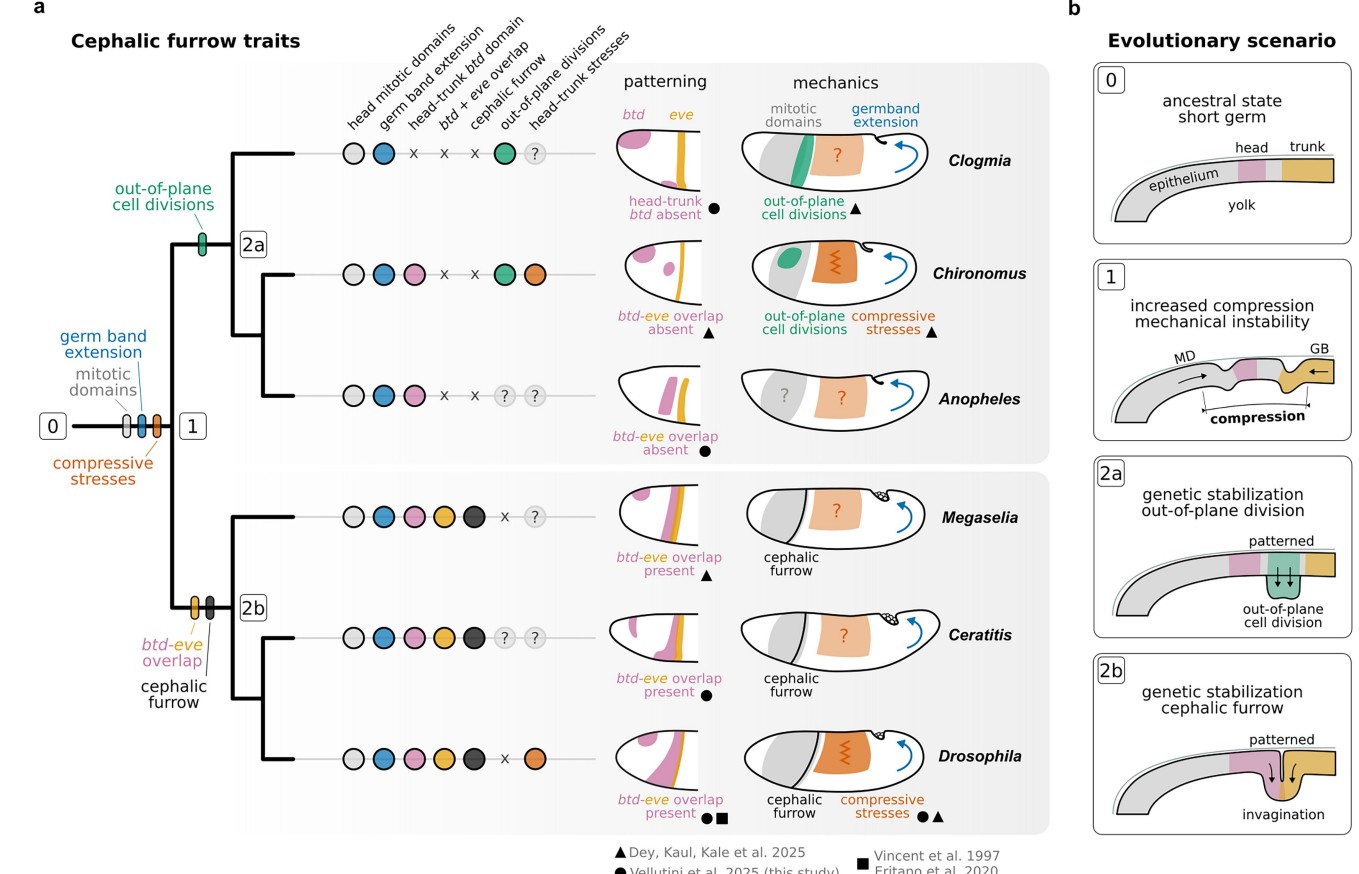

**a** Cephalic furrow traits

**b** Evolutionary scenario

▲ Dey, Kaul, Kale et al. 2025
● Vellutini et al. 2025 (this study)
■ Vincent et al. 1997
Eritano et al. 2020

**Extended Data Fig. 9 | Extended summary figure of cephalic furrow evolution including data from *Anopheles* and *Ceratitis*.** Please refer to the legend of Fig. 5 for the full details. **a**, Cephalic furrow traits mapped onto a simplified dipteran phylogeny. **b**, Evolutionary scenario for the origin of morphogenetic innovations in Diptera.

# Reporting Summary

## Statistics

For all statistical analyses, confirm that the following items are present in the figure legend, table legend, main text, or Methods section.

| n/a | Confirmed | |
|---|---|---|
| ☐ | ☒ | The exact sample size (*n*) for each experimental group/condition, given as a discrete number and unit of measurement |
| ☐ | ☒ | A statement on whether measurements were taken from distinct samples or whether the same sample was measured repeatedly |
| ☐ | ☒ | The statistical test(s) used AND whether they are one- or two-sided *Only common tests should be described solely by name; describe more complex techniques in the Methods section.* |
| ☒ | ☐ | A description of all covariates tested |
| ☒ | ☐ | A description of any assumptions or corrections, such as tests of normality and adjustment for multiple comparisons |
| ☐ | ☒ | A full description of the statistical parameters including central tendency (e.g. means) or other basic estimates (e.g. regression coefficient) AND variation (e.g. standard deviation) or associated estimates of uncertainty (e.g. confidence intervals) |
| ☐ | ☒ | For null hypothesis testing, the test statistic (e.g. *F*, *t*, *r*) with confidence intervals, effect sizes, degrees of freedom and *P* value noted *Give P values as exact values whenever suitable.* |
| ☒ | ☐ | For Bayesian analysis, information on the choice of priors and Markov chain Monte Carlo settings |
| ☒ | ☐ | For hierarchical and complex designs, identification of the appropriate level for tests and full reporting of outcomes |
| ☒ | ☐ | Estimates of effect sizes (e.g. Cohen's *d*, Pearson's *r*), indicating how they were calculated |

*Our web collection on statistics for biologists contains articles on many of the points above.*

## Software and code

Policy information about availability of computer code

Data collection
We acquired our live imaging data using a Zeiss Lightsheet Z.1 microscope running ZEN 2014 SP1 v9.2.10.54. For imaging fixed samples from in situ hybridization experiments, we used an inverted Zeiss LSM 700 confocal microscope running ZEN 2012 SP5 FP3 v14.0.25. We performed laser cauterization experiments in two microscope setups, a lightsheet Luxendo MuVi SPIM with a photomanipulation module and a confocal Zeiss LSM 780 NLO with multiphoton excitation running ZEN Black v14.024.201. For the laser ablation experiments, we used a Yokogawa CSU-X1 spinning disk confocal microscope running AndorIQ for acquisition and an inverted Zeiss Axio Observer.Z1 spinning disk confocal microscope running ZEN Blue v3.2 with a Rapp OptoElectronic setup for photo-manipulation running SysCon2. All the data supporting the findings of this study are available on the Zenodo repositories with the identifiers https://doi.org/10.5281/zenodo.7781947 (imaging data), https://doi.org/10.5281/zenodo.7784906 (simulation data), and https://doi.org/10.5281/zenodo.7781916 (high-resolution figures and videos).

Data analysis
We processed imaging datasets and performed most image analyses using custom macros and plugins in Fiji/ImageJ v2.16.0/1.54p; Java 1.8.0_172. To improve the signal-to-noise ratio and Z-resolution of lateral lightsheet datasets, we used CARE CSBDeep v0.3.0. To generate 3D renderings, we used the Fiji plugin 3Dscript v0.2.1. We created cartographic projections using the ImSAnE toolbox v3a7be24 pipeline using MATLAB R2015b and ilastik v1.3.3b2. We used the Fiji plugin bUnwarpJ v2.6.13 to register the surface of individual embryos. We generated color-coded temporal projections using the script Temporal Color Code v101122 with the mpl-viridis color map, both bundled in Fiji. We used the Fiji plugin MorphoLibJ v1.6.0 to segment, measure, and color-code the cell apical areas, and the plugin Linear Stack Alignment with SIFT v1.5.0 to register cells between timepoints. We extracted measurements from the epithelium outlines using the Fiji plugin Analyze Skeleton v3.4.2. We generated kymographs using the Multi Kymograph v3.0.1 Fiji plugin. We created illustrations and assembled figure plates using Inkscape v1.2.2. For microscopy videos, we exported the original timelapse stacks to uncompressed AVI using Fiji and converted them to MPEG-4 format using HandBrake v1.6.1. We performed the data wrangling, statistical analyses, and plotting in R v4.2.1 using R Markdown notebooks in RStudio v2022.7.2.576, and in Python 3.10.7 using Jupyter notebooks v6.5.4. All the code necessary to reproduce the data

processing and downstream analyses of this study are available on the Zenodo repositories with the identifiers https://doi.org/10.5281/zenodo.7781947 (imaging analyses) and https://doi.org/10.5281/zenodo.7784906 (model analyses).

For manuscripts utilizing custom algorithms or software that are central to the research but not yet described in published literature, software must be made available to editors and reviewers. We strongly encourage code deposition in a community repository (e.g. GitHub). See the Nature Portfolio guidelines for submitting code & software for further information.

## Data

Policy information about availability of data

All manuscripts must include a data availability statement. This statement should provide the following information, where applicable:
- Accession codes, unique identifiers, or web links for publicly available datasets
- A description of any restrictions on data availability
- For clinical datasets or third party data, please ensure that the statement adheres to our policy

All the data supporting the findings of this study have been deposited on Zenodo. The main repository containing the raw data, analyses pipelines, and source files for figures and text is available under the identifier https://doi.org/10.5281/zenodo.7781947. The imaging data for the lightsheet and in situ hybridization experiments is available under the identifier https://doi.org/10.5281/zenodo.15876638. The imaging data for the laser perturbation experiments is available under the identifier https://doi.org/10.5281/zenodo.15876646. The theory repository containing the code and scripts of the model, the output data of simulations, and notebooks of analyses and plotting is available under the identifier https://doi.org/10.5281/zenodo.7784906. The media repository containing the high-resolution figures and videos is available under the identifier https://doi.org/10.5281/zenodo.7781916.

## Research involving human participants, their data, or biological material

Policy information about studies with human participants or human data. See also policy information about sex, gender (identity/presentation), and sexual orientation and race, ethnicity and racism.

| | |
|---|---|
| Reporting on sex and gender | NA |
| Reporting on race, ethnicity, or other socially relevant groupings | NA |
| Population characteristics | NA |
| Recruitment | NA |
| Ethics oversight | NA |

Note that full information on the approval of the study protocol must also be provided in the manuscript.

# Field-specific reporting

Please select the one below that is the best fit for your research. If you are not sure, read the appropriate sections before making your selection.

☒ Life sciences  ☐ Behavioural & social sciences  ☐ Ecological, evolutionary & environmental sciences

For a reference copy of the document with all sections, see nature.com/documents/nr-reporting-summary-flat.pdf

# Life sciences study design

All studies must disclose on these points even when the disclosure is negative.

| | |
|---|---|
| Sample size | We performed no prior estimation of sample sizes for experiments. Following previous studies in the field, we defined the number of experiments and sample size based on the nature of the sample (whether it is well know or entirely novel genotype, for example), on the variability of the observed phenotypes (larger sample size for more variable phenotypes), and on the complexity of the experimental setup (challenging setups had smaller sample size). The exact sample sizes are described in the figure legends. |
| Data exclusions | We excluded no datasets from the analyses. |
| Replication | The phenotypes that we report in this study were reproducible across multiple independent experiments. We performed the experiments in different periods, using different batches, and different experimental and microscopy setups. For the live lightsheet imaging, we performed 7 experiments in btd mutants (total of 50 embryos), 5 experiments in eve mutants (total of 36 embryos), 3 experiments in prd mutants (total of 41 embryos), 6 experiments in slp mutants (total of 39 embryos), 3 experiments in stg mutants (total of 46 embryos), 6 experiments in btd-eve double mutants (total of 35 embryos), and 2 experiments in wildtype embryos (total of 36 embryos). For the germ band cauterization, we performed 6 experiments in btd mutants (total of 10 embryos), 5 experiments in eve mutants (total of 10 embryos), and 8 experiments in wildtype embryos (total of 12 embryos). For the gene expression, the wildtype patterns of btd, eve, and slp were highly consistent across 3 experiments in Drosophila (total of 26 embryos), 3 experiments in Ceratitis (total of 38 embryos), 4 experiments in Anopheles (total of 43 embryos), and 4 experiments in Clogmia (total of 44 embryos). We also obtained consistent patterns of prd expression across 4 experiments in Drosophila (total of 10 embryos) and 1 experiment in Clogmia (total of 20 embryos). Expression patterns in mutant embryos were |

repeatable across 4 independent experiments in slp mutants (total of 30 embryos), 5 experiments in btd mutants (total of 20 embryos), 2 experiments in eve mutants (total of 12 embryos), and 2 experiments in prd mutants (total of 12 embryos).

| | |
|---|---|
| Randomization | We performed no randomization of samples into experimental groups. However, our control and experimental groups was determined by the zygosity of the embryos (heterozygote or homozygote) which follows genetic inheritance patterns and, therefore, were not known prior or during data acquisition. |
| Blinding | We performed no blinding of sample for experiments. However, the experimental group (genotype) of the embryos remained unknown during imaging data acquisition and perturbation experiments (laser ablations and cauterizations). We determined the genotypes during data analyses. |

# Reporting for specific materials, systems and methods

We require information from authors about some types of materials, experimental systems and methods used in many studies. Here, indicate whether each material, system or method listed is relevant to your study. If you are not sure if a list item applies to your research, read the appropriate section before selecting a response.

## Materials & experimental systems

| n/a | Involved in the study |
|---|---|
| ☒ | ☐ Antibodies |
| ☒ | ☐ Eukaryotic cell lines |
| ☒ | ☐ Palaeontology and archaeology |
| ☐ | ☒ Animals and other organisms |
| ☒ | ☐ Clinical data |
| ☒ | ☐ Dual use research of concern |
| ☒ | ☐ Plants |

## Methods

| n/a | Involved in the study |
|---|---|
| ☒ | ☐ ChIP-seq |
| ☒ | ☐ Flow cytometry |
| ☒ | ☐ MRI-based neuroimaging |

## Animals and other research organisms

Policy information about studies involving animals; ARRIVE guidelines recommended for reporting animal research, and Sex and Gender in Research

| | |
|---|---|
| Laboratory animals | Drosophila melanogaster, Ceratitis capitata, Clogmia albipunctata, and Anopheles stephensi. |
| Wild animals | The study did not involve wild animals. |
| Reporting on sex | We did not perform any sex-based analyses as it is not possible to identify the sex of individual flies during embryonic stages. |
| Field-collected samples | The study did not involve samples collected from the field. |
| Ethics oversight | No ethical approval is required for research on Drosophila and other dipteran flies. |

Note that full information on the approval of the study protocol must also be provided in the manuscript.

## Plants

| | |
|---|---|
| Seed stocks | *Report on the source of all seed stocks or other plant material used. If applicable, state the seed stock centre and catalogue number. If plant specimens were collected from the field, describe the collection location, date and sampling procedures.* |
| Novel plant genotypes | *Describe the methods by which all novel plant genotypes were produced. This includes those generated by transgenic approaches, gene editing, chemical/radiation-based mutagenesis and hybridization. For transgenic lines, describe the transformation method, the number of independent lines analyzed and the generation upon which experiments were performed. For gene-edited lines, describe the editor used, the endogenous sequence targeted for editing, the targeting guide RNA sequence (if applicable) and how the editor was applied.* |
| Authentication | *Describe any authentication procedures for each seed stock used or novel genotype generated. Describe any experiments used to assess the effect of a mutation and, where applicable, how potential secondary effects (e.g. second site T-DNA insertions, mosiacism, off-target gene editing) were examined.* |

