## [Peer Review file · Nature]

Patterned invagination prevents mechanical instability during gastrulation

Corresponding Author: Dr Pavel Tomancak

Version 0:

Reviewer comments:

Referee #1

(Remarks to the Author)

The authors have framed their work around a question that lies at the interface of the fields of morphogenesis and evolution. The central question can be summarized as follows: While mechanical forces are essential for morphogenesis, are they also relevant for the evolution of morphogenetic processes? To address this question, the authors conducted an investigation into the function of the cephalic furrow (CF) in *Drosophila*, a transient and deep fold that forms at the boundary between the head and the trunk during early embryogenesis. They reported that in the absence of genes controlling CF formation, additional extra folds of greater magnitude are produced in response to concomitant embryo elongation and local cell proliferation. Using simulations, they confirmed that compressive stresses alone are sufficient to promote extra and ectopic fold formation. As a result, they have proposed that the absence of CF triggers stronger mechanical instabilities. Furthermore, the authors compared the head-trunk genetic patterning system in *Drosophila* to that of the dipteran fly *Clogmia albipunctata* and observed that the lack of expression of *btd* correlates with the absence of CF formation. Based on these findings, the authors have suggested that mechanical instabilities could act as a selective pressure for the evolution of the CF.

While the genetic and mechanical analysis of the function and role of the CF is valuable, there are opportunities to enhance the study further. Specifically, the authors could: (i) Provide a more comprehensive characterization of the mechanisms by which changes in tissue mechanics could promote ectopic morphogenetic processes. (ii) Offer a deeper substantiation of how concomitant morphogenetic processes could be buffered by spatiotemporally patterned fold formation. While the first part of the manuscript has room for improvement, the data provided by the authors on the evolution of morphogenetic processes primarily support the idea that *btd* is necessary for CF formation. However, it remains to be demonstrated conclusively that mechanical instabilities are indeed a significant selective pressure for the evolution of morphogenetic processes. Consequently, I am unable to support the publication of this work in Nature.

1. Role of the CF in the control of extra-fold formation.

a. In their manuscript, the authors have provided a more comprehensive characterization of the phenotypes associated with *btd* and *eve*, along with an analysis of the *prd* gene's phenotype. Notably, the authors' data support that both in wild-type and mutant conditions, extra folds form in the head and trunk regions. They have also demonstrated that the number and shape of these extra folds are more pronounced in mutants compared to the wild-type. However, a limitation of the genetic analysis performed by the authors is that they did not differentiate whether the loss of mechanical forces associated with fold formation or a distinct function of *btd* or *eve* alone is sufficient to trigger extra-fold formation. While Dey et al. only commented on the formation of a misplaced fold in *btd* or *eve* mutants, they indeed utilized a specific *eve* mutant that affects only its expression in the head-trunk stripe, along with optogenetic inactivation of Rho, to demonstrate that a misplaced fold can indeed form in the head-trunk region under such mutant conditions. This approach allows for a more focused study of the role of local gene patterning and the contribution of local contractility.

b. The statement, "Mitotic expansions always precede ectopic folding (Figure 2c, Figure S1a)," would benefit from further support through quantification.

c. The authors mentioned, "To estimate the forces acting on the infolding tissue, we measured the rate of tissue deformation

(strain rate) at the head–trunk boundary using particle image velocimetry." This assertion is critical in demonstrating that physical forces are the driving factor behind the formation of ectopic folds. While I acknowledge that the strain rate can be measured by PIV, it remains unclear how one can estimate the forces acting on the infolding tissue solely based on the strain rate. In my opinion, the analysis of the strain rate offers a quantification of the observed phenotype but does not inherently provide an estimation of the forces acting on the infolding tissue. Would it be possible for the authors to consider using laser ablation to further substantiate this point?

d. The authors stated, "Overall, our in silico modeling and in vivo experiments suggest that the epithelial buckling at the head–trunk interface in cephalic furrow mutants only occurs when both the mitotic domain expansion and the germ band extension happen concomitantly." I believe this conclusion may be overstated. The authors have experimentally shown that both factors are necessary, but their in silico model allows them to adjust the bending energy to achieve this regime. However, there is no evidence in the model that both factors are required. The modeling primarily demonstrates that compressive stress can promote bending within a certain range of compressive stress and bending energy (an intuitive result), yet it does not definitively establish that the observed compressive stresses in vivo are sufficient to trigger mechanical instability.

e. The authors wrote, "Taken together, our physical model provides a theoretical basis that an early patterned invagination can effectively absorb compressive stresses in the tissue, preventing mechanical instabilities in the embryo epithelium during gastrulation." It would be valuable if the authors could go beyond this theoretical analysis by verifying whether the predictions of the model semi-quantitatively align with the experimental observations.

2. Role of mechanical instabilities as a selective pressure.

Expanding on their assertion that the cephalic furrow (CF) plays a role in buffering mechanical instabilities associated with cell proliferation and gastrulation, the authors delved into the evolution of genetic patterning at the head-trunk boundary. They conducted a comparative analysis of gene expression patterns between *Drosophila* and the dipteran fly *Clogmia albipunctata*, a species from the basally-branching Psychodidae family that lacks a cephalic furrow. In their analysis, the authors identified *slp* as an additional regulator of the expression pattern of the *eve* stripe 1 and *btd* and proposed that the combined expression patterns of *slp*, *prd*, *eve*, and *btd* define the unique transcriptional identity of the CF. They proceeded to compare and contrast the expression profiles of *slp*, *prd*, *eve*, and *btd* in the two species, highlighting that the key differentiation lies in the expression of *btd*. As a result, they concluded that *btd* represents a genetic innovation crucial for cephalic formation. In their abstract, they further surmised that "These data suggest that the genetic control of cephalic furrow formation was established through the integration of a new player into the ancestral head–trunk patterning system, and that mechanical instability may have been the selective pressure associated with the evolution of the cephalic furrow." To substantiate the initial claims regarding genetic patterning, it would be relevant to perform similar analyses in additional species. However, the second claim, and indeed the central claim of the manuscript, revolves around the role of mechanical instabilities as a selective pressure in the evolution of the cephalic furrow. The authors have suggested that the loss of *btd* leads to mechanical instabilities, but they cannot definitively rule out the existence of other selective pressures associated with *btd* function. Additionally, while the theoretical model provides evidence that compressive stress can promote extra-folding in conjunction with concomitant morphogenetic processes, it remains to be demonstrated whether these mechanical instabilities could indeed be detrimental to embryo development, thereby acting as a selective pressure. I am uncertain about how one can directly and conclusively establish such a claim. At a minimum, the authors should consider exploring whether: (i) The absence of CF, triggered without affecting genetic patterning, leads to changes in the robustness of development. (ii) Correcting such mechanical instabilities through alternative means can subsequently rectify the observed developmental defects and enhance viability. While I acknowledge that these experiments are complex and challenging, they are crucial in substantiating the main claim and novelty of the work.

Referee #2

(Remarks to the Author)

This is a beautiful study that combines light-sheet live imaging, genetics, state-of-the-art image analysis and morphometrics, analysis of gene expression, and computer simulations to unravel the developmental function and the evolutionary origin of a mysterious structure: the cephalic furrow (CF). Anyone who has ever looked at *Drosophila* embryos is aware of the obvious and conspicuous presence of the CF, and has also had to come to terms with the frustrating fact that this massive structure is both transient and has no clear function. This study elegantly solves this conundrum, by hypothesizing that CF invagination absorbs compressive stress generated by other morphogenetic movements (cell division within mitotic domains and germ band extension). The authors show that, when CF formation is genetically abolished, ectopic folds (EF) form instead, with more variable timing, shape and location. They also show that EF formation in those mutants requires both mitotic domains and germ band extension - so that it's really "tissue collision" that generates those folds, and that CF formation normally adverts. The authors conclude by a comparison to a dipteran without CF (*Clogmia*) and by an apt reference to the Newman-Müller theory on the origin of developmental patterning. Thus, the paper presents a completely novel hypothesis (to my knowledge), with a dual broad significance: (1) elucidating the biological meaning of a long-known structure in a major model organism; (2) showing an example of a structure that likely evolved as a self-organized byproduct of physical forces, before its formation evolved to be programmed and genetically patterned (as per Newman & Müller).

Overall I am convinced by the authors' case and feel very supportive of the paper. However, a number of small points are unclear to me and I am taking the opportunity to suggest that the authors clarify them.

Scientific questions:

* The high frequency of EF formation in wild-type (WT) embryos (80%) is perplexing, and contrasts with the lower frequencies observed in *btd*, *eve* and *prd* heterozygous embryos (12.9, 12 and 25.9% respectively) - even though these heterozygous embryos are phenotypically WT (in my understanding). Even *prd*^{-/-} embryos have lower EF frequency than the WT (42.9 vs 80.6%). Fig. S2j shows that EFs in WT flies only occupy a small area compared to mutants, but their very high frequency remains puzzling. Could the authors comment on that difference? Does something about having a half-dosage of *eve/prd/btd* reduce the frequency of EF? Alternatively (and perhaps more parsimoniously), might this be due to a difference in genetic background between strains unrelated to the loci of interest? If so, could one expect the frequency of EFs to vary between different nominally "wild-type" *Drosophila* strains?

* Related to the above: l. 240: "Compared to wildtype, the ectopic folds in *stg* mutants are less frequent" This is true, but only because the WT have a very high frequency of EF formation. The difference between heterozygous and homozygous *stg* mutants is actually small (27.3% vs 23.1%, 33 vs 13 embryos) and perhaps not significant. I suggest removing this point or arguing it differently.

* l. 333: "The function of the cephalic furrow in preventing mechanical instabilities depends on the correct positioning and timing of the invagination; it must occur at the head-trunk boundary." I am not sure the authors have shown the CF must form at the head-trunk boundary, as other positions don't seem to have been explored in the simulations or in the diverse mutant phenotypes reported (unless I missed something).

* Fig. 3e-f: since mitotic domains are said to contribute to the formation of EFs, it is unclear to me why there are more EFs without mitotic domains than with them in the ($g=0.4$ $Kb=1.2$) and ($g=0.2$ $Kb=0.8$) conditions.

Discussion points:

* Could the authors speculate on the ancestral function of *btd* before the evolution of the CF? Did it already pattern some kind of invagination? Perhaps some insights can be gleaned from its acron expression domain (in *Drosophila* and *Clogmia*), from the *Drosophila* mutant phenotype or from its ventral foregut domain in *Clogmia*?

Data presentation and phrasing:

* Fig. 1c only shows a single EF, fairly close to where the CF normally forms. However, the text and fig. 1f make it clear that there can be multiple EFs, sometimes very far from the normal CF location. This variability could be reported and depicted more extensively, by showing multiple examples of embryos with variable numbers and positions of EFs. Video S2 helps a bit (by showing an embryo with several EFs), but still only shows a single embryo. Some figure panels directly comparing multiple embryos would be better.

* l. 160: CF formation is said to be a "progressive invagination (about 14 min)" while formation of EFs is said to be "abrupt (4 min)". Looking at the figure, depending on the genotype, EFs can take up to 8 min to reach maximal/final depth, which does not seem all that abrupt. Maybe "faster" would be a more accurate term.

* l. 220-221: "in stiffer conditions ($Kb^* \sim 1.2e-4$) the germ band alone, even at its maximum 221 extension, cannot drive the formation of ectopic folds". However I do see a small ectopic fold at $Kb^* = 1.2e-4$ and $g=0.4$. Perhaps the authors meant $Kb^* \sim 1.4e-4$

* l. 313-315: "Curiously, this buffering effect diminished with the extension of the germ band for intermediate values of $KOCF$ ". I don't quite see how this is curious. Shouldn't the mechanical challenge created by germ band extension be expected to reduce the buffering effect of the CF? And shouldn't that buffering effect get stronger with $KOCF$? Also 2 sentences later: "in these conditions, the forces generated by the mitotic expansions and by the germ band extension dominate over the infolding pull of the cephalic furrow." - it's not quite clear what conditions are referred to as "these conditions" - does it mean under intermediate $KOCF$?

* l. 360-366: the verbal explanation of *slp*, *eve* and *btd* expression is somewhat confusing. Fortunately Fig. 5f-i is very clear and allows to rescue understanding. Particularly perplexing is the statement l. 364 "eve expression is initially ubiquitous": *eve* is clearly not expressed in all cells at mid-cycle 13, but in a broad domain with a sharp anterior boundary and a fuzzier posterior boundary.

* Fig. 3e: I think the fact that this panel depicts a mutant without CF should be written on the figure itself somewhere

* Figure S2d: the color bar depicting time is lacking a scale.

Referee #3

(Remarks to the Author)

In this manuscript Vellutini and colleagues address the function of the Cephalic Furrow (CF), a small transient multicellular pattern on the anterior third of the early *Drosophila* embryo which has remained a curiosity for years. The authors turn what at first sight is a somewhat insignificant feature of the gastrulating *Drosophila* embryo into an example of the evolution of morphogenetic mechanisms. The manuscript combines excellent genetics and experimental embryology with evolutionary biology. Having shown that the main role of the CF is to buffer mechanical stress in *Drosophila*, they go on to show that other

flies from a different evolutionary lineage, whose embryos lack a CF, have evolved a different mechanism to cope with the same stresses. The experimental work is complemented with an interesting and carefully crafted model.

The authors start by characterising the morphological phenotypes of ectopic fold formation in gastrulating *Drosophila* embryos with impaired cephalic furrow (CF) formation. This leads them to identify a correlation between ectopic fold formation with mitotic domains (MD) cell division activity and germ band extension (GBE) supported by extrapolating local strain rates based on imaging data. The contribution of MD and GBE to mechanical instability is further explored through mechanical simulations, experimental perturbations (e.g. genetic, cauterization) and laser dissection recoil speed measurements, testing simulation predictions allowing to explore properties and developmental events contributions towards tissue mechanical instability and morphogenesis. They show that GBE is not required for CF formation and develop physical model predictions arguing that CF formation can absorb compressive stresses preventing ectopic furrow formation resulting from mechanical instabilities.

A loss of function screen of ~50 candidate genes was then carried out to identify genes involved in CF formation in *Drosophila* resulting in the identification 3 new regulatory genes, with sloppy paired (*slp*) mutant flies exhibiting the strongest phenotype. Members of this family of Transcription Factors, are shown to regulate the position of the buttonhead (*btd*) and even-skipped (*eve*) genes in an overlapping domain that establishes the position of the CF. The authors then look at flies that form (*Drosophila*) or do not form (*Clogmia*) a CF and notice that the trunk-head boundary expression domain of buttonhead is absent from the latter with the remaining CF regulatory genes *slp*, *eve* and *prd* having conserved expression patterns across both species. The authors propose the acquisition of the *btd* trunk-head expression was a central event in the evolution of CF formation. It is also reported that *Clogmia* exhibits out-of-plane head cell divisions, a likely alternative mechanism to resolve mechanical instabilities within species that do not form CF. The manuscript does a good job at exploring the role of mechanical stability in developmental robustness, complemented with a mechanical model that provides an understanding of the process.

The authors provocatively argue that there might be a role for mechanical stabilities in the evolution of morphogenesis. The reported new observations and findings are of interest to the fields of evolution and development.

Version 1:

Reviewer comments:

Referee #1

(Remarks to the Author)

The authors have thoroughly and carefully considered the points raised in my initial review, and they have made meaningful improvements to the manuscript. Points 1b, 1c, 1d, and 1e have been adequately addressed. These revisions have strengthened the modeling component of the work and further clarified how opposing flows may lead to tissue conflicts resulting in buckling.

Regarding points 1a and 2, I acknowledge the authors' efforts to address them to the best of their ability. Concerning point 1a, I still believe that defects in patterning cannot be formally excluded as a contributing factor. Point 2 was understandably challenging to address experimentally, but it would have provided more direct support for the authors' central claim.

In its current form, the main conclusions of the manuscript rely on the companion study by Dey et al., which presents compelling evidence for a specific role of *eve* and, using optogenetic approaches, delineates the contribution of the cephalic furrow to developmental robustness. The Vellutini et al. manuscript complements this work by providing theoretical modeling that strengthens and extends these observations. Overall, the two manuscripts complement each other by combining experimental and theoretical evidence to support the idea that mechanical instabilities can shape morphogenetic evolution. The authors have revised their manuscript to more carefully frame their conclusions and explicitly acknowledge that their key claims are, in part, supported by the Dey et al. study. I also note that the two other reviewers are supportive of the significance of this work for the evo-devo field.

In summary, I am happy to support publication of this manuscript in *Nature*, contingent on the acceptance of the companion manuscript by Dey et al.

That said, the authors should consider the following points in their final revision:

- Title: The current title remains overly conclusive, particularly since the manuscript does not present direct evidence for evolutionary constraint driven by mechanical conflict. I recommend that the authors temper the title accordingly.
- Page 4, final sentence of the Introduction: The sentence "Our findings reveal an interplay..." should be revised to align with the more cautious tone of the abstract—e.g., by stating that the findings suggest such an interplay

(Remarks on code availability)

Referee #2

(Remarks to the Author)

I thank the authors for their thoughtful answer and for their clarification regarding the high frequency of ectopic folds in wild-type embryos.

Although I had not raised this point, I also enjoyed reading the Appendix addressing Reviewer 1's comment on the potential detrimental effects of a specific CF ablation. I found the 'evil twin' experiments remarkably creative. I commend the authors both for designing it and performing it, and for being upfront in stating that it does not yet allow for specific CF ablation. Coming to terms with what must have been a disappointing outcome after so much effort cannot have been easy.

Nonetheless, the morphogenetic importance of the CF as a buffer is in my view convincingly (and independently) demonstrated by Dey et al.'s optogenetic experiments. I think evidence from both manuscripts needs to be considered together, is compelling as a whole, and that the present manuscript contributes crucial and complementary insights (notably from simulations). I thus strongly support its publication.

(Remarks on code availability)

Referee #3

(Remarks to the Author)

As indicated in the original review, this is a most significant study on a difficult, but very important, emerging subject: the relationship between gene activity and morphogenesis. The prevalent view, even if it is not openly articulated, is that morphogenesis is a linear consequence of gene activity. Part of the problem with challenging this view is that there are no simple systems to study it and that, even if there were, it is experimentally difficult. While some hints have emerged from in vitro systems that the relationship is likely to be complex, tests in vivo are rare or non-existent.

In this work, and the accompanying paper, the authors focus on a simple, easily overlooked, feature of early *Drosophila* development, the Cephalic Furrow (CF), to test the relationship between genes and morphogenesis and, importantly, the role that mechanics might have as a driver or evolutionary novelty. They suggest that the CF arose as an evolutionary novelty in response to mechanical instabilities that increase the fitness of the organism. Although they don't put it like that but it seems as if the close relationship between the overlap between *btd* and *eve* is a way to canalize the mechanical response of the system.

The reviews of the original manuscript were positive but some of them raised a number of detailed technical issues that, if properly addressed, would strengthen their argument which is (and remains) largely, correlational. However, it is built on a number of independent experiments and observations. Furthermore, the accompanying manuscript provides further evidence in this direction.

The authors have gone a long way to accommodate the requests, particularly the technical ones, of the reviewers. There is now new data about the relationship between Mitotic Domains and the CF, a detailed analysis of mechanical features of cell ensembles, new data on compression at the single cell and group level, an expansion of their model and also of the genetic analysis. Furthermore, in response to an important comment from reviewer 1, the authors have done what I would refer to as a heroic effort to separate the genetic and the mechanical components associated with CF formation cleanly but, as they acknowledged, have not succeeded to the extent that they would have liked. Nonetheless, their efforts, have added some additional indirect evidence for their hypothesis. Their efforts are described in an extensive appendix that should be published together with the manuscript, maybe online.

There are a few issues that remain unexplained e.g. the increase in the number of buckles in heterozygotes for *btd*, *eve* and *prd* relative to homozygotes but, as the evidence is well presented I see this as a strength of the paper. It is possible that this observation reflects features of the interaction between the genetic and mechanical components of the system that warrant further study and it is good that it is pointed out.

This is an important piece of work as it links development and evolution from a novel perspective. I also feel that together with Dey et al. (accompanying manuscript), this makes a strong case for an important issue from the evolutionary perspective.

Alfonso Martinez Arias

(Remarks on code availability)

Response to reviewers

We have carefully addressed the comments raised by the reviewers, performing a series of additional experiments and analyses that contribute to consolidating our main findings. We have streamlined the text to clarify the relation between model and experiments, which is now divided into three clear sections: in vivo experiments and analyses, model simulations, and gene expression. Our updated manuscript includes new data on:

- **Mutant phenotypes:** We revised and consolidated the analysis of ectopic folding in the different cephalic furrow mutants, quantified the relative timing between mitotic domains and ectopic folding, and added a new figure to highlight the individual variability of ectopic folding.
- **In vivo experiments:** We performed additional laser ablation experiments to show evidence of tissue compression and reorganized the figure panels for clarity.
- **Model and simulations:** We updated all the panels in the simulation figures to show the most representative seeds, compared our reference bending rigidity value to existing in vivo measurements, and performed a whole set of new simulations to analyze the effect of cephalic furrow position and timing in preventing mechanical instability.
- **Gene expression:** We performed in situ hybridization experiments in two other dipteran species to expand the taxon sampling in our comparative analysis, adding several new figures and panels to better characterize the differences between the developmental expression of cephalic furrow genes across the dipteran tree.

We spared no efforts to address the main concern raised by the Reviewer #1: “demonstrate conclusively that mechanical instabilities are a significant selective pressure for the evolution of morphogenetic processes”. We devised three high-risk, high-gain approaches to prevent cephalic furrow formation without disturbing genetic interactions or other embryonic processes. This is a first, required step for assessing the detrimental effects of mechanical instability to the fitness of individuals without the cephalic furrow. Over the past year, we established new transgenic lines and performed experiments using split-GAL4 system, FLP-FRT recombination, and hypoxia-induced interference. Unfortunately, our attempts to prevent the invagination in a large number of embryos in a controlled manner without affecting other processes encountered technical hurdles. While we made significant advances, for example managing to delay cephalic furrow formation without affecting the germ band extension using local hypoxia, the data is still preliminary and needs further characterization and optimization. Therefore, the data will not be included in the manuscript. Please see our response below and the detailed report at the bottom of this document for a complete overview.

Overall, we have substantially improved the manuscript with the new analyses and experiments and would like to thank the reviewers for their constructive feedback.

Referee #1 (Remarks to the Author):

The authors have framed their work around a question that lies at the interface of the fields of morphogenesis and evolution. The central question can be summarized as follows: While mechanical forces are essential for

morphogenesis, are they also relevant for the evolution of morphogenetic processes? To address this question, the authors conducted an investigation into the function of the cephalic furrow (CF) in Drosophila, a transient and deep fold that forms at the boundary between the head and the trunk during early embryogenesis. They reported that in the absence of genes controlling CF formation, additional extra folds of greater magnitude are produced in response to concomitant embryo elongation and local cell proliferation. Using simulations, they confirmed that compressive stresses alone are sufficient to promote extra and ectopic fold formation. As a result, they have proposed that the absence of CF triggers stronger mechanical instabilities. Furthermore, the authors compared the head-trunk genetic patterning system in Drosophila to that of the dipteran fly Clogmia albipunctata and observed that the lack of expression of btd correlates with the absence of CF formation. Based on these findings, the authors have suggested that mechanical instabilities could act as a selective pressure for the evolution of the CF.

While the genetic and mechanical analysis of the function and role of the CF is valuable, there are opportunities to enhance the study further. Specifically, the authors could: (i) Provide a more comprehensive characterization of the mechanisms by which changes in tissue mechanics could promote ectopic morphogenetic processes. (ii) Offer a deeper substantiation of how concomitant morphogenetic processes could be buffered by spatiotemporally patterned fold formation. While the first part of the manuscript has room for improvement, the data provided by the authors on the evolution of morphogenetic processes primarily support the idea that btd is necessary for CF formation. However, it remains to be demonstrated conclusively that mechanical instabilities are indeed a significant selective pressure for the evolution of morphogenetic processes. Consequently, I am unable to support the publication of this work in Nature.

We thank the reviewer for the positive appraisal of our genetic and mechanical analyses, and for the suggestions to enhance our study.

Following these suggestions, we performed new analyses and simulations to better substantiate the mechanisms promoting and preventing ectopic folding in embryonic tissues, and also incorporated data from two new species to our comparative expression analysis of cephalic furrow genes. The data extends and corroborates our initial findings about the mechanical role of the cephalic furrow during gastrulation and the importance of the head-trunk domain of btd for the evolution of this invagination.

As mentioned above and explained extensively below, our attempts to “demonstrate conclusively that mechanical instabilities are a significant selective pressure for the evolution of morphogenetic processes” only provide initial data for preventing the formation the cephalic furrow in a highly controlled spatiotemporal manner. Blocking the cephalic furrow formation on an embryo population level without disturbing gene expression or other embryonic processes is a first, required step for evaluating the fitness between individuals with and without the cephalic furrow. However, it proved to be extremely challenging.

To address the main concern raised by the reviewer—demonstrating conclusively that mechanical instabilities are a significant selective pressure for the evolution of

morphogenetic processes—we coordinated with the other team (Dey et al.) to use complementary attacks on this hard problem. While they have extended their initial optogenetic approach, we designed and performed three high-risk experiments using genetic and hypoxia-induced interference (the hypoxia angle emerged from a striking observation we made while executing the genetic strategies).

Our point-by-point response discussing these updates is detailed below.

1. Role of the CF in the control of extra-fold formation.

a. In their manuscript, the authors have provided a more comprehensive characterization of the phenotypes associated with btd and eve, along with an analysis of the prd gene's phenotype. Notably, the authors' data support that both in wild-type and mutant conditions, extra folds form in the head and trunk regions. They have also demonstrated that the number and shape of these extra folds are more pronounced in mutants compared to the wild-type. However, a limitation of the genetic analysis performed by the authors is that they did not differentiate whether the loss of mechanical forces associated with fold formation or a distinct function of btd or eve alone is sufficient to trigger extra-fold formation. While Dey et al. only commented on the formation of a misplaced fold in btd or eve mutants, they indeed utilized a specific eve mutant that affects only its expression in the head-trunk stripe, along with optogenetic inactivation of Rho, to demonstrate that a misplaced fold can indeed form in the head-trunk region under such mutant conditions. This approach allows for a more focused study of the role of local gene patterning and the contribution of local contractility.

The reviewer raises the fact that our observations are based on genetic mutants, and that this makes it difficult to distinguish the origin of the phenotype we describe for cephalic furrow mutants: the formation of ectopic folds. We agree with the reviewer that this is a limitation of our study and that knocking down eve stripe 1 or using optogenetics are more specific perturbations. However, we argue that our data and observations based on the btd and eve mutants are sufficient to draw the link between the absence of the cephalic furrow and the appearance of ectopic folds. First is the fact that the phenotype and processes that we describe are occurring at a short timescale of about 30min after gastrulation (3hpf on stage 6). The defects in head and trunk formation known for btd and eve mutants, respectively, become visible 2-3h after gastrulation with the appearance of segmental furrows (5-7hpf on stage 11). If the ectopic folds that we observe formed late in development, it would indeed have been difficult to make any association with the absence of the cephalic furrow. However, ectopic folds form in a limited time window soon after gastrulation making it possible to draw this connection. We have seriously considered and pursued the alternative hypothesis that ectopic folds are not the result of passive buckling, but folds that actively invaginate as a result of mispatterned cells in btd and eve mutants. If ectopic btd or eve expression would be responsible for triggering ectopic folds, we would expect ectopic expression of these genes to match the distribution of ectopic folds. However, our data does not support this interpretation for several reasons. First, the expression pattern of btd in eve mutants does not change, while the expression of eve in btd mutants only exhibits a fainter stripe 1 (Extended Data Fig. 6g). Neither prd nor slp expression in btd or eve mutants exhibit a pattern consistent with the distribution of ectopic folds. While we cannot rule out the possibility of other mispatterned genes to be generating ectopic folds, we can at least conclude that none of the cephalic furrow genes would be involved. Second, is the fact that the formation of ectopic folds is variable within individuals of the same genotype (Fig.1f, Extended

Data Fig. 2a, Supplementary Fig. 1). This indicates that the folding events are not patterned, but a stochastic process that is a result of dynamic tissue interactions. Third, we have analyzed the formation of ectopic folds in different genetic backgrounds (btd, eve, prd, stg and slp mutants and wildtype embryos), which have severe or mild perturbation to cephalic furrow formation. We observe ectopic folding is more prominent in embryos that fail to shorten the initiator cells and invaginate the cephalic furrow (btd and eve mutants). Finally, the phenotype we describe using genetic mutants were corroborated by Dey et al. using RNAi approaches as well as the more specific eve stripe 1 knock down and optogenetic experiments, which altogether provides solid evidence for the phenotype that we describe: the absence of the cephalic furrow results in the formation of ectopic folds around the head-trunk boundary.

b. The statement, "Mitotic expansions always precede ectopic folding (Figure 2c, Figure S1a)," would benefit from further support through quantification.

We thank the reviewer for this interesting and relevant suggestion. To determine if mitotic expansions always precede ectopic folding, we quantified the relative timing between mitotic expansions and ectopic folding in btd and eve mutants. We defined the start of a mitotic expansion as the first frame where we detect an increase in the apical area of blastoderm cells. In turn, the start of an ectopic fold is the first frame where we detect the infolding of the tissue. To standardize the data across multiple embryos, we converted frames into minutes and normalized it using the relative time after gastrulation (min).

In wildtype embryos, the formation of the cephalic furrow (CF=7.2±2.2 min) always precedes the formation of mitotic domains (MD=14.6±2.4 min). The difference is statistically significant in a Mann-Whitney test (p=0.000432, n=16). We obtain the same pattern for heterozygote embryos (CF=10.4±2.6 min and MD=16.6±3.2, p=0.00164, n=13). However, in homozygote embryos, the pattern is reversed. We find that the formation of mitotic domains (MD=17.0±2.6 min) always precedes the formation of ectopic folds (EF=21.9±3.6 min) (p=0.001953, n=10). This quantification corroborates our original statement and provides support that mitotic expansions always precede ectopic folding.

We included the relative timing data for mitotic domains and ectopic folds into Table 1 and created a new panel in Fig2d for visualizing this difference.

c. The authors mentioned, "To estimate the forces acting on the infolding tissue, we measured the rate of tissue deformation (strain rate) at the head-trunk boundary using particle image velocimetry." This assertion is critical in demonstrating that physical forces are the driving factor behind the formation of ectopic folds. While I acknowledge that the strain rate can be measured by PIV, it remains unclear how one can estimate the forces acting on the infolding tissue solely based on the strain rate. In my opinion, the analysis of the strain rate offers a quantification of the observed phenotype but does not inherently provide an estimation of the forces acting on the infolding tissue. Would it be possible for the authors to consider using laser ablation to further substantiate this point?

We agree with the reviewer that experimental validation of compressive stress will strengthen our conclusion. While laser ablation of cell membranes is a good method to estimate tension, it cannot be used to estimate pressure in tissues: ablating the membrane of a cell under pressure would result in no recoil, regardless of the amount of pressure. Since current methods to directly measure pressure in vivo (e.g.,

deformable oil/polyacrylamide/ferrofluid microdroplets) remain under active development and have not been optimized in *Drosophila* embryos, we designed a modified laser ablation protocol to test for the presence of compression in the blastoderm tissues. We performed large laser cuts across the apical membrane of multiple cells, orthogonal to the presumed force vectors, in the region between the germ band tip and the mitotic domains. If the region is under compression, as our initial evidence suggests, we would expect cells on each side of the cut to collapse onto themselves over a longer timescale. While this type of longer laser cutting is challenging, as one needs to strike several cells without bursting the tissue, we were able to consistently perform the experiments in wildtype embryos. To quantify the tissue dynamics after the large-scale cut, we tracked several pairs of vertices from non-ablated cells on each side of the cut and calculated the distances between them over time. We find that in non-ablated control embryos, the distance between cell vertices in the same region remains unchanged, whereas in ablated embryos the distance decreases for several minutes after the cut. In our interpretation, this indicates that the surrounding tissues are collapsing over the row of ablated cells, which is congruent with the hypothesis that the region between MD6 and the germ band is under compression.

We have edited the text to include these results and added a panel in Fig2g with the quantification.

d. The authors stated, "Overall, our in silico modeling and in vivo experiments suggest that the epithelial buckling at the head–trunk interface in cephalic furrow mutants only occurs when both the mitotic domain expansion and the germ band extension happen concomitantly." I believe this conclusion may be overstated. The authors have experimentally shown that both factors are necessary, but their in silico model allows them to adjust the bending energy to achieve this regime. However, there is no evidence in the model that both factors are required. The modeling primarily demonstrates that compressive stress can promote bending within a certain range of compressive stress and bending energy (an intuitive result), yet it does not definitively establish that the observed compressive stresses in vivo are sufficient to trigger mechanical instability.

We agree with the reviewer that our presentation of the interplay between experiment and simulation led to an insufficiently supported overall claim. To address this, we rearranged the text sections to clarify the relation between in vivo and in silico results. The parameter sweep that we describe in Fig4e,f reveals the dynamics of tissue folding across different bending rigidities. However, we do not know a priori, and our model cannot determine by itself, where in this parameter space our embryo is. The in vivo experiments are crucial for that. The double mutant experiment (cephalic furrow and mitotic domain loss of function) shows that in a condition without mitotic domains and only the germ band, there are no ectopic folds. Based on this information, we can find the bending rigidity that recapitulates this dynamics—no ectopic folding in germ-band-only or mitotic-domain-only conditions. For that, we determined at which bending rigidity the ectopic fold number becomes lower than 1 (=no ectopic folding in vivo). For both conditions, this is around the bending rigidity of $K_b^* \approx 1.0 \times 10^{-4}$ (Extended Data Fig. 5c,f). With this value, we can estimate where our embryo is in the parameter space of our model. This reference bending rigidity is essential for connecting the model with experiments, providing a biologically relevant reference for the subsequent simulations. It is also in the same order of magnitude of existing direct measurements in other epithelial monolayers (see next point). Using this reference bending rigidity, we asked if the presence of both mitotic domains and germ band

results in more ectopic folding compared to the mitotic-domain-only and germ-band-only conditions. We find that when both are present, the number and speed of ectopic folding is maximized, especially around 20% of germ band extension (Extended Data Fig. 5g,h). These simulations suggest that the combined activity of mitotic domains and germ band facilitates ectopic buckling. Since the main aspect of this condition is an increased compression between particles, it provides a theoretical link between increased compressive stresses and ectopic buckling. For this reason, we stated that both simulations and experiments support this finding, and hypothesized that a similar mechanism of increased compressive stress may explain the ectopic folding in vivo.

e. The authors wrote, "Taken together, our physical model provides a theoretical basis that an early patterned invagination can effectively absorb compressive stresses in the tissue, preventing mechanical instabilities in the embryo epithelium during gastrulation." It would be valuable if the authors could go beyond this theoretical analysis by verifying whether the predictions of the model semi-quantitatively align with the experimental observations.

As discussed above, the simulations that we performed to evaluate the function of the cephalic furrow use a bending rigidity value that was based on our experimental data. This reference value reveals where the embryo resides in the parameter space of our model. Simulations within this parameter space are biologically relevant and align semi-quantitatively with the experimental observations. We were, however, curious how our reference bending rigidity compares to direct measurements of bending rigidity in other epithelial monolayers. Performing such measurements in the blastoderm epithelium of *Drosophila* is experimentally challenging. Therefore, we compared the data from two studies on MDCK monolayers (Trushko et al. 2020 and Fouchard et al. 2020). They obtained bending and stretching rigidity estimates of $K_b \approx 5 \times 10^{-13} \text{ Nm}$ and $K_s \approx 0.2 \text{ Nm}^{-1}$, respectively. Given that in elastic sheets the bending/stretching ratio changes proportionally with the square of the height ($K_b/K_s \propto h^2$) (Efrati et al. 2009), we adjusted the calculation to account for the difference in thickness between *Drosophila*'s blastoderm (40 μm) and MDCK monolayers (18 μm). We obtained a value of $K_b^* \approx 1.8 \times 10^{-4}$. This is higher than our reference bending rigidity of $K_b^* \approx 1.0 \times 10^{-4}$, but still in the same order of magnitude. This indicates that the parameter space that we obtained from the experimental data and used in the simulations is not entirely off charts, but consistent with existing direct measurements of bending rigidity. We have included the comparison of bending rigidities in the text.

2. Role of mechanical instabilities as a selective pressure.

*Expanding on their assertion that the cephalic furrow (CF) plays a role in buffering mechanical instabilities associated with cell proliferation and gastrulation, the authors delved into the evolution of genetic patterning at the head-trunk boundary. They conducted a comparative analysis of gene expression patterns between *Drosophila* and the dipteran fly *Clogmia albipunctata*, a species from the basally-branching Psychodidae family that lacks a cephalic furrow. In their analysis, the authors identified *slp* as an additional regulator of the expression pattern of the *eve* stripe 1 and *btd* and proposed that the combined expression patterns of *slp*, *prd*, *eve*, and *btd* define the unique transcriptional identity of the CF. They proceeded to compare and contrast the expression profiles of *slp*, *prd*, *eve*, and *btd* in the two species, highlighting that the key differentiation lies in the expression of *btd*. As a result, they concluded that *btd* represents a genetic innovation crucial for cephalic*

formation. In their abstract, they further surmised that "These data suggest that the genetic control of cephalic furrow formation was established through the integration of a new player into the ancestral head-trunk patterning system, and that mechanical instability may have been the selective pressure associated with the evolution of the cephalic furrow." To substantiate the initial claims regarding genetic patterning, it would be relevant to perform similar analyses in additional species.

Following the reviewer's suggestion, we expanded our gene expression analysis to two other dipteran species, each sampled from a different region of the phylogenetic tree. One is the mediterranean fruit fly *Ceratitis capitata*, a species from the Tephritidae family known to have a cephalic furrow. The other is the malaria mosquito *Anopheles stephensi*, a species from the Culicidae family, where no cephalic furrow has been observed. Our sampling now includes two non-cyclorraphan lineages that diverged before the evolution of the cephalic furrow (*Clogmia* and *Anopheles*), and two cyclorraphan lineages that diverged afterward (*Ceratitis* and *Drosophila*) (as described in Dey et al. 2023). The new results corroborate our previous claims about the evolution of the genetic patterning of the cephalic furrow and provide a more granular and complete overview of the gene expression diversity at the dipteran head-trunk boundary.

First, the data shows that the expression of *slp* and *eve* is highly conserved. In the four species that we analyzed, the expression domains of these two genes demarcate the head and trunk regions of the embryo early in development (Fig. 5 and Extended Data Fig. 6). Moreover, the interface between the domains gives rise to the head-trunk boundary at the onset of gastrulation. This spatiotemporal similarity provides further evidence that *slp* and *eve* are important players in the patterning of the dipteran head-trunk boundary.

Second, it corroborates the hypothesis that changes in *btd* expression are associated with the evolution of the cephalic furrow. While *Drosophila* and *Ceratitis* show a *btd* head-trunk territory that overlaps with *eve* stripe 1, *Anopheles* and *Clogmia* do not have a *btd*-*eve* overlap at the head-trunk boundary. In *Anopheles*, *btd* is expressed near the head-trunk boundary, but more anterior than *eve* stripe 1, a pattern similar to what has been described in *Chironomus*, another non-cyclorraphan diptera without cephalic furrow (Dey et al. 2023). In *Clogmia*, as we reported previously, the *btd* territory at the head-trunk boundary is entirely absent. This broader comparative picture, therefore, shows a clear correlation between the presence of a *btd*-*eve* overlap and the presence of the cephalic furrow, suggesting that the evolution of this invagination is associated with changes in the expression of *btd* at the head-trunk boundary.

We have updated the text and consolidated the figures about the gene expression analyses, including new panels of altered patterns in mutant embryos and the expression along the developmental sequences of the new species (Fig. 5, Extended Data Fig. 6 and 7).

*However, the second claim, and indeed the central claim of the manuscript, revolves around the role of mechanical instabilities as a selective pressure in the evolution of the cephalic furrow. The authors have suggested that the loss of *btd* leads to mechanical instabilities, but they cannot definitively rule out the existence of other selective pressures associated with *btd* function.*

We agree with the reviewer that other selective pressures linked to *btd* function cannot be ruled out. In fact, our working hypothesis assumes that the changes in the

expression of *btd* at the head-trunk boundary was indeed the result of other selective pressures, probably related to the reduction of the larval head in cyclorraphan flies (Schneeberg & Beutel, 2014). Once established, the *btd*-*eve* overlap might have provided the genetic substrate to be coopted into the cephalic furrow program. Therefore, the establishment of a *btd*-*eve* overlap at the head-trunk interface and the origin of the cephalic furrow morphogenetic program were potentially two independent events. We argue that the mechanical instability may have acted as a selective pressure, leading to the cooption of independently established *btd*-*eve* genetic landscape. This is consistent with the findings that lineages without the *btd*-*eve* overlap evolved alternative solutions to increased mechanical instability (Dey et al. 2023).

Additionally, while the theoretical model provides evidence that compressive stress can promote extra-folding in conjunction with concomitant morphogenetic processes, it remains to be demonstrated whether these mechanical instabilities could indeed be detrimental to embryo development, thereby acting as a selective pressure. I am uncertain about how one can directly and conclusively establish such a claim. At a minimum, the authors should consider exploring whether: (i) The absence of CF, triggered without affecting genetic patterning, leads to changes in the robustness of development. (ii) Correcting such mechanical instabilities through alternative means can subsequently rectify the observed developmental defects and enhance viability. While I acknowledge that these experiments are complex and challenging, they are crucial in substantiating the main claim and novelty of the work.

We agree that demonstrating that the absence of the cephalic furrow is detrimental to the embryo's fitness would provide strong evidence in favor of the hypothesis raised in our discussion. However, as noted by the reviewer, testing this experimentally is technically complex and challenging. There are two required steps. First, a method to prevent cephalic furrow formation with precise spatiotemporal specificity that does not perturb the underlying genetic interactions. Second, an assay to quantify the fitness of individual embryos with large sample sizes to overcome developmental variability. While we considered establishing these experiments as beyond the scope of the current paper, we took the reviewer's feedback as an opportunity to expand our analysis, and attempted a clean inhibition of the cephalic furrow using three different approaches. We provide a summary of our findings in the paragraphs below, but have compiled a detailed report with our results in the **Appendix** section at the end of this document.

We reasoned that, to affect the rest of the embryo the least, we should target a downstream regulator of the signaling network that controls the invagination. The cephalic furrow invagination is driven by the activity of lateral myosin in the initiator cells (Eritano et al. 2020). Therefore, we decided to target Rho1, one of the downstream-most regulators of myosin, by driving the expression of a dominant-negative form of Rho1 (DNRho1) to compete with the native protein and inhibit myosin activity specifically in the initiator cells. With this, we expected to inhibit the mechanism driving the invagination without affecting the genetic patterning or differentiation of these cells during subsequent developmental stages. To drive the expression of DNRho1 specifically in initiator cells, we designed two approaches (Appendix Fig. 1). The first using *Drosophila*'s split-GAL4 system to activate DNRho1 in cells expressing both *btd* and *eve* (i.e. the initiator cells). The second, generating a transgenic line with a FLP-out cassette to activate DNRho1 expression under the control of the enhancer of *eve* stripe 1 (Appendix Fig. 3 and Fig. 4). Unfortunately, neither approach proved effective to inhibit cephalic furrow formation. Embryos containing both hemidriviers and responder (Appendix Fig. 2 and Table 1) or the

flipped-out cassette in its active form (Appendix Fig. 5 and Table 2) show no cephalic furrow defects. We believe that this is due to a combination of factors. Since the cephalic furrow initiates early at the onset of gastrulation, the time required for the drivers to become active was probably a limiting factor for the production of enough DNRho1 before gastrulation in the GAL4 experiments. Moreover, because we observe no cephalic furrow defects in our transgenic line where DNRho1 expression was directly under the control of the enhancer of *eve* stripe 1—thus likely expressed at the onset of gastrulation—we suspect that the amount of the dominant-negative form of the protein was insufficient to prevent the activity of the native form. Optimizing the construct or targeting a different molecule might yield better results in the future.

[TEXT REDACTED]

[TEXT REDACTED]

As mentioned above, we prepared a report at the bottom of this document describing our three approaches to prevent the formation of the cephalic furrow without affecting other embryonic processes. These experiments provide a solid starting point and significant advances for addressing the role of tissue mechanics for the evolution of morphogenesis. However, they cannot yet be used to address the reviewer's concern due to technical hurdles that still needs years of research to be addressed. Mainly, none of the experiments cleanly blocked the invagination. We will include the split-GAL4 and transgenesis data as supplementary information. The hypoxia data will become part of a separate publication.

Given this, we have edited the discussion to provide a more balanced, toned down view of the current state. When raising the hypothesis, we now indicate that this is based on comparative data from both papers and on the optogenetic experiments showing midline distortions from Dey et al. 2023.

Referee #2 (Remarks to the Author):

This is a beautiful study that combines light-sheet live imaging, genetics, state-of-the-art image analysis and morphometrics, analysis of gene expression, and computer simulations to unravel the developmental function and the evolutionary origin of a mysterious structure: the cephalic furrow (CF). Anyone who has ever looked at Drosophila embryos is aware of the obvious and conspicuous presence of the CF, and has also had to come to terms with the frustrating fact that this massive structure is both transient and has no clear function. This study elegantly solves this conundrum, by hypothesizing that CF invagination absorbs compressive stress generated by other morphogenetic movements (cell division within mitotic domains and germ band extension). The authors show that, when CF formation is genetically abolished, ectopic folds (EF) form instead, with more variable timing, shape and location. They also show that EF formation in those mutants requires both mitotic domains and germ band extension - so that it's really "tissue collision" that generates those folds, and that CF formation normally adverts. The authors conclude by a comparison to a dipteran without CF (Clogmia) and by an apt reference to the Newman-Müller theory on the origin of developmental patterning. Thus, the paper presents a completely novel hypothesis (to my knowledge), with a dual broad significance: (1) elucidating the biological meaning of a long-known structure in a major model organism; (2) showing an example of a structure that likely evolved as a self-organized byproduct of physical forces, before its formation evolved to be programmed and genetically patterned (as per Newman & Müller).

Overall I am convinced by the authors' case and feel very supportive of the paper. However, a number of small points are unclear to me and I am taking the opportunity to suggest that the authors clarify them.

We thank the reviewer for the positive appraisal of our work and for the constructive comments. Our detailed response to the individual points is below.

Scientific questions:

*- The high frequency of EF formation in wild-type (WT) embryos (80%) is perplexing, and contrasts with the lower frequencies observed in *btd*, *eve* and *prd* heterozygous embryos (12.9, 12 and 25.9% respectively) - even though these heterozygous embryos are phenotypically WT (in my understanding). Even *prd*^{-/-} embryos have lower EF frequency than the WT (42.9 vs 80.6%). Fig. S2j shows that EFs in WT flies only occupy a small area compared to mutants, but their very high frequency remains puzzling. Could the authors comment on that difference? Does something about having a half-dosage of *eve/prd/btd* reduce the frequency of EF? Alternatively (and perhaps more parsimoniously), might this be due to a difference in genetic background between strains unrelated to the loci of interest? If so, could one expect the frequency of EFs to vary between different nominally "wild-type" *Drosophila* strains?*

We were also surprised by the higher frequency of EFs in wildtype embryos compared to heterozygote embryos, considering that both form a cephalic furrow. To investigate this difference, we thoroughly reviewed our recordings to ensure the consistency of our scoring of ectopic folds across datasets. Using our high temporal resolution in toto datasets, we can identify putative ectopic folds in 3D lateral renderings and objectively confirm their existence by checking the profile view of the region in our 3D cartographic projections. Building upon the previous data, we classified individual folds regarding their identity (CF or EF) and their position relative to the head-trunk boundary (anterior, mid, posterior) consistently for the different genetic backgrounds and zygositys. This reanalysis consolidated the statistics around the frequency and number of folds. Most importantly, we found that we had underestimated the frequency of ectopic folding in *btd* and *eve* mutant heterozygotes. The values changed from 12.9 to 18.2% for *btd* and from 12.0 to 26.9% for *eve*. This was mostly due to ectopic folds forming at the posterior region (between MD6 and GB) which we had not consistently scored in our first analysis. Nevertheless, apart from this difference, the other values have remained the same overall, corroborating our initial results. Therefore, the frequency of EFs is indeed higher in wildtype (~78%) compared to mutant heterozygotes (~27%) (see Table 4).

We had already identified differences between wildtype and heterozygote embryos in our folded area and depth analyses. For example, while the average CF area in wildtype (11.4 ± 1.2 , $n=16$) is not significantly larger than in *btd* heterozygotes (10.3 ± 1.9 , $n=6$, $p=0.133515607$), it is larger than in *eve* (9.7 ± 1.1 , $n=7$, $p=0.00053843$) and *prd* (7.6 ± 2.6 , $n=9$, $p=0.00009496$) heterozygotes (see Fig1h and Table 2). Moreover, there is a difference in CF maximum depth between *btd* (71.6 ± 8.0 , $n=32$) and *eve* (59.0 ± 6.8 , $n=20$, $p=0.000000001$) (see Extended Data Fig. 2f and Table 3). While we suspect that the differences in CF area and depth in the *eve* line might be due to *eve*-specific dosage effects, we cannot exclude that these differences might be due to unrelated loci within the genetic background of these strains. The original mutant lines come from different bloomington stocks, and have different green balancers on the first (*btd*) and second (*eve* and *prd*) chromosomes. Nevertheless, there is one genetic difference between wildtype and mutant lines that

could explain the different in EF frequency. While they all have the same membrane marker, the chromosome holding the locus is in the wildtype was outcrossed from the original fluorescent strain. In that case, the difference would be due to general genetic background differences. Unfortunately, we were unable to re-generate the mutant lines with the wildtype chromosome or re-acquire the datasets due to focus on the other experiments for the revision.

Independent of the differences being due to genetic background, we sought to identify possible mechanisms that would lead to higher frequency of ectopic folding in wildtype embryos. First, we hypothesized that differences in germ band speed could account for differences in EF frequency. The idea is that a faster germ band extension in the wildtype lineage, due to less genetic baggage like balancers, might make ectopic folding more likely. Therefore, we estimated the speed of germ band extension in the different strains. However, we found no significant difference in the germ band speed between wildtype ($10.2 \pm 0.5 \mu\text{m}/\text{min}$, $n=16$) and *btd* ($10.4 \pm 1.7 \mu\text{m}/\text{min}$, $n=9$, $p=0.798641249$), *eve* ($10.6 \pm 1.0 \mu\text{m}/\text{min}$, $n=7$, $p=0.569222418$), and *prd* ($10.5 \pm 1.2 \mu\text{m}/\text{min}$, $n=10$, $p=0.579080902$) heterozygotes. Therefore, germ band speed is not a contributing factor to the higher EF frequency in wildtype.

- Related to the above: l. 240: “Compared to wildtype, the ectopic folds in *stg* mutants are less frequent” This is true, but only because the WT have a very high frequency of EF formation. The difference between heterozygous and homozygous *stg* mutants is actually small (27.3% vs 23.1%, 33 vs 13 embryos) and perhaps not significant. I suggest removing this point or arguing it differently.

We removed this point.

- l. 333: “The function of the cephalic furrow in preventing mechanical instabilities depends on the correct positioning and timing of the invagination; it must occur at the head-trunk boundary.” I am not sure the authors have shown the CF must form at the head-trunk boundary, as other positions don't seem to have been explored in the simulations or in the diverse mutant phenotypes reported (unless I missed something).

The reviewer is correct. We made observations about the cephalic furrow position during the testing of the model, but had not systematically run simulations for this purpose. Therefore, to explore the impact of the position of the cephalic furrow in preventing mechanical instability, we quantified ectopic folding with the cephalic furrow positioned more anteriorly and posteriorly along the anteroposterior axis. We find that, without germ band extension, the cephalic furrow only prevents ectopic folding at the anterior end when positioned above 50% of the embryo length (Fig4h, left column). Conversely, in simulations with the extended germ band, the cephalic furrow only prevents ectopic folding at the posterior end when positioned below 50% of the embryo length (Fig4h, right column). While the extent that the cephalic furrow can prevent mechanical instabilities in our simulations varies according to the stage of germ band extension, our data suggests that the invagination is most effective when positioned around the middle of the embryo (between 40 and 60%). In wildtype embryos, the cephalic furrow is positioned nearby at the anterior limit of this range (65%). Interestingly, in *slp* mutants, the cephalic furrow is shifted forward and forms around 73% of the embryo length, and homozygote embryos usually show a posterior ectopic fold located more dorsally even before the expansion of mitotic domains (Fig 5a and Supplementary Video 17). These mutants suggest that, similar to what we observe in our simulations, an anterior shift in the cephalic furrow may lead to more frequent ectopic folding events in posterior regions also in vivo.

- Fig. 3e-f: since mitotic domains are said to contribute to the formation of EFs, it is unclear to me why there are more EFs without mitotic domains than with them in the ($g=0.4$ $Kb=1.2$) and ($g=0.2$ $Kb=0.8$) conditions.

We thank the reviewer for pointing this out. To create the panels with the folding visualizations, we had selected and plotted random seeds from the simulations. Some of these randomly selected “embryos” were at the fringes of the data distribution, far from the average value quantified for each simulation (the data shown in the heatmaps of Fig4e,f). As a consequence, some were not accurate representations of the average value for each condition in the parameter sweep. To address this issue, we added a constraint to limit our random selection to seeds close to the mean of the data distribution. In this manner, we ensured that the individual panels show single seed visualizations that are representative of the average simulation results. We updated the panels in Figure 4e,f and Extended Data Fig. 5l using the same approach.

Discussion points:

- Could the authors speculate on the ancestral function of *btd* before the evolution of the CF? Did it already pattern some kind of invagination? Perhaps some insights can be gleaned from its acron expression domain (in *Drosophila* and *Clogmia*), from the *Drosophila* mutant phenotype or from its ventral foregut domain in *Clogmia*?

In *Drosophila*, *btd* is required for the formation of mandibular, intercalary, and antennal segments (Cohen & Jürgens 1990, Wimmer et al. 1996), the establishment of ventral imaginal discs and leg formation (Estella et al. 2003, Estella & Mann 2010), regulation of neuroblast identities (Komori et al. 2014, Xie et al. 2014), and control of mitotic formation (Momen-Roknabadi et al. 2016). However, *btd* is mostly acting upstream on the specification of cell fates via transcriptional regulation (*Dll*, *wg*, *hh*, *stg*, and *en* are some of the known targets). We have not found descriptions of acron- or foregut-related phenotypes for *btd* mutants in the literature. Therefore, in general, *btd* activity does not seem to be associated with epithelial invaginations.

Data presentation and phrasing:

- Fig. 1c only shows a single EF, fairly close to where the CF normally forms. However, the text and fig. 1f make it clear that there can be multiple EFs, sometimes very far from the normal CF location. This variability could be reported and depicted more extensively, by showing multiple examples of embryos with variable numbers and positions of EFs. Video S2 helps a bit (by showing an embryo with several EFs), but still only shows a single embryo. Some figure panels directly comparing multiple embryos would be better would be even better.

Following the reviewer’s suggestion, we prepared a new figure to highlight ectopic folding variability, showing the EF outlines on individual *btd* and *eve* embryos (Supplementary Figure 1). In addition, Supplementary Video 8 also shows EF formation in four different *btd* embryos to highlight the variable dynamics of folding.

- l. 160: CF formation is said to be a “progressive invagination (about 14 min)” while formation of EFs is said to be “abrupt (4 min)”. Looking at the figure, depending on the genotype, EFs can take up to 8 min to reach maximal/final

depth, which does not seem all that abrupt. Maybe “faster” would be a more accurate term.

We replaced “abrupt” by “faster” in the specific sentence and in the video describing the process (Supplementary Video 6).

- l. 220-221: “in stiffer conditions ($Kb^ \sim 1.2e-4$) the germ band alone, even at its maximum extension, cannot drive the formation of ectopic folds”. However I do see a small ectopic fold at $Kb^* = 1.2e-4$ and $g = 0.4$. Perhaps the authors meant $Kb^* \sim 1.4e-4$*

After replacing the panels of this parameter sweep for representative samples of the simulation (see explanation above), this inconsistency has been resolved. The samples with $Kb^* 1.2 \times 10^{-4}$ show less than one ectopic fold on average.

- l. 313-315: “Curiously, this buffering effect diminished with the extension of the germ band for intermediate values of $K0CF$ ”. I don’t quite see how this is curious. Shouldn’t the mechanical challenge created by germ band extension be expected to reduce the buffering effect of the CF? And shouldn’t that buffering effect get stronger with $K0CF$? Also 2 sentences later: “in these conditions, the forces generated by the mitotic expansions and by the germ band extension dominate over the infolding pull of the cephalic furrow.” - it’s not quite clear what conditions are referred to as “these conditions” - does it mean under intermediate $K0CF$?

We thank the reviewer for bringing up this point. We did not clearly convey why we thought this result was intriguing. Before running this particular parameter sweep, we expected that at higher percentages of germ band extension, the cephalic furrow would become deeper. This was based on the intuition that the greater mechanical challenge exerted by the extended germ band would push the cephalic furrow inwards. However, the opposite happened: the cephalic furrow becomes shallower (Extended Data Fig. 5j). We were surprised, but the explanation is, in fact, straightforward. The cephalic furrow is shallower because it is competing with ectopic folds. In these simulations, the cephalic furrow and mitotic domains form at the same time. With the greater mechanical challenge caused by the extended germ band, there is an increase in ectopic folding around the posterior region, which, in turn, inhibits the programmed invagination, specially for weaker K_0CF s. On the other hand, when we run simulations adding a delay to MD formation (Extended Data Fig. 5k), which reflects the timing of events in the real embryo, the cephalic furrow takes precedence, inhibiting ectopic folding around the invagination. In this condition, where the cephalic furrow invaginates early without the interference of ectopic folds, its depth increases with the degree of germ band extension, as we had initially predicted. This supports the idea of a germ band push influencing the final depth of the cephalic furrow, for which we find some evidence in vivo (Fig. 3e,j). To clarify all these points, we have edited the text for this paragraph in the results.

*- l. 360-366: the verbal explanation of *slp*, *eve* and *btd* expression is somewhat confusing. Fortunately Fig. 5f-i is very clear and allows to rescue understanding. Particularly perplexing is the statement l. 364 “*eve* expression is initially ubiquitous”: *eve* is clearly not expressed in all cells at mid-cycle 13, but in a broad domain with a sharp anterior boundary and a fuzzier posterior boundary.*

Thanks for pointing this out. The paragraph mixed previous observations with new results. The ubiquitous expression of *eve* was observed in a previous paper (Andrioli et al. 2012), for which we added the citation. Moreover, we included a more complete figure with additional gene expression patterns to support the results described in the text (Extended Data Fig. 6 and 7). We show, for instance, that *eve* at nuclear cycle 11 is almost ubiquitously expressed. Moreover, the entire gene expression section was revised and edited to improve the clarity of the descriptive statements.

- Fig. 3e: I think the fact that this panel depicts a mutant without CF should be written on the figure itself somewhere

Great suggestion. We added a label above the heatmap to indicate that the cephalic furrow is absent in these simulations.

- Figure S2d: the color bar depicting time is lacking a scale.

We added the timescale to the panel.

Referee #3 (Remarks to the Author):

In this manuscript Vellutini and colleagues address the function of the Cephalic Furrow (CF), a small transient multicellular pattern on the anterior third of the early Drosophila embryo which has remained a curiosity for years. The authors turn what at first sight is a somewhat insignificant feature of the gastrulating Drosophila embryo into an example of the evolution of morphogenetic mechanisms. The manuscript combines excellent genetics and experimental embryology with evolutionary biology. Having shown that the main role of the CF is to buffer mechanical stress in Drosophila, they go on to show that other flies from a different evolutionary lineage, whose embryos lack a CF, have evolved a different mechanism to cope with the same stresses. The experimental work is complemented with an interesting and carefully crafted model.

The authors start by characterising the morphological phenotypes of ectopic fold formation in gastrulating Drosophila embryos with impaired cephalic furrow (CF) formation. This leads them to identify a correlation between ectopic fold formation with mitotic domains (MD) cell division activity and germ band extension (GBE) supported by extrapolating local strain rates based on imaging data. The contribution of MD and GBE to mechanical instability is furthered explored through mechanical simulations, experimental perturbations (e.g. genetic, cauterization) and laser dissection recoil speed measurements, testing simulation predictions allowing to explore properties and developmental events contributions towards tissue mechanical instability and morphogenesis. They show that GBE is not required for CF formation and develop physical model predictions arguing that CF formation can absorb compressive stresses preventing ectopic furrow formation resulting from mechanical instabilities.

*A loss of function screen of ~50 candidate genes was then carried out to identify genes involved in CF formation in Drosophila resulting in the identification 3 new regulatory genes, with sloppy paired (*slp*) mutant flies exhibiting the strongest phenotype. Members of this family of Transcription Factors, are shown to regulate the position of the buttonhead (*btd*) and even-skipped (*eve*)*

genes in an overlapping domain that establishes the position of the CF. The authors then look at flies that form (Drosophila) or do not form (Clogmia) a CF and notice that the trunk-head boundary expression domain of buttonhead is absent from the latter with the remaining CF regulatory genes slp, eve and prd having conserved expression patterns across both species. The authors propose the acquisition of the btd trunk-head expression was a central event in the evolution of CF formation. It is also reported that Clogmia exhibits out-of-plane head cell divisions, a likely alternative mechanism to resolve mechanical instabilities within species that do not form CF. The manuscript does a good job at exploring the role of mechanical stability in developmental robustness, complemented with a mechanical model that provides an understanding of the process.

The authors provocatively argue that there might be a role for mechanical stabilities in the evolution of morphogenesis. The reported new observations and findings are of interest to the fields of evolution and development.

We thank the reviewer for the positive assessment of our study.

[TEXT REDACTED]

[FIGURE REDACTED]

[TEXT REDACTED]

[TEXT REDACTED]

[FIGURE REDACTED]

[TEXT REDACTED]

[TABLE REDACTED]

[TEXT REDACTED]

[TEXT REDACTED]

[FIGURE REDACTED]

[TEXT REDACTED]

[FIGURE REDACTED]

[TEXT REDACTED]

[TEXT REDACTED]

[FIGURE REDACTED]

[TEXT REDACTED]

[TABLE REDACTED]

[TEXT REDACTED]

[TEXT REDACTED]

[FIGURE REDACTED]

[TEXT REDACTED]

[TEXT REDACTED]

[FIGURE REDACTED]

[TEXT REDACTED]

[FIGURE REDACTED]

[TEXT REDACTED]

[FIGURE REDACTED]

[TEXT REDACTED]

[TEXT REDACTED]

[FIGURE REDACTED]

[TEXT REDACTED]

[TEXT REDACTED]

[FIGURE REDACTED]

[TEXT REDACTED]

[TEXT REDACTED]

Response to reviewers

We thank the reviewers for their constructive feedback. We have streamlined the manuscript text and figures to adhere to the formatting guidelines and addressed the remaining comments as described below.

Referee #1 (Remarks to the Author):

The authors have thoroughly and carefully considered the points raised in my initial review, and they have made meaningful improvements to the manuscript. Points 1b, 1c, 1d, and 1e have been adequately addressed. These revisions have strengthened the modeling component of the work and further clarified how opposing flows may lead to tissue conflicts resulting in buckling.

Regarding points 1a and 2, I acknowledge the authors' efforts to address them to the best of their ability. Concerning point 1a, I still believe that defects in patterning cannot be formally excluded as a contributing factor. Point 2 was understandably challenging to address experimentally, but it would have provided more direct support for the authors' central claim.

In its current form, the main conclusions of the manuscript rely on the companion study by Dey et al., which presents compelling evidence for a specific role of eve and, using optogenetic approaches, delineates the contribution of the cephalic furrow to developmental robustness. The Vellutini et al. manuscript complements this work by providing theoretical modeling that strengthens and extends these observations. Overall, the two manuscripts complement each other by combining experimental and theoretical evidence to support the idea that mechanical instabilities can shape morphogenetic evolution. The authors have revised their manuscript to more carefully frame their conclusions and explicitly acknowledge that their key claims are, in part, supported by the Dey et al. study. I also note that the two other reviewers are supportive of the significance of this work for the evo-devo field.

In summary, I am happy to support publication of this manuscript in Nature, contingent on the acceptance of the companion manuscript by Dey et al.

That said, the authors should consider the following points in their final revision:

** Title: The current title remains overly conclusive, particularly since the manuscript does not present direct evidence for evolutionary constraint driven by mechanical conflict. I recommend that the authors temper the title accordingly.*

** Page 4, final sentence of the Introduction: The sentence "Our findings reveal an interplay..." should be revised to align with the more cautious tone of the abstract—e.g., by stating that the findings suggest such an interplay*

We thank the reviewer for the detailed and thoughtful feedback.

To address the original concern 1a, we added a remark in the results to formally state that patterning defects cannot be excluded as a mechanism of ectopic folding: “While we cannot exclude that defects in patterning may contribute to the formation of ectopic folds (...).”

We have also updated the title to not be overly conclusive about the evolutionary implication of our study. The new version focuses on the developmental role of the cephalic furrow, the core of our results: “Patterned invagination prevents mechanical instability during gastrulation.”

Finally, with the shortening of the manuscript, we have removed the mentioned final sentence of the introduction but made sure the revised version maintained a cautious tone in concluding statements.

Referee #2 (Remarks to the Author):

I thank the authors for their thoughtful answer and for their clarification regarding the high frequency of ectopic folds in wild-type embryos.

Although I had not raised this point, I also enjoyed reading the Appendix addressing Reviewer 1’s comment on the potential detrimental effects of a specific CF ablation. I found the ‘evil twin’ experiments remarkably creative. I commend the authors both for designing it and performing it, and for being upfront in stating that it does not yet allow for specific CF ablation. Coming to terms with what must have been a disappointing outcome after so much effort cannot have been easy.

Nonetheless, the morphogenetic importance of the CF as a buffer is in my view convincingly (and independently) demonstrated by Dey et al.’s optogenetic experiments. I think evidence from both manuscripts needs to be considered together, is compelling as a whole, and that the present manuscript contributes crucial and complementary insights (notably from simulations). I thus strongly support its publication.

We thank the reviewer for the comments.

Referee #3 (Remarks to the Author):

As indicated in the original review, this is a most significant study on a difficult, but very important, emerging subject: the relationship between gene activity and morphogenesis. The prevalent view, even if it is not openly articulated, is that morphogenesis is a linear consequence of gene activity. Part of the problem with challenging this view is that there are no simple systems to study it and that, even if there were, it is experimentally difficult. While some hints have emerged from in vitro systems that the relationship is likely to be complex, tests in vivo are rare or non-existent.

In this work, and the accompanying paper, the authors focus on a simple, easily overlooked, feature of early Drosophila development, the Cephalic Furrow (CF), to test the relationship between genes and morphogenesis and,

importantly, the role that mechanics might have as a driver or evolutionary novelties. They suggest that the CF arose as an evolutionary novelty in response to mechanical instabilities that increases the fitness of the organism. Although they don't put it like that but it seems as if the close relationship between the overlap between bth and eve is a way to canalize the mechanical response of the system.

The reviews of the original manuscript were positive but some of them raised a number of detailed technical issues that, if properly addressed, would strengthen their argument which is (and remains) largely, correlational. However, it is built on a number of independent experiments and observations. Furthermore, the accompanying manuscript provides further evidence in this direction.

The authors have gone a long way to accommodate the requests, particularly the technical ones, of the reviewers. There is now new data about the relationship between Mitotic Domains and the CF, a detailed analysis of mechanical features of cell ensembles, new data on compression at the single cell and group level, an expansion of their model and also of the genetic analysis. Furthermore, in response to an important comment from reviewer 1, the authors have done what I would refer to as a heroic effort to separate the genetic and the mechanical components associated with CF formation cleanly but, as they acknowledged, have not succeeded to the extent that they would have liked. Nonetheless, their efforts, have added some additional indirect evidence for their hypothesis. Their efforts are described in an extensive appendix that should be published together with the manuscript, maybe on line.

There are a few issues that remain unexplained e.g the increase in the number of buckles in heterozygotes for btd, eve and prd relative to homozygotes but, as the evidence is well presented I see this as a strength of the paper. It is possible that this observation reflects features of the interaction between the genetic and mechanical components of the system that warrant further study and it is good that it is pointed out.

This is an important piece of work as it links development and evolution from a novel perspective. I also feel that together with Dey et al. (accompanying manuscript), this makes a strong case for an important issue from the evolutionary perspective.

Alfonso Martinez Arias

We thank Alfonso for the comments. We have included the Split-GAL4 and transgenesis experiments described in the appendix of the response to the data repository of the paper, and they will be available online. However, the local hypoxia experiments will be part of another study and will be published separately of the current manuscript.